# Structural recognition and stabilization of tyrosine hydroxylase by the J-domain protein DNAJC12

Mary Dayne S. Tai [1,2], Lissette Ochoa [3], Marte I. Flydal [1,4], Lorea Velasco-Carneros [5], Jimena Muntaner [3], César Santiago [3], Gloria Gamiz-Arco [1,2], Fernando Moro [5], Kunwar Jung-KC [1,2,6], David Gil-Cantero [3], Miguel Marcilla [3], Juha P. Kallio [1], Arturo Muga [5], José María Valpuesta [3,7] ✉, Jorge Cuéllar [3] ✉ & Aurora Martinez [1,2,6] ✉

Pathogenic variants of the J-domain protein DNAJC12 cause parkinsonism, which is associated with a defective interaction of DNAJC12 with tyrosine hydroxylase (TH), the rate-limiting enzyme in dopamine biosynthesis. In this work, we characterize the formation of the TH:DNAJC12 complex, showing that DNAJC12 binding stabilizes both TH and the variant TH-p.R202H, associated with TH deficiency. This binding delays their time-dependent aggregation in an Hsp70-independent manner, while preserving TH activity and feedback regulatory inhibition by dopamine. DNAJC12 alone barely activates Hsc70 but synergistically stimulates Hsc70 ATPase activity when complexed with TH. Cryo-electron microscopy supported by crosslinking-mass spectroscopy reveals two DNAJC12 monomers bound per TH tetramer, each embracing one of the two regulatory domain dimers, leaving the active sites available for substrate, cofactor and inhibitory dopamine interaction. Our results also reveal the key role of the C-terminal region of DNAJC12 in TH binding, explaining the pathogenic mechanism of the DNAJC12 disease variant p.W175Ter.

Proteostasis is regulated by the coordinated interaction of an extensive network of molecular chaperones that assist in the stabilization, folding, remodeling, translocation and degradation of proteins[1]. Chaperones function alone or in a synchronized manner, such as the family of heat shock Hsp40 or J-domain proteins (JDPs) that assist the ATP-dependent, client remodeling activity of the Hsp70 protein family[2]. These chaperones are involved in many cellular processes, not only due to different isoforms (13 encoded in the human genome, including the constitutive isoform Hsc70) but also because of the sequence diversity and variable expression profile of the many members of the

JDP family (49 human members)[3]. The greater number of JDPs indicates that a given Hsp70 can interact with several JDPs, which helps define the functional diversity of Hsp70s by recognizing, binding, and presenting specific client proteins to Hsp70s[4].

JDPs are characterized by a consensus J-domain (JD) with a conserved His-Pro-Asp (HPD) motif that stimulates Hsp70 ATPase activity[5]. Besides assisting Hsp70 in the reactivation of aggregated or unfolded proteins (disaggregating/refolding activities), some JDPs, such as DNAJA2[6], DNAJB6[7,8], DNAJB8[9], DNAJC7[10] and DNAJC8[11], also delay the intermolecular interaction of aggregation-prone protein conformers,

[1]Department of Biomedicine, University of Bergen, Bergen, Norway. [2]Neuro-SysMed Center, Department of Neurology, Haukeland University Hospital, Bergen, Norway. [3]Centro Nacional de Biotecnología (CNB-CSIC), Madrid, Spain. [4]Department of Medical Genetics, Haukeland University Hospital, Bergen, Norway. [5]Instituto Biofisika (UPV/EHU, CSIC) and Departamento de Bioquímica y Biología Molecular, Facultad de Ciencia y Tecnología, Universidad del País Vasco (UPV/EHU), Barrio Sarriena, Leioa, Spain. [6]K.G Jebsen Center for Translational Research in Parkinson's Disease, University of Bergen, Bergen, Norway. [7]Unidad de Nanobiotecnología, CNB-CSIC-IMDEA Nanociencia Associated Unit, Madrid, Spain. ✉e-mail: jmv@cnb.csic.es; jcuellar@cnb.csic.es; aurora.martinez@uib.no

which is an ATP- and Hsp70-independent, autonomous ability (holdase or holding activity).

JDPs are divided into classes A, B and C, depending on their domain organization[2]. Class A and B are formed by an N-terminal JD, followed by a Gly/Phe-rich linker region and one or two C-terminal domains (CTDs) that often play important roles in client binding[12,13]. Unique to class A is a zinc-binding domain inserted in the CTD. Class C (DNAJC) is the most diverse group of JDPs, and its members share no similarity apart from the JD[2,3]. Due to the divergence within the DNAJC class, there is little information about how they recognize and bind their clients or whether they serve other purposes than canonical JDPs. While the exact molecular mechanisms behind their function remain unclear, it is becoming more evident that the loss of protein homeostasis due to dysfunctional DNAJCs can lead to disease, notably neurological disorders[14,15].

Recently, DNAJC12 has been identified as a specific cochaperone for the tetrahydrobiopterin (BH$_4$)-dependent aromatic amino acid hydroxylases (AAAHs). This protein family is comprised of phenylalanine hydroxylase (PAH), tyrosine hydroxylase (TH) and tryptophan hydroxylases (TPH1 and TPH2)[16–18]. DNAJC12 is highly expressed in several tissues, mainly in the brain, adrenal gland, pituitary, and liver[19], where AAAHs are also typically detected. Moreover, variants in DNAJC12 cause a neurometabolic disorder, now defined as DNAJC12 deficiency, that manifests as hyperphenylalaninemia, accompanied by dopamine (DA) and serotonin depletion and parkinsonism[16,18,20,21], revealing reduced levels of functional AAAHs. Hypodopaminergia was also observed in the *Dnajc12* knockout mouse model[22]. In particular, variants in TH, the enzyme that converts L-tyrosine (L-Tyr) to the DA precursor L-3,4-dihydroxyphenylalanine (L-Dopa)[23], result in TH deficiency (THD), with phenotypes ranging from L-Dopa-responsive infantile parkinsonism with dystonia to severe encephalopathy with neonatal onset[24]. Parkinsonism and neurodevelopmental delay are also characteristic of DNAJC12 deficiency[16,20,21], and restless legs syndrome, a disorder associated with reduced DA levels, has also been reported as a clinical manifestation of DNAJC12 deficiency[25].

DNAJC12 consists of 198 residues (Fig. 1a), and the structure of the first 100 residues, including the JD, has been solved by NMR (PDB 2CTQ) (Supplementary Fig. 1a). The structure of the C-terminal half of the protein is currently unsolved, but has been modeled by AlphaFold (ID AF-Q9UKB3-F1)[26] (Fig. 1b). Apart from the JD, the other defining feature of DNAJC12 is its conserved C-terminal heptapeptide sequence (residues 192-198; KFRNYEI) which was earlier hypothesized to play a role in client recognition or catalytic functions[27]. Nevertheless, the absence of studies using isolated DNAJC12 and lack of knowledge regarding its role as a cochaperone of Hsp70s and the molecular determinants for DNAJC12 binding to AAAHs have hampered the understanding of conformational and functional effects of complex formation and the interpretation of pathogenic mutations.

Here, we report the purification and biophysical characterization of full-length DNAJC12 and the structural characterization of its complex with TH. DNAJC12 is purified as a monomer with a mostly unstructured C-terminal region that binds with high affinity to TH. The TH:DNAJC12 complex maintains TH activity and inhibition by DA and stimulates the ATPase activity of Hsc70. Importantly, complex formation also stabilizes TH and the THD-associated variant TH-p.R202H (hereafter referred to as TH-R202H) and protects them from aggregation in an Hsc70-independent manner. The complex can be purified in a stable form amenable to structural characterization by cryoelectron microscopy (cryo-EM) that provides a structural model validated by crosslinking-mass spectrometry (XL-MS). The C-terminal octapeptide RKFRNYEI is essential for DNAJC12 binding to TH, explaining the pathogenic mechanism of the disease-associated C-terminal truncated variant DNAJC12-p.W175Ter.

## Results

### DNAJC12 is a monomer with intrinsically disordered regions

In this work, we focus on and refer to DNAJC12 as the experimentally reported isoform *a*, with the canonical sequence (NM_021800.3; 198 residues)[28]. A shorter splicing variant, the 107-residue isoform *b*, has been predicted, but its expression in cells has not been proven[27,28]. We first conducted in silico analyses on human DNAJC12 based on its amino acid sequence (UniProt accession number Q9UKB3). To gain more information about potential functional regions we searched the Pfam database (pfam.xfam.org), which revealed a characteristic tetra-helical JD (PF00226) at residues 14-79 (Fig. 1a, b) that shows high structural similarity with the JDs of canonical JDPs DNAJA2 and DNAJB1 (Supplementary Fig. 1a–c), as also deduced from sequence alignments (Supplementary Figs. 2 and 3). No other structural or functional domains were predicted. The alignment of DNAJC12 sequences from distantly related species also confirmed that the C-terminal hepta-peptide (KFRNYEI) is fully conserved even as far as *D. melanogaster* (Supplementary Fig. 2).

The AlphaFold model of DNAJC12 (AF-Q9UKB3-F1) is based on the NMR structure of the N-terminal half (PDB 2CTQ), which includes a 13-residue N-terminal tail, the JD and an additional helix (linker-helix; residues 87-100), followed by an extended conformation up to the C-terminal residue I198 (Fig. 1b). Both the N-terminal tail prior to the JD and most of the C-terminal half are disordered regions according to the IUPred2A prediction[29], which also estimates that in addition to the JD, only the C-terminal end of the protein (residues 188-198) is structured (Fig. 1c and Supplementary Fig. 4). Furthermore, the hydropathicity index[30] revealed the generally hydrophilic character of DNAJC12, except for short regions (Fig. 1d and Supplementary Fig. 4).

In this work, recombinant full-length wild-type (WT) DNAJC12 and the selected variants (Supplementary Fig. 5) were expressed in *E. coli* as a fusion protein with N-terminal 6xHistidine and maltose binding protein (MBP) tags. The fusion protein was purified and the tags were subsequently removed by TEV-protease cleavage to achieve final purification of DNAJC12, with an average yield of ~3.5 mg/L culture for the WT protein. Analysis of purified DNAJC12 by SDS-PAGE revealed a single band (molecular mass ~27 kDa; theoretical 23.4 kDa) (Supplementary Fig. 6a). The slower-than-expected migration of purified DNAJC12 is consistent with previous western blot analyses in lysates of human prostate cells[28] and murine liver[17], which showed that endogenous DNAJC12 also presents a higher apparent molecular mass when partly purified in co-immunoprecipitated samples than in whole lysates. Using analytical size exclusion chromatography (SEC), purified DNAJC12 eluted as an apparent dimer (~49 kDa), but by coupling the column to multi-angle light scattering (SEC-MALS), a molecular mass of 22.7 ± 0.14 kDa was obtained (Fig. 1e), showing that the protein is monomeric. The low elution volume and larger apparent size of DNAJC12 on SDS-PAGE (Fig. 1e and Supplementary Fig. 6a) indicates the presence of disordered regions, as the hydration spheres of intrinsically disordered proteins are larger than expected, even in the presence of denaturants like SDS[31].

We then investigated the conformation and stability of purified DNAJC12 by synchrotron radiation circular dichroism (SRCD) spectroscopy (Fig. 1f). Limitations in available reference datasets for quantitative analysis of secondary structure in proteins with intrinsically disordered regions are known to result in their inaccurate assignment as β-sheets, notably in analyses of SRCD spectra[32]. This limitation does not seem to affect the assessment of α-helical content, which for DNAJC12 was estimated to be 30.5 ± 0.5%, corresponding well with the α-helix content in the JD (Fig. 1b, c and Supplementary Fig. 1). SRCD-monitored thermal denaturation measured at a wavelength of 222 nm provided a melting temperature ($T_m$) of 52.2 ± 6.0 °C for this structured region (Fig. 1f, inset).

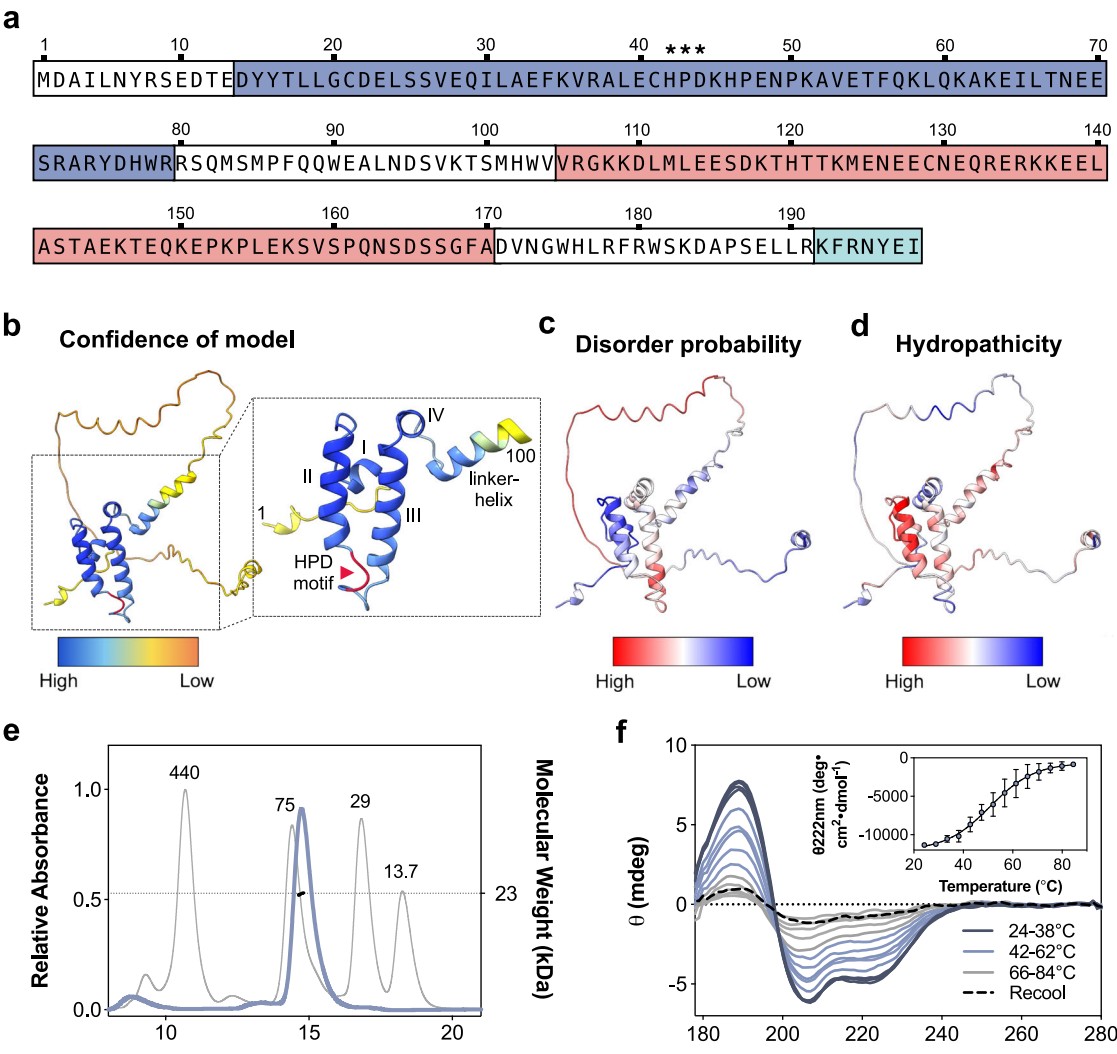

**Fig. 1 | Conformational characterization of purified DNAJC12 reveals regions with intrinsic disorder. a** Amino acid sequence of human DNAJC12, highlighting the J-domain (JD; dark blue), conserved HPD motif (asterisks), predicted disordered region (light red; see (**b** and **c**), and C-terminal conserved heptapeptide (light blue). **b** AlphaFold (AF) structural model of DNAJC12 (AF-Q9UKB3-F1), showing the JD with the highest prediction confidence. Excerpt: the first 100 residues, including the JD, are labeled with helices and the HPD motif in red. **c** Disorder prediction by IUPred2A mapped on the AF model. The JD and C-terminal end are predicted to be structured, while residues 104-171 (in the linker) are disordered. **d** Hydropathicity index by ProtScale (Kyte-Doolittle scale) mapped on the

AlphaFold model, indicating hydrophobicity and hydrophilicity. **e** SEC-MALS analysis of DNAJC12. Elution profile of calibration proteins ferritin, 440 kDa; conalbumin, 75 kDa; carbonic anhydrase, 29 kDa; and RNAse A, 13.7 kDa (gray line) provide an apparent molecular mass for DNAJC12 (blue line) of ~49 kDa. Coupling the column to a MALS detector provides the actual mass (22.7 ± 0.14 kDa), confirming that DNAJC12 is monomeric. **f** Far-UV SRCD spectra at increasing temperatures, showing the thermal unfolding of DNAJC12 (14 μM) (representative curves for n = 3 independent samples). Inset, mean ± SD of normalized CD signal at 222 nm; fitting to the Boltzmann equation provides the melting temperature ($T_m$) = 52.2 ± 6.0 °C. Source data are provided as a Source Data file.

## Two DNAJC12 monomers bind to TH through a client-binding C-terminal domain

We investigated the molecular basis for the interaction of DNAJC12 with TH using the purified proteins, and as an initial binding test we used SEC to analyze samples containing TH alone or with excess DNAJC12. With an excess of DNAJC12, the elution peak for the tetrameric TH decreased and eventually vanished at saturation (molar ratios of four monomeric DNAJC12 per tetrameric TH or higher). The elution peak of TH shifted to an earlier volume, indicating increased apparent size and the formation of a TH:DNAJC12 complex (Fig. 2a, left panel). However, the elution of a protein on SEC is not solely dependent on size, but also on shape, hence we investigated further by collecting fractions with the putative TH:DNAJC12 complex and analyzing them by SDS- and native PAGE. SDS-PAGE showed two bands for this peak, with proteins sized at about 56 and 27 kDa (Supplementary

Fig. 6b), corresponding to TH and DNAJC12 respectively, confirming the co-elution of both proteins. Moreover, in accordance with the SEC results, the isolated TH:DNAJC12 complex migrated slower than the DNAJC12 and TH control samples on native PAGE (Fig. 2b), and a band corresponding to unbound TH tetramer was not observed in the TH:DNAJC12 sample, indicating its complete recruitment to the complex. Moreover, we performed immunoblotting on the same samples with anti-DNAJC12 antibody showing that DNAJC12 is detected in the expected band for the TH:DNAJC12 complex (Fig. 2b), revealing a small proportion of dissociated complex that could only be detected by immunoblotting and not by the less sensitive Coomassie staining.

The stoichiometry of the TH:DNAJC12 complex was determined by SEC-MALS. Whereas TH alone showed the expected mass for the tetrameric enzyme (217.8 ± 1.4 kDa), the complex presented a mass of 266.1 ± 2.4 kDa, ~48 kDa larger than unbound tetrameric TH,

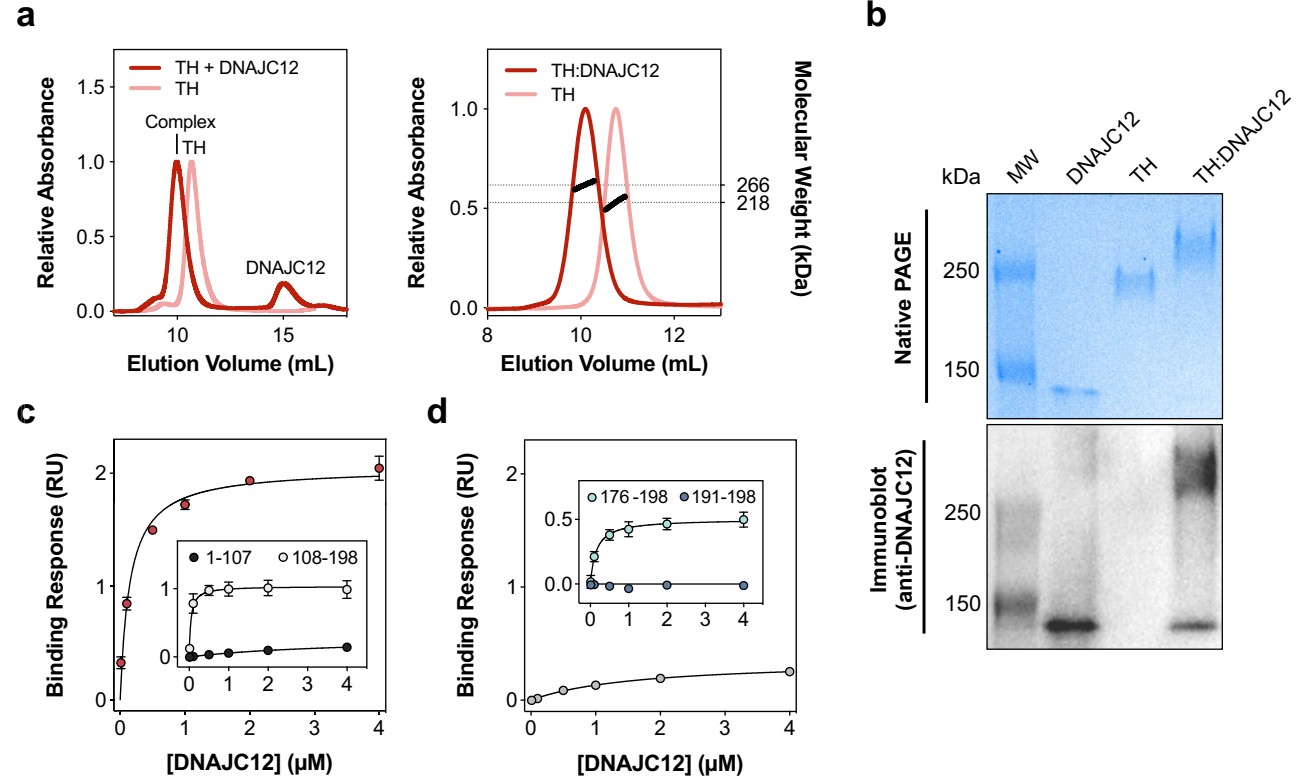

**Fig. 2 | Two DNAJC12 monomers bind via their C-terminal region to one TH tetramer. a** Left, SEC chromatograms show that excess DNAJC12 (20 µM) shifts the elution of TH (5 µM), indicating complex formation. Right, SEC-MALS analysis estimates the size of the TH:DNAJC12 complex as 266.1 ± 2.4 kDa, consistent with two DNAJC12 monomers binding to each TH tetramer. **b** Native PAGE (top) and immunoblot (bottom) analyses (2 µg protein in each sample) reveal that the TH:DNAJC12 complex migrates slower than TH alone through the native gel, and DNAJC12 is immunodetected in the complex. The results shown here are representative of at least three protein preparations tested in independent experiments. **c** BLI binding analyses of DNAJC12 and its truncated forms to TH, measured at varying concentrations (0.01-4 µM) of full-length DNAJC12 (red symbols),

DNAJC12(1-107) (black symbols; inset) or DNAJC12(108-198) (white symbols; inset). **d** Binding analyses of C-terminal-truncated DNAJC12(1-190) (gray symbols) and peptides DNAJC12(191-198) (dark blue symbols; inset) and DNAJC12(176-198) (light blue symbols; inset) to TH. Removal of the C-terminal octapeptide reduces binding affinity of DNAJC12(1-190) >10-fold compared with full-length DNAJC12 (red symbols in **c**). The 23-residue C-terminal peptide DNAJC12(176-198) binds with high affinity, while the 8-residue DNAJC12(191-198) does not bind. Binding responses in (**c** and **d**) are presented as mean ± 95% CI for n = 3 independent samples, and $K_D$ values estimated by non-linear regression (see Supplementary Fig. 7 and Table 1). Source data are provided as a Source Data file.

indicating that two DNAJC12 monomers bind to one TH tetramer (Fig. 2a, right panel). Further, the binding affinity, measured by biolayer interferometry (BLI), where biotinylated TH was immobilized on streptavidin sensors and its interaction with DNAJC12 measured at a range of concentrations, provided a dissociation constant ($K_D$) of 148 ± 18 nM DNAJC12 (Fig. 2c, Supplementary Fig. 7 and Table 1).

We then performed deletion mutagenesis on DNAJC12 to determine the regions that interact with TH. We designed and purified two truncated DNAJC12 variants containing either the N-terminal (residues 1-107) or C-terminal (residues 108-198) half (see Supplementary Fig. 5a for schematic representation of DNAJC12 variants prepared in this work). The typical yield for purification of recombinant DNAJC12(1-107), containing the JD, was much higher (~47 mg/L culture) than for the full-length protein (~3.5 mg/L culture), while the yield for DNAJC12(108-198) was much lower (~0.6 mg/L culture), suggesting that the C-terminal half imposes some instability and leads to increased intracellular proteolytic degradation during expression in *E. coli*. Using BLI, we found that the C-terminal DNAJC12(108-198) fragment binds to TH with high affinity ($K_D = 42 ± 6$ nM), even greater than full-length DNAJC12, while almost no binding response was recorded with the N-terminal DNAJC12(1-107) fragment (Fig. 2c, inset; Supplementary Fig. 7 and Table 1). Additional corroboration of binding of the C-terminal section to TH was obtained by SEC, followed by SDS-PAGE analysis of the co-eluted peak, while no binding was observed for

the N-terminal half (Supplementary Fig. 8a,b), further demonstrating that client binding is associated with the C-terminal half.

To better identify the client-binding determinants in DNAJC12, we focused on the highly conserved C-terminal octapeptide RKFRNYEI, predicted to have α-helical propensity (Fig. 1b). We generated the truncated form DNAJC12(1-190) (Supplementary Fig. 5a), and showed by BLI and SEC that removal of the C-terminal section nearly eliminated binding (Fig. 2d, and Supplementary Fig. 8c). The C-terminal eight amino acids of DNAJC12 thus seem crucial for interaction with TH, but since a peptide consisting of residues 191-198 was unable to bind to TH (Fig. 2d, inset) it appears that residues upstream of this conserved region may also be essential for the interaction. Indeed, since sequence alignments showed a high degree of sequence identity from residue 176 (Supplementary Fig. 2), we also synthesized the 23-residue peptide DNAJC12(176-198) (Supplementary Fig. 5a), which is incidentally the same region missing in the pathogenic DNAJC12 variant W175Ter[18] (see below, last section in Results). Remarkably, this DNAJC12(176-198) peptide exhibited a binding affinity to TH ($K_D = 152 ± 15$ nM) (Fig. 2d, inset) similar to full-length DNAJC12 ($K_D = 148 ± 18$ nM), underscoring the essential nature of the 23-residue C-terminal region for client binding. Given this finding, we have designated this region as a client-binding domain (CTD) (Supplementary Fig. 5a) in analogy to the terminology for other JDP-CTDs, despite the lack of sequence and structural similarity of this region in DNAJC12

**Table 1 | Summary of $K_D$ values for the interaction of TH and DNAJC12 obtained by bio-layer interferometry (BLI); effect of dopamine (DA) and variants**

| TH | DNAJC12 | $K_D$ (mean ± SEM) | Adjusted p-value | Significance |
|---|---|---|---|---|
| wild-type | wild-type | 148 ± 18 nM | - | - |
| wild-type + DA | wild-type | 147 ± 24 nM | 0.9460 | Ns |
| wild-type | (1-107) | 3987 ± 281 nM | <0.0001 | **** |
| wild-type | (108-198) | 42 ± 6 nM | 0.0001 | *** |
| wild-type | (1-174) | 7674 ± 1237 nM | <0.0001 | **** |
| wild-type | (1-190) | 1666 ± 127 nM | <0.0001 | **** |
| wild-type | (176-198) | 152 ± 15 nM | 0.9996 | Ns |
| wild-type | F193G | 1184 ± 0.378 nM | <0.0001 | **** |
| RD (1-158) | (176-198) | 156 ± 11 nM | 0.9990 | Ns |
| CD + OD (155-497) | (176-198) | 2501 ± 746 nM | <0.0001 | **** |
| L141A | wild-type | 39 ± 13 nM | 0.0019 | ** |
| L145A | wild-type | 33 ± 10 nM | 0.0012 | ** |
| R202H | wild-type | 162 ± 20 nM | 0.9751 | Ns |

The effect of dopamine (DA), truncations and variants on the binding affinity for the interaction of TH and DNAJC12 was studied by BLI. Values represent the mean ± SEM of $K_D$ calculated from the non-linear fitting of binding responses recorded from n = 3 independent experiments (see also Supplementary Fig. 7). The $K_D$ values derived using DNAJC12 or TH variants, or DA treatment, were compared to the calculated $K_D$ of the wild-type TH:DNAJC12 interaction. Significant differences between samples were determined by one-way ANOVA and Tukey's post-hoc HSD test. Ns non significant. Source data are provided as a Source Data file.

with other JDPs[2,8,13]. Furthermore, the central variable section (residues 87-175 in human DNAJC12), which contains the linker-helix (residues 87-100), appears as an interdomain linker (Supplementary Fig. 5a), as observed in other JDPs[8].

## DNAJC12 stabilizes TH and maintains its activity and inhibition by DA

Based on the relatively high affinity interaction between DNAJC12 and TH, we expected a stabilizing effect on TH conformation upon DNAJC12 binding[33]. Thermal unfolding of TH both alone and in complex with DNAJC12 was monitored using circular dichroism (CD) spectroscopy (Fig. 3a) and differential scanning fluorimetry (DSF; Fig. 3b) with SYPRO Orange dye. The melting temperature ($T_m$) values indicated a significant increase in thermal stability upon complex formation with DNAJC12 (Fig. 3a, b). DA is a physiological feedback inhibitor and stabilizer of TH[34], and adding equimolar amounts of DA (per TH subunit) increased the $T_m$ of TH as expected. Interestingly, we also observed a $T_m$ increase for the TH:DNAJC12 complex in the presence of DA (Fig. 3b), supporting that the catecholamine binds and stabilizes TH also in the complex. In addition, the affinity of DNAJC12 for TH with DA ($K_D = 147 ± 24$ nM) (Table 1) is comparable to that determined without DA, indicating that DNAJC12 also binds to the inhibited state of TH. These results point to the independence of the interactions between TH and either DA or DNAJC12, with no apparent mutual influence between the two TH ligands, and the binding of the two having an additive, non-excluding stabilizing effect on TH.

At least in vitro, TH is prone to time-dependent aggregation[35], which can be delayed by DA[34]. Since several JDPs have previously been reported to inhibit the aggregation of proteins associated with neurodegenerative diseases in an Hsp70-independent manner[6,8,10], we tested the effect of DNAJC12 on the propensity of TH to aggregate. As measured by dynamic light scattering (DLS), the aggregation of TH caused by incubation at 37 °C was largely reduced when DNAJC12 was added at saturating conditions (1:4 TH tetramer:DNAJC12 monomer ratio) (Fig. 3c). While DLS is an appropriate and effective method for

measuring aggregation kinetics, to determine the aggregate size we applied negative staining electron microscopy which seems advantageous in samples with mixed populations[36]. We also compared the effect of DNAJC12 in samples of TH without and with DA after 100 minutes of incubation at 37 °C, and as seen in Fig. 3d, the micrographs showed that DNAJC12 is more effective in preventing TH aggregation than DA. In the presence of DA we observed a reduction of the aggregates from 30 ± 3 nm (TH alone) to 21 ± 3 nm (TH(DA)) (Fig. 3d, left panels). In conditions where DNAJC12 is present, regardless of the presence of DA (Fig. 3d, right panels), a homogeneous particle size distribution (11 ± 2 nm) is observed, closely matching the diameter of individual TH molecules. This observation strongly suggests that DNAJC12 effectively prevents TH aggregation over time.

To determine whether DNAJC12 binding and subsequent conformational stabilization of TH influenced activity and inhibition by DA, we measured the $BH_4$-dependent conversion of L-Tyr to L-Dopa catalyzed by TH, both alone and in the presence of increasing concentrations of DNAJC12. The specific activity of the TH:DNAJC12 complex was similar to that of TH alone (1324 ± 218 nmol L-Dopa/min·mg TH) (Fig. 3e) and was only slightly reduced at the highest concentration of DNAJC12 (1 μM). We also measured the activity for TH and TH:DNAJC12 with varying concentrations of DA (Fig. 3f). The half-maximal inhibitory concentration ($IC_{50}$) for DA with TH alone (363 ± 82 nM) was similar to that with TH:DNAJC12 (356 ± 103 nM) (Fig. 3f, inset), indicating that even in complex with the cochaperone, TH maintains its enzymatic activity and remains sensitive to feedback inhibition by DA. These results reinforce the notion that the interactions of DNAJC12 and DA with TH involve distinct regions in TH.

## DNAJC12 interacts with TH by embracing its regulatory domains

We set out to determine by cryo-EM the location of DNAJC12 in the TH:DNAJC12 complex. Aliquots of the complex were placed on quantifoil EM grids and vitrified, and the best grid was used for data acquisition on a 300 kV Titan Krios at the Cell and Molecular Imaging Platform facility (SciLifeLab; Stockholm University), using the parameters described in Supplementary Table 1. The image processing and subsequent 3D reconstruction procedures are detailed in the Methods section and Supplementary Fig. 9. Upon visualization of the first 3D reconstructions and comparison with the volume of apo-TH (TH without bound DA)[34], the existence of an extra mass in the two regulatory domains (RDs) became evident, placed on opposite sides of the tetramer. Given that the established stoichiometry of the TH:DNAJC12 complex was two molecules of cochaperone per TH tetramer (Fig. 2a; right panel), we assumed that each DNAJC12 molecule could bind to each side of the TH tetrameric structure, and we introduced D1 symmetry in subsequent steps of the 3D reconstruction. The final volume (Fig. 4a; 4.25 Å resolution at FSC = 0.143) was very similar, albeit at a lower resolution, to that obtained for apo-TH (3.9 Å)[34]. The overall structure of the TH:DNAJC12 complex showed a central, butterfly-like structure formed by part of the four monomers, corresponding to the catalytic (CD) and oligomerization (OD) domains[34]. Two smaller masses were located on opposite sides of this central structure, each one corresponding to a dimer of RDs. A calculation of the local resolution of the TH:DNAJC12 complex using MonoRes[37] revealed that the central structure showed higher resolution (3–7 Å) than the two RD dimers (10–14 Å) (Supplementary Fig. 10a). Furthermore, comparison of the 3D reconstructions of apo-TH and the TH:DNAJC12 complex revealed the presence of two extra masses on opposite sides from these two RD dimers (shown in orange and red asterisks in Fig. 4a, and as orange and red masses in Supplementary Fig. 11a). The two masses have different sizes and none of them occlude the TH active sites (red circles in Fig. 4a), explaining why DNAJC12 binding does not affect TH activity. Docking of both domains on the atomic model of apo-TH fitted well in the 3D reconstruction of TH:DNAJC12 (Fig. 4b and Supplementary Movie 1).

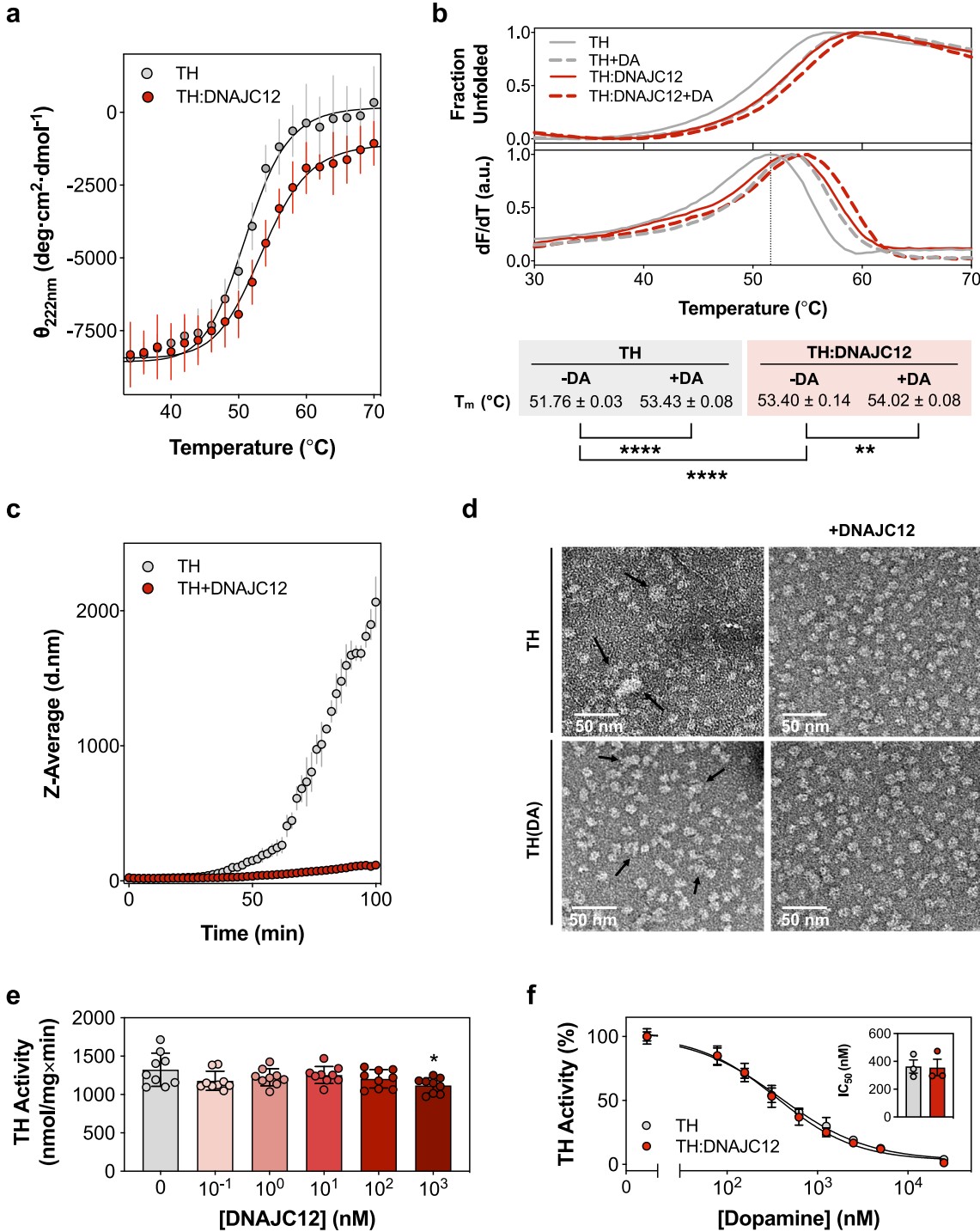

**Fig. 3 | DNAJC12 stabilizes and protects TH from aggregation without affecting its activity or DA feedback inhibition. a** CD-monitored (at 222 nm) thermal unfolding of TH (0.42 μM tetramer; gray symbols) and TH:DNAJC12 (0.42 μM complex; red symbols). Melting temperatures ($T_m$, mean ± SD): 51.21 ± 0.23 °C (TH) and 53.58 ± 0.47 °C (TH:DNAJC12), obtained from independent samples (n = 3) and triplicate measurements for each sample, and fitting to the Boltzmann equation. **b** DSF-monitored unfolding. Representative thermograms for TH (0.23 μM) and TH:DNAJC12 (0.23 μM) with/without 9 μM DA. $T_m$ values are presented as mean ± SD (n = 3 independent experiments) and analyzed by one-way ANOVA and Tukey's post-hoc HSD test (**p = 0.0034; ****p < 0.0001). **c** DLS-monitored aggregation. Z-average hydrodynamic diameter (d.nm; mean ± SD; n = 3 independent samples) of initially tetrameric TH (2.5 μM tetramer), for 100 min at 37 °C without (gray symbols) and with DNAJC12 (10 μM) (red symbols). **d** Visualization of aggregates by negative staining electron microscopy. Micrographs of samples initially containing tetrameric TH or TH(DA) (2.5 μM TH tetramer; 10 μM DA) after 100 min incubation

at 37 °C without (left panels) and with DNAJC12 (5 μM) (right panels). Black arrows point to TH aggregates. Aggregate diameter, measured with ImageJ, decreased from 30 ± 3 nm for TH alone to 21 ± 3 nm for TH(DA) (n = 20 aggregates for each sample). DNAJC12 completely prevents TH aggregation over time. Scale bar: 50 nm. **e** Effect of DNAJC12 on TH activity (0.4 nM tetramer, concentration in the assay). The bars represent the mean ± SD (n = 3 independent samples, each with technical triplicates). Statistical significance was assessed by one-way ANOVA and pair-wise multiple comparisons by Tukey's post-hoc test: *p = 0.0274. **f** DA concentration-dependent inhibition of activity of TH (0.4 nM tetramer; gray dots) and of TH:DNAJC12 (0.4 nM complex; red dots), presented as mean ± SD (n = 3 independent samples, each with technical triplicates). Fitting to a four-parameter logistic nonlinear regression provides similar half maximal inhibitory concentrations ($IC_{50}$) for TH (363 ± 82 nM) and TH:DNAJC12 (356 ± 103 nM). Source data are provided as a Source Data file.

Since only the 176-198 fragment of DNAJC12, assigned here as the CTD, seemed crucial for TH binding (Fig. 2d, inset), we surmised that this part of the sequence should correspond to one of these TH-interacting masses, and that the other mass could correspond to the JD, which is the only clearly structured part of DNAJC12 (Fig. 1b and Supplementary Fig. 1a). This domain amounts to half of the mass of the whole cochaperone, so we assumed that the JD should be assigned to the largest of the two masses described (orange asterisk in Fig. 4a and orange mass in Supplementary Fig. 11a). Indeed, the atomic structure of the human DNAJC12 JD and linker-helix (PDB 2CTQ; residues 1-100) (Supplementary Fig. 1a) fitted well in the above mentioned mass of the TH:DNAJC12 complex (no other docking is possible), and so did the AlphaFold predicted structure for the CTD (Fig. 4b). No mass in the volume of TH:DNAJC12 could be assigned to the unstructured linker region (residues 101-175).

It was surprising to find the DNAJC12 JD apparently interacting with TH since this domain does not bind to the enzyme on its own, as seen with the truncated form DNAJC12(1-107) (Fig. 2c, inset). It may be the case that a weak and transient interaction between the JD and some area of the RD dimer takes place after the tight binding between the cochaperone CTD and TH occurs. The 3D reconstruction of a sub-population of particles lacking the putative JD mass seems to strengthen this notion (populations without red circle in Supplementary Fig. 9e; ~30% of the total population), suggesting that the DNAJC12 N-terminus undergoes dynamic fluctuations, in agreement with its low affinity interaction with TH compared with the CTD. Still, since the resolution was not sufficient to visualize the structural features of the JD and to verify the presence of this domain in the TH:DNAJC12 complex, we carried out a 3D reconstruction of the complex between TH and DNAJC12(108-198), lacking the JD (Fig. 4c, d), which binds to TH with even higher affinity than full-length DNAJC12 (Table 1). The TH:DNAJC12(108-198) complex was purified, vitrified, and the best cryogrid used to acquire data as described above. The collected particles were used for further processing, and this time D1 symmetry was applied from the beginning. The final volume (Fig. 4c; 5.0 Å resolution; see also the docking of the atomic model of apo-TH into this volume; Fig. 4d) showed similar features to those of the TH:DNAJC12 complex (Fig. 4a, b), including a lower resolution for the two RDs (Supplementary Fig. 10b). The major difference between TH:DNAJC12 and TH:DNAJC12(108-198) was that one of the extra masses described above has disappeared (see Supplementary Fig. 11a,b, with an overlay of the two maps), confirming the identification of the JD. Docking of the AlphaFold model of the 23-residue CTD of DNAJC12 was good (Fig. 4d), and again, no mass could be assigned to the unstructured linker region.

## The feedback inhibitor DA and DNAJC12 bind to TH at different sites

We also carried out a 3D reconstruction of TH:DNAJC12 in the presence of the feedback inhibitor DA (TH(DA):DNAJC12). Biochemical results (Fig. 3b, f; Table 1) indicated that DA does not affect DNAJC12 binding to the enzyme and that DA-induced TH inhibition is unaffected in the TH:DNAJC12 complex as with apo-TH, which point to DA and DNAJC12 binding to TH as two apparently independent processes. Once the TH(DA):DNAJC12 complex was formed, aliquots were vitrified, and the best cryogrid was used for acquiring data. After several rounds of 2D classification, the best particles were selected and used for 3D reconstruction of the TH(DA):DNAJC12 complex. Again, D1 symmetry was used throughout the 3D reconstruction procedure, and the final volume (Fig. 4e; 5.0 Å resolution) confirmed the anisotropy in the resolution observed in the two previous 3D reconstructions, with the two RD dimers having much lower resolution than the central CD + OD domains (Supplementary Fig. 10c).

In general, the 3D reconstruction of TH(DA):DNAJC12 showed similar features to those of TH:DNAJC12, but with the added presence of four cylinder-like masses (the red asterisks in Fig. 4e point to some of them), each one corresponding to the 20-residue N-terminal regulatory α-helix (residues 39-58) in the RD, that occludes the TH active sites when DA is present[34] (see docking of the atomic model of TH with DA (PDB 6ZVP) into the 3D reconstruction of TH(DA):DNAJC12; Fig. 4f). The analysis of the volume revealed again the two extra densities of different sizes on opposite sides of the RD dimers (Supplementary Fig. 11c). These extra masses are close to but do not interact with any of the masses corresponding to the four regulatory α-helices described, which could explain in structural terms why DNAJC12 binding to TH does not affect DA binding to the enzyme. We then performed a docking of the JD and the CTD fragments already used for TH:DNAJC12 modeling into the described extra masses of the TH(DA):DNAJC12 complex (Fig. 4f). Docking of the two domains on opposite sides of the RD dimers was similar to that carried out with TH:DNAJC12 (Fig. 4a). In summary, the results with DNAJC12 and DA show that both bind in different places of the tetramer and their protective effect on the enzyme is additive.

## Validation of TH:DNAJC12 models by XL-MS and site-directed mutagenesis

Because of the low resolution of the masses assigned to the CTD (in the TH:DNAJC12 and TH:DNAJC12(108-198) 3D reconstructions) and to the JD (in the TH:DNAJC12 3D reconstruction) (Fig. 4), we resorted to a XL-MS analysis of the two complexes, using BS3 as crosslinker (see Materials and Methods). The results (Fig. 5a, b for TH:DNAJC12 and TH:DNAJC12(108-198) respectively; see also Supplementary Tables 2–4) reinforced the position of the JD and the CTD on opposite sides of the two RD dimers. In both experiments, part of the CTD, in particular residues 181-191, showed a large number of intermolecular crosslinks. It is worth noting that K183 exhibited exceptional reactivity in all analyses, emerging as the most frequent interprotein crosslink for the CTD of DNAJC12. In the case of the TH:DNAJC12 complex, five out of six crosslinks take place between TH and the CTD of DNAJC12, most of them connecting it with the RD, and all of them very close to where the CTD is placed in the 3D reconstruction of the complex (Fig. 5a and Supplementary Tables 2 and 3). One crosslink between K63 in the JD of DNAJC12 and K92 in the TH-RD was found, confirming the proximity of the JD to this TH domain. Additionally, the linker hosts five crosslinks with TH (Supplementary Table 2), but as this is a very flexible region that is not visible in the final volume, it was decided not to represent these crosslinks in the structural model because they clearly exceed the constraints allowed by BS3. Except for the expected disappearance of the crosslink with the JD, a similar crosslinking pattern was found when TH:DNAJC12(108-198) was analyzed (Fig. 5b and Supplementary Tables 2 and 3): four interprotein crosslinks were found and all of them take place between K183 of the CTD and TH, generating quite reasonable distances for BS3 (Fig. 5b). Moreover, 15 crosslinks with TH involve the flexible linker. All in all, the XL-MS results support the docking carried out for the CTD and the JD. In summary, the cryo-EM and XL-MS results point to DNAJC12 interacting with TH by embracing the RD dimers via binding of the CTD to one side and the interaction of the JD with the opposite side. The linker seems disordered and, therefore not visible in the 3D reconstructions.

The TH(DA):DNAJC12 model was also reinforced by an XL-MS analysis of the complex (Fig. 5c and Supplementary Tables 2–4). A total of nine crosslinks were found and, unlike what was observed for TH:DNAJC12, five interprotein crosslinks include a lysine from the JD, which helped us to validate the position proposed for this domain around the RDs. Note that these crosslinks are established between residues belonging to the RDs. This speaks of the quality of the XL-MS analysis, which also revealed four more crosslinks between the CTD and TH, three of them again between RD residues (Fig. 5c). Again, both the cryo-EM and XL-MS results with the TH(DA):DNAJC12 complex strengthen the notion of DNAJC12 embracing the RD dimers, without

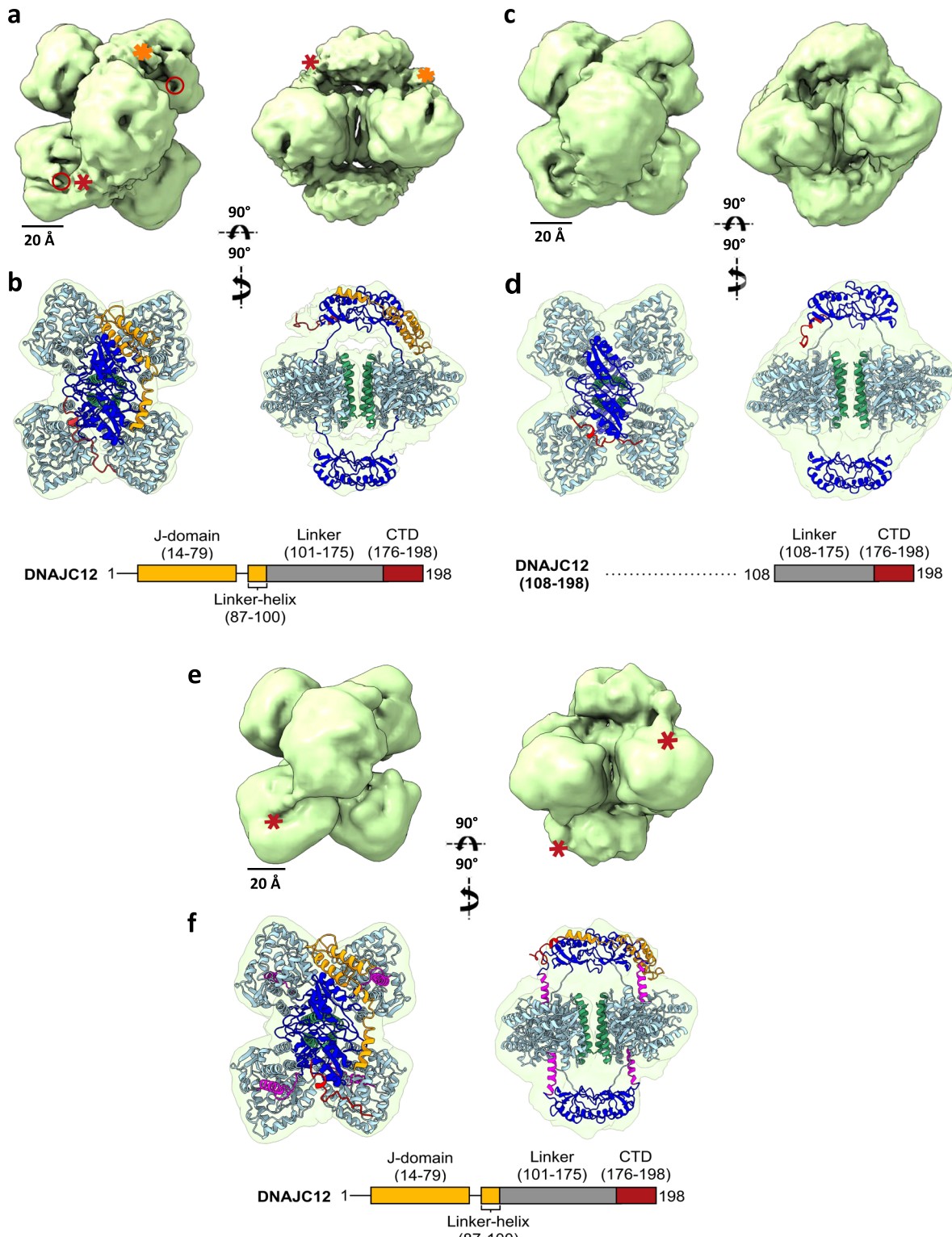

**Fig. 4 | 3D reconstruction by cryo-EM of the TH:DNAJC12, TH:DNAJC12(108-198) and TH(DA):DNAJC12 complexes. a**, **c**, and **e** display two views of the TH:DNAJC12, TH:DNAJC12(108-198) and TH(DA):DNAJC12 complexes, respectively. In (**a**), red and orange asterisks indicate extra masses compared to apo-TH (PDB 7A2G). Red circles mark two of the four TH active sites. In (**e**), red asterisks show the N-terminal regulatory α-helix (residues 39-58) of TH in TH(DA) (PDB 6ZVP) entering the TH active sites. **b**, **d**, and **f** show the same views with apo-TH (in **b**, **d**) and TH(DA) (in **f**), and the AlphaFold model of DNAJC12 docked into the cryo-EM volume. In all panels, the TH domains are colored in dark blue (RD), cyan (CD), and green (OD), and the regulatory domain α-helix (39-58) in TH(DA) is colored in magenta (**f**). The Alpha-Fold model of DNAJC12 has the JD in orange and the CTD in red. For simplicity, note that the models only show DNAJC12 bound to one of the dimeric TH-RDs. Schematic diagrams of the relevant DNAJC12 variants used are represented below the figures.

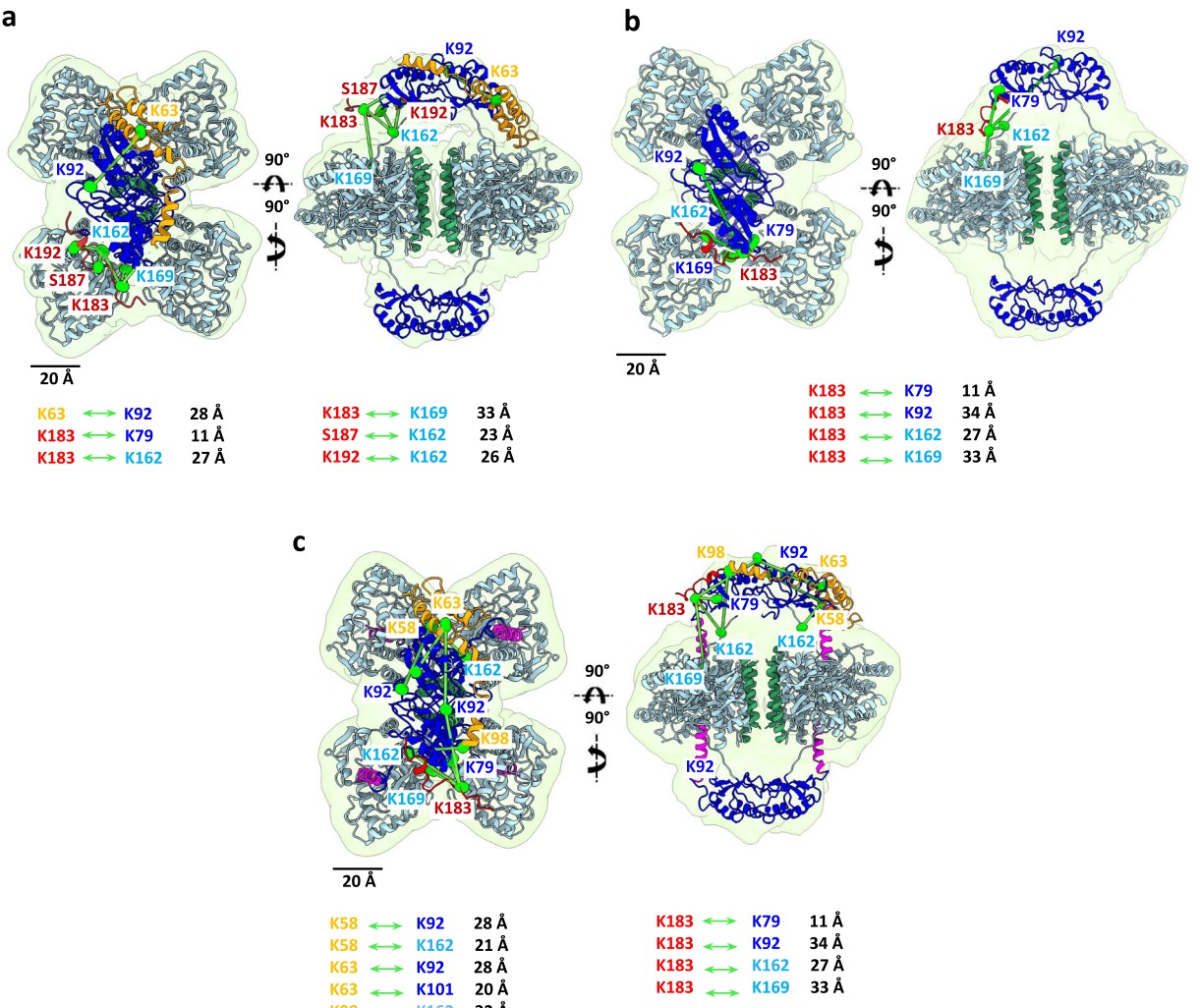

**Fig. 5 | Crosslinks obtained for the different TH:DNAJC12 complexes using XL-MS. a, b** and **c** show the same two views as in Fig. 4 for the TH:DNAJC12, TH:DNAJC12(108-198), and TH(DA):DNAJC12 complexes, respectively. The maps, structures and the color code used in all panels are identical to those in Fig. 4. The displayed crosslinks (highlighted in gray in Supplementary Table 2) are shown (in green) only for one side of the different TH complexes. Note that the distances of all crosslinks are also depicted.

affecting the ability of the regulatory α-helix to internalize into the TH active site in the presence of DA, in agreement with binding and activity data (Fig. 3b, f). Additionally, we performed an XL-MS-guided docking for all the complexes (TH:DNAJC12, TH:DNAJC12(108-198) and TH(DA):DNAJC12) using HADDOCK 2.4, a program that allows to define specific unambiguous distance restraints from XL-MS crosslinks and structural data from cryo-EM maps[38]. By incorporating these data as restraints, we generated new docking models that closely resemble our structural models (Supplementary Fig. 12). This HADDOCK-based validation strengthens our confidence in the positioning of DNAJC12 around the RD dimers and the proposed complex architecture.

To investigate if the interaction of DNAJC12 with TH only engages the TH-RDs independently of their interaction with the CDs, we prepared truncated variants including either the isolated RD (residues 1-158) or the CD + OD (residues 155-497) (Supplementary Fig. 5b). The RD was purified as a dimer and the CD + OD as tetramer (Supplementary Fig. 13), as expected from previous reports on the isolated RD and CD + OD[39,40]. As observed in Fig. 6a, the peptide DNAJC12(176-198), corresponding to the CTD, only bound to the dimeric RDs, and with a similar affinity as for full-length TH (Table 1). XL-MS (Supplementary Tables 2–4) and molecular docking of the DNAJC12-CTD to the dimeric TH-RDs (Supplementary Fig. 13c,d) also support the models of the complex of full-length DNAJC12 and TH (Figs. 4, 5 and 6b, c), showing

the specific interaction with TH-RDs. Moreover, the models present the JD-HPD motif accessible for interaction with Hsc70 (shown in Fig. 6b for the TH(DA):DNAJC12 complex). This is also supported by the models of the ternary complexes of TH:DNAJC12 interacting with either the ATP-bound open conformation of *E. coli* Hsp70 (DnaK) with bound JD (PDB 5NRO)[41] (Supplementary Fig. 14a,b), and with the ADP-bound closed structure of the Hsp70 SBD (PDB 1DKX)[42] (Supplementary Fig. 14c). Moreover, these models also place the TH-RDs very close to the SBD of Hsp70 (Hsp70$_{SBD}$). See also Supplementary Movie 2.

Finally, we performed site-directed mutagenesis at the conserved TH residues L141, L144, and L145, which are at the center of a hydrophobic cluster (L141-A142-A143-L144-L145) in Helix 2 in the ACT domain of TH-RD. This helix represents the closest interacting area with hydrophobic residues of the CTD of DNAJC12, L189, L190 and F193, notably the latter (Fig. 6c, excerpt I). The TH variants L141R, L144A, and L145R were unstable when expressed in *E. coli* and could not be purified, supporting the importance of these residues in maintaining the structure of the RDs through hydrophobic interactions with residues in Helix 1 of TH-RD (notably V102 and F106) (Fig. 6c, excerpt I). On the other hand, Ala mutations at L141 and L145 were less deleterious, and the variants could be purified, showing a small but significant increase in binding affinity for DNAJC12 (2-4-fold reduction in K$_D$ compared with TH-WT) (Table 1). Variants L141A and L145A thus

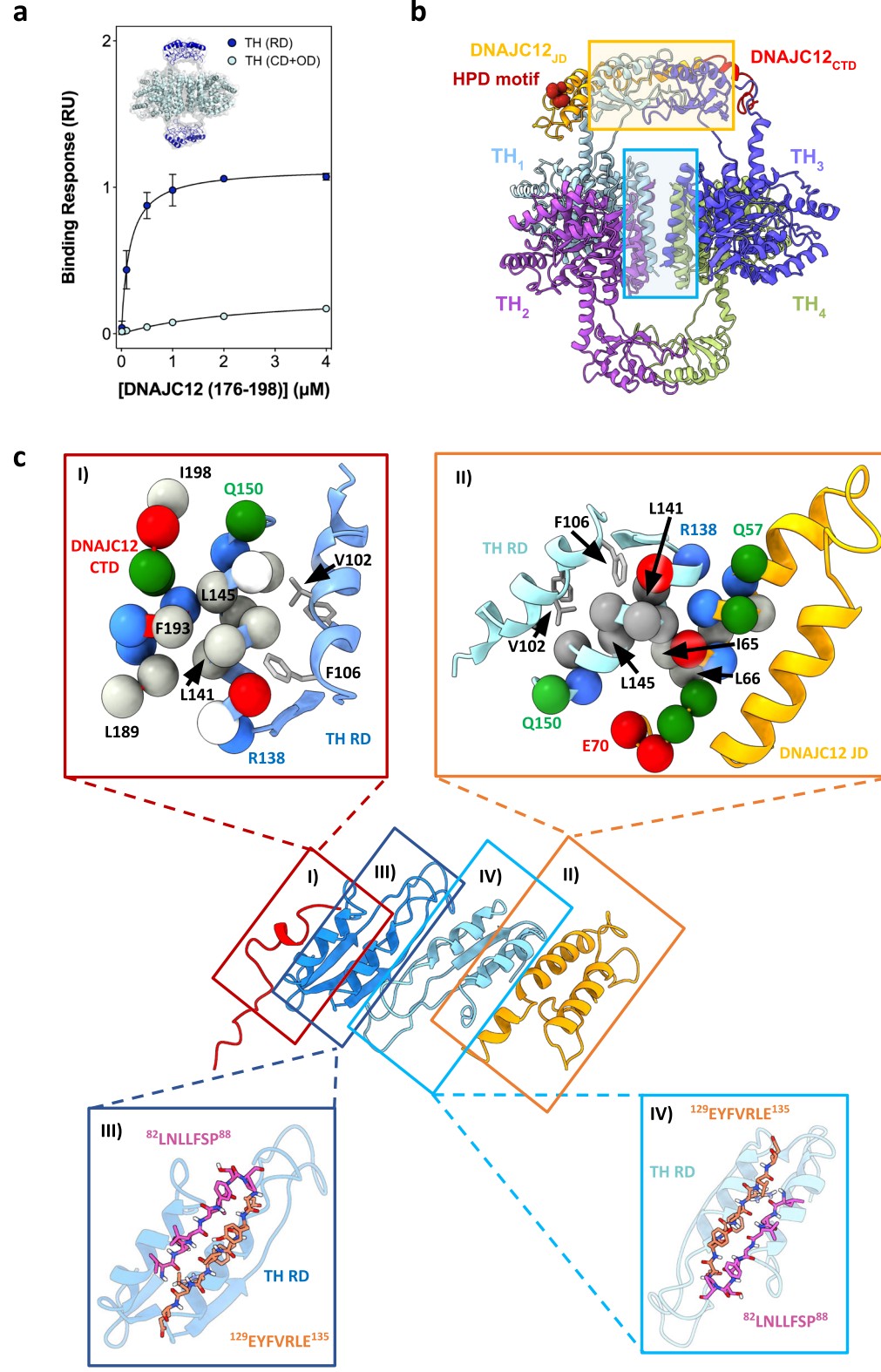

**Fig. 6 | Binding analyses and docking of the DNAJC12(176-198) peptide to the regulatory domains of TH. a** BLI binding analyses of DNAJC12(176-198) to purified dimeric TH-RDs (residues 1-158; dark blue) and tetrameric TH(CD+OD) (residues 155-497; light blue). Fitting to non-linear regression curves of the binding responses (mean ± 95% CI; n = 3 independent samples) provided similar affinity to TH-RDs and full-length TH (see $K_D$-values in Table 1). **b** Assembly of the TH:DNAJC12 complex. Tetrameric TH (PDB 6ZVP) assembles as a dimer of dimers (one dimer shown in light blue and purple and the other in green and dark blue), with the ODs facing the center (light blue rectangle). Complex formation with DNAJC12 (JD in orange, CTD in red) further stabilizes the tetramer by embracing the RD-dimer, composed by

monomers from opposite dimers (yellow rectangle). **c** Structural details of TH:DNAJC12 complex; I) DNAJC12-CTD (red ribbon and residues L189-I198 as balls) interact with Helix 2 of TH-RDs (cyan ribbon and residues R138-Q150 as balls); II) Helix III of DNAJC12-JD (yellow ribbon and residues Q57-E70 as balls) interact with Helix 2 of TH-RDs (light blue ribbon and residues V136-V151 as balls); III and IV) Regions [82]LNLLFSP[88] (pink) and [129]EYFVRLE[135] (orange), shown in both TH-RDs as ball and sticks in β-strands of the RDs. Both regions are predicted to bind to the Hsp70 family by Limbo[73] and ChaperISM[74], and residues 82-88 are also predicted to have high propensity for aggregating intermolecular cross-β interactions by TANGO[35,43]. Source data are provided as a Source Data file.

present some loss of the interactions with Helix 1 and/or higher packing towards F193 in the CTD of DNAJC12 (Fig. 6c, excerpt I). The involvement of this residue in the protein-protein interaction (PPI) was proven with the variant DNAJC12-F193G, which resulted in a reduced affinity for TH (8-fold increased $K_D$) (Table 1), supporting the importance of this very conserved C-terminal residue for interactions with the hydrophobic hotspot in Helix 2 of TH-RD. Regarding the interaction of the TH-RDs with DNAJC12-JD, the structural model of the complex shows that L141 and L145 in Helix 2 of TH only interact in this case with I65 in Helix III in the DNAJC12-JD (Fig. 6c, excerpt II). The weaker and lesser interactions of the DNAJC12-JD compared with the interface of the same region of TH with the DNAJC12-CTD (Fig. 6c, excerpt I) are consistent with the very low affinity binding of the isolated JD with TH (Fig. 2c, Supplementary Fig. 7 and Table 1). Together with the XL-MS analysis, the results from site-directed mutagenesis substantiate the cryo-EM structural model of the TH:DNAJC12 complex, which places the region 82-88, predicted by TANGO[35,43] to be highly predisposed to forming aggregating cross-β interactions, very close to DNAJC12 (Fig. 6c, excerpts III and IV).

## DNAJC12 stimulates Hsc70 ATPase activity when in complex with TH

As shown in Fig. 3c, d, DNAJC12 stabilizes and prevents aggregation of its client TH in an ATP-independent manner. On the other hand, the disaggregating/refolding activity, which would require the collaboration with the central chaperone Hsc70[44], can be assessed by monitoring the reactivation of luciferase aggregates. As seen in Fig. 7a, class B member DNAJB1 assists Hsc70 in refolding better than class A member DNAJA2, as previously reported[45], especially in the presence of Apg2, an Hsp110-type nucleotide exchange factor (NEF) that plays a crucial role in the function of the Hsp70 chaperone system and enhances the reactivation process. On the other hand, DNAJC12 was unable to recover luciferase activity regardless of the presence of Apg2 (Fig. 7a, inset). Also, to test the refolding of Hsc70-bound unfolded clients, we denatured the substrate protein in the presence of Hsc70, which prevents aggregation of the thermally unfolded client by binding aggregation-prone intermediates. Refolding is achieved by addition of the accessory JDPs and Apg2. While DNAJA2 and specially DNAJB1 assisted Hsc70 in luciferase refolding, DNAJC12 was again incapable of doing so both in the absence and presence of Apg2 (Fig. 7b). These data thus indicate that DNAJC12 does not collaborate with Hsc70 in the remodeling of aggregated or chaperone-bound luciferase. In addition, DNAJC12 did not affect either the onset or rate of Tau aggregation, unlike DNAJA2 that delayed the onset of Tau aggregation as expected from its efficient inhibition of this process[6] (Fig. 7c). Taken together, our results indicate that DNAJC12 may not recognize these unfolded proteins and may have a more specific role by dealing with specific clients.

We then compared Hsc70 activation by DNAJC12 with DNAJA2 or DNAJB1 in the presence of the NEF Apg2. DNAJC12 stimulated Hsc70 ATPase activity less efficiently than DNAJA2 and DNAJB1 (Fig. 7d). TH alone did not activate Hsc70, but with TH, DNAJC12 synergistically enhanced Hsc70 ATPase activity, unlike DNAJA2 and DNAJB1 (Fig. 7d), which do not interact with TH (Supplementary Fig. 7o,p). We also measured the ATPase activity of Hsc70-V438F, a variant defective in substrate binding[46,47]. This mutation completely abrogated the activation of Hsc70 ATPase activity by TH:DNAJC12 (Supplementary Fig. 15a), indicating the necessity of simultaneous binding of the JD of DNAJC12 and TH to Hsc70 to activate the chaperone. Our data suggest that TH is recognized as a client by Hsc70 in a DNAJC12-dependent manner.

For other JDPs, inefficient stimulation of Hsc70 ATPase activity without the client may be due to interdomain regulatory mechanisms, such as the autoinhibitory helix in the G/F-rich linker region in DNAJBs[48] or the CTD of DNAJB8[49], which shield the Hsp70 binding site.

The AlphaFold model of DNAJC12 did not show regions that could hinder interaction with Hsp70, but we analyzed variants lacking potential autoinhibitory regions, such as the N-terminal tail (variant DNAJC12(12-198)), the C-terminal section (variant DNAJC12(1-107)), and the linker-helix and the C-terminal section (variant DNAJC12(1-86)) (Supplementary Fig. 5a). None of these deletion mutants directly activated Hsc70 ATPase activity to the levels seen with DNAJA2, DNAJB1, or with the TH:DNAJC12 complex (Supplementary Fig. 15a, b).

## The pathogenic variant DNAJC12-W175Ter does not interact with TH

Recently, several pathogenic variants of DNAJC12 have been reported, associated with hyperphenylalaninemia and parkinsonism[16,18,21,50]. To further understand the pathogenic mechanisms, we characterized two of the most common disease-related variants identified to date, i.e., DNAJC12-R72P (NM_021800.3:c.215 G > C) and DNAJC12-W175Ter (NM_021800.3:c.524 G > A)[16,18]. We attempted to express and purify DNAJC12-R72P, but this variant was unstable and formed large insoluble aggregates, probably due to the disruption of the hydrogen bond between the highly conserved R72 (Supplementary Figs. 2,3) and S25[16], which appears essential to maintain the structural integrity of the JD. On the other hand, DNAJC12-W175Ter is a truncated disease variant lacking the last 24 residues due to a premature stop codon in the last exon of *DNAJC12*[18]. This variant, corresponding to DNAJC12(1-174) (Supplementary Fig. 5a), was successfully purified, with a yield of ~21 mg/L of culture. As the C-terminal of DNAJC12 is essential for client binding (Fig. 2d, inset), DNAJC12-W175Ter was expected to lose the ability to recognize and bind TH, which was indeed proven by BLI and SEC (Supplementary Fig. 16), explaining its pathogenic mechanism.

## The THD-associated variant TH-R202H is also stabilized by DNAJC12

We expressed, purified and characterized TH-R202H, one of the most frequent variants associated with TH deficiency (THD). The mutated residue is located in the CD, where it does not seem to be involved in interactions with DNAJC12 (Fig. 8a). TH-R202H has previously been shown to present reduced stability and increased susceptibility to proteolysis compared with TH-WT[51]. As measured by SEC-MALS, DNAJC12 bound to TH-R202H with a similar stoichiometry (4 TH subunits: 2 DNAJC12 subunits) (Fig. 8b) and affinity as TH-WT (Supplementary Fig. 7 and Table 1), and its interaction with DNAJC12 also resulted in thermal stabilization (Fig. 8c). Compared with TH-WT (Fig. 3c), the TH-R202H variant presents a higher aggregation propensity but is equally protected from aggregation by DNAJC12 (Fig. 8d), and provided a similar activation of Hsc70 ATPase activity than TH-WT when in complex with DNAJC12 (Fig. 8e).

## Discussion

Patients with congenital DNAJC12 deficiency present monoamine neurotransmitter decline and heterogenous clinical spectrum, including hyperphenylalaninemia and parkinsonism of varying severity, spanning from L-Dopa-responsive dystonia to severe neurodevelopmental delay and intellectual disability[16,21]. These traits led to the identification of DNAJC12 as a specific cochaperone for the AAAHs, including TH. The interaction of TH and DNAJC12 has been previously demonstrated through co-immunoprecipitation of both proteins in mammalian cells[16]. Additionally, it has been shown a reduction in TH expression in fibroblasts from patients with pathogenic DNAJC12 variants[16,18] and decreased DA levels in *Dnajc12* knockout mice[22]. However, the molecular details and functional effects of TH:DNAJC12 complex formation, as well as the pathogenic mechanisms behind DNAJC12 variants are currently not well-understood.

JDPs typically form dimers[2] as well as higher order oligomers in the case of DNAJA2, DNAJB6 and DNAJB8[8,49,52]. Unlike these better

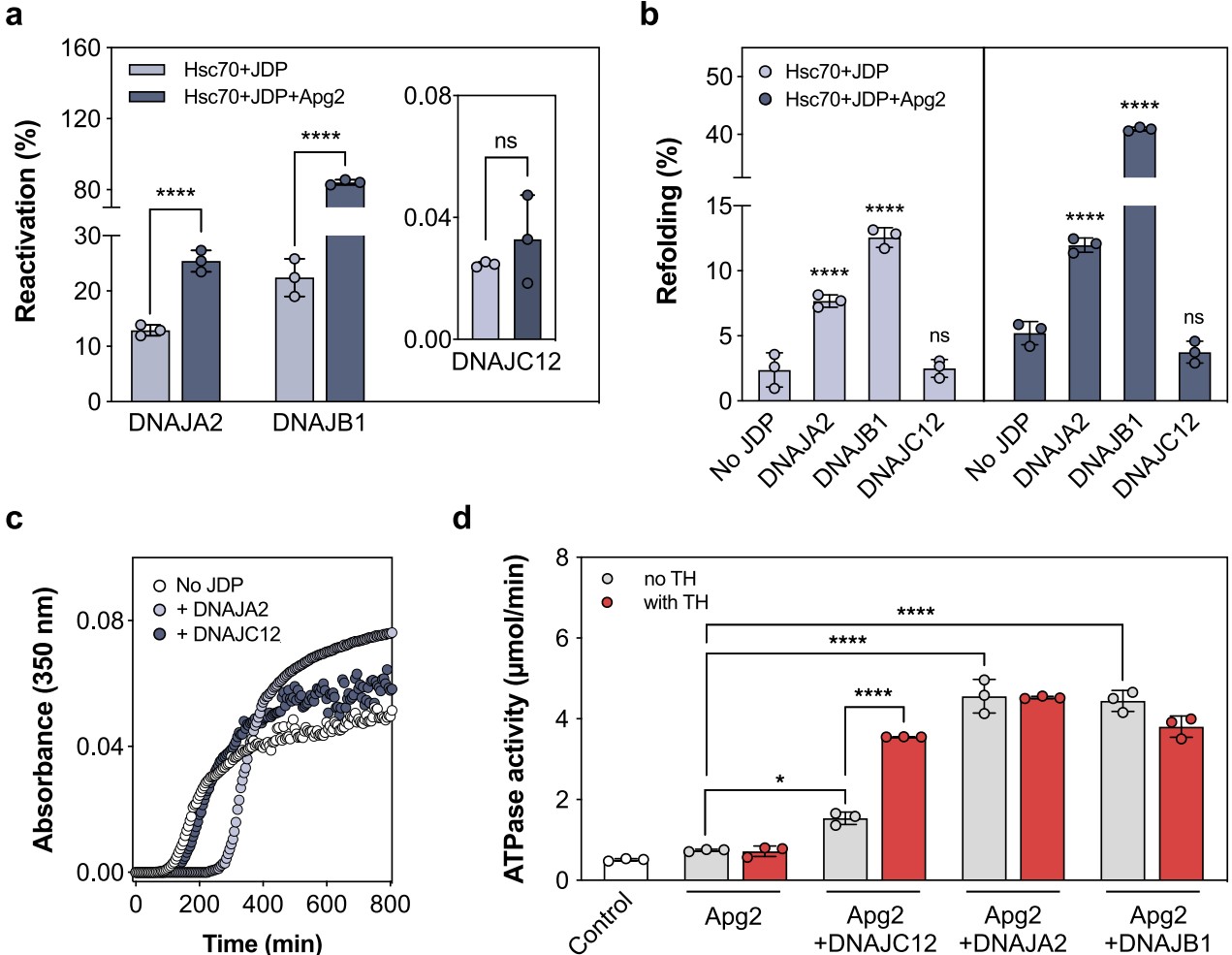

**Fig. 7 | DNAJC12 is unable to assist Hsc70 in remodeling aggregated and chaperone-bound unfolded luciferase, does not delay Tau aggregation, and only activates Hsc70 ATPase when complexed with TH. a** Reactivation of aggregated luciferase (20 nM) by Hsc70 and JDPs (DNAJB1, DNAJA2, DNAJC12) after 120 min without (light gray) and with (dark gray) nucleotide exchange factor (NEF) Apg2. DNAJB1 is most effective, and DNAJA2 also has good disaggregating/refolding activity[45], but DNAJC12 shows negligible activity ( < 0.05%; inset) and did not increase upon doubling the amount of DNAJC12 in the assay (data not shown). The reactivation efficiencies (mean ± SD; n = 3 independent experiments) were analyzed by one-way ANOVA and Tukey's post-hoc HSD test (****p < 0.0001). **b** Refolding of Hsc70-bound luciferase by JDPs alone (light gray) or with Apg2 (dark gray). Luciferase (0.1 μM) was denatured at 42 °C with 2 μM Hsc70, then 1 μM JDPs added without or with 0.4 μM Apg2. The percentage of refolded luciferase by the different JDPs, compared to their respective controls without or with Apg2, that only have Hsc70 and do not contain any JDP are presented as mean ± SD (n = 3 independent samples), and analyzed by one-way ANOVA and Tukey's post-hoc HSD test (****p < 0.0001). **c** Delay of heparin-induced Tau K18 P301L aggregation at 37 °C by JDPs, assessed by light-scattering. The data show representative time courses for Tau aggregation over time (n = 3, independent experiments) without (white) and with DNAJA2 (light gray) or DNAJC12 (dark gray). Only DNAJA2 successfully delays the aggregation, as seen by an extended lag phase in the aggregation onset. **d** Hsc70 ATPase activity stimulation by JDPs with and without TH. The ATPase activity of Hsc70 in the presence of DNAJC12, DNAJA2 and DNAJB1 in absence (gray) or presence (red) of TH were compared to the control with Hsc70 and Apg2 but without JDP. Data presented as mean ± SD (n = 3 independent experiments) by one-way ANOVA and Tukey's post-hoc HSD test (*p = 0.0288; ****p < 0.0001). Source data are provided as a Source Data file.

characterized JDPs, recombinantly purified DNAJC12 is monomeric with an extended conformation, but its oligomeric state in vivo is still unknown. Two DNAJC12 monomers bind to each tetrameric TH enzyme via a 23-residue client-binding domain (CTD), which interacts with similar affinity to TH as full-length DNAJC12. The loss of this region, in particular the evolutionarily-conserved C-terminal heptapeptide sequence KFRNYEI, predicted to have helical propensity, precludes binding and explains the pathological effect of the C-terminal protein-truncating variant DNAJC12-W175Ter, which has been identified in 40.4% of DNAJC12-deficient patients[53].

Although the resolution of the TH:DNAJC12 complex obtained via cryo-EM was insufficient to determine its atomic structure, it allowed modeling based on two additional masses interacting with each TH-RD dimer. These masses were not present in the cryo-EM model of apo-

TH[34] and were assigned to the N-terminal JDs and the TH-binding CTD based on the 3D reconstruction of the complex formed by TH and the DNAJC12 variant lacking the JD (TH:DNAJC12(108-198)). The N-terminal variant DNAJC12(1-107) that includes the JD shows very low binding affinity for TH alone, but binding through the CTD to one RD monomer may position the JD close enough to interact with the opposite RD monomer. This binding mode explains why tetrameric TH, which has four potential binding sites for the CTD, only interacts with two DNAJC12 molecules. Moreover, the flexibility of the disordered linker (residues 101-175) that connects the JD and CTD likely prevents it from being visible in the cryo-EM envelope of the TH complexes and indicates that it is unlikely to become structured upon binding to TH.

Other DNAJs, such as DNAJA2[6], DNAJB6[8] and DNAJC7[10], have been shown to prevent aggregation by blocking non-productive

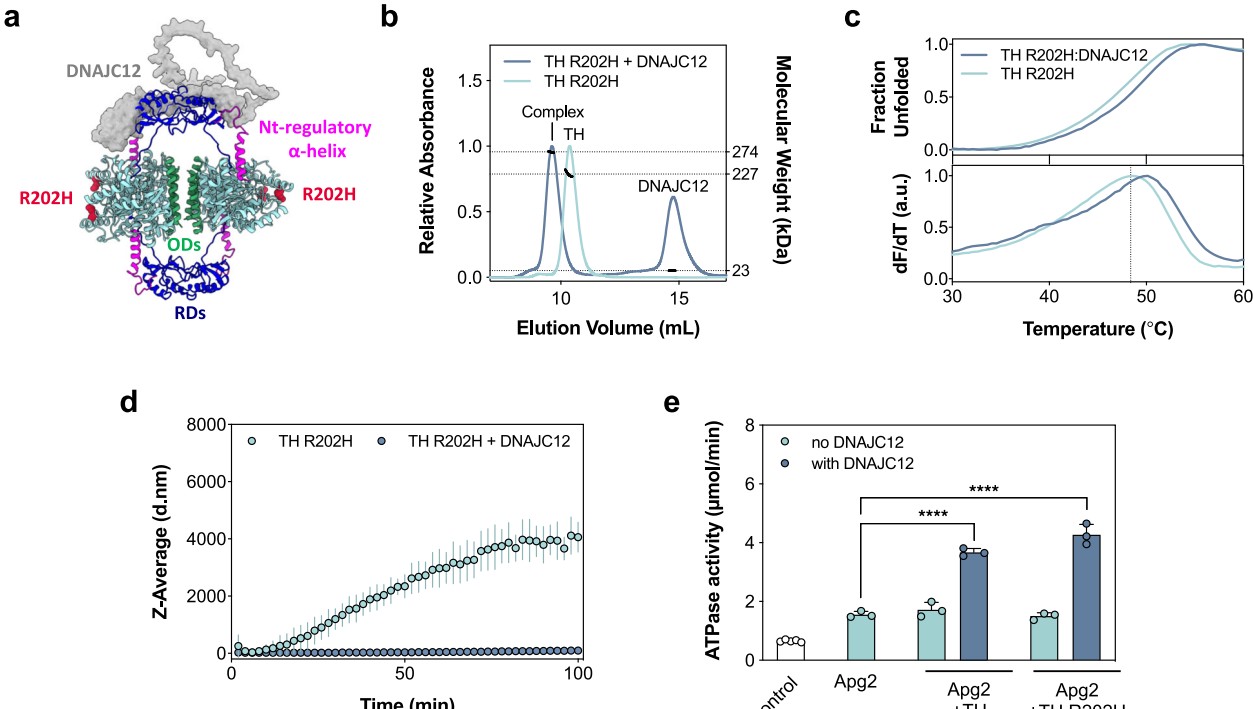

**Fig. 8 | DNAJC12 binds and stabilizes the disease-associated TH-R202H variant.**
**a** Model of the TH:DNAJC12 complex showing the position of the variant TH-R202H.
DNAJC12 is represented as a gray surface, while TH is shown as a cartoon. The
catalytic and oligomerization domains of TH are colored in cyan and green,
respectively, the two dimeric RDs in dark blue, the N-terminal regulatory α-helix in
magenta and R202H as red spheres. **b** Determination of TH-R202H:DNAJC12
binding stoichiometry by SEC-MALS. Excess DNAJC12 shifts TH-R202H elution,
indicating the formation of the TH-R202H:DNAJC12 complex (273.7 ± 0.95 kDa
size), consistent with two DNAJC12 monomers per tetrameric TH-R202H
(227.0 ± 4.5 kDa). **c** DSF-monitored unfolding. Upper panel, for TH-R202H (0.23 μM
tetramer; light blue) and TH-R202H:DNAJC12 (0.23 μM complex; dark blue). Lower

panel, the first derivative of the thermograms providing the melting temperatures
($T_m$; mean ± SD) for TH-R202H alone (48.38 ± 0 °C) and with DNAJC12
(49.81 ± 0.18 °C). **d** DLS-monitored aggregation. Z-average hydrodynamic diameter
(d.nm) of initially tetrameric TH-R202H (2.5 μM tetramer) monitored for 100 min at
37 °C, without (light blue symbols) and with DNAJC12 (10 μM; dark blue symbols),
represented as mean ± SD (n = 3 independent samples). **e** Stimulation of Hsc70
ATPase activity by DNAJC12 with either TH-WT or TH-R202H. Data represent the
mean ± SD for n = 3 independent experiments. The ATPase activity was compared
to the control containing only Hsc70 and Apg2 by one-way ANOVA and Tukey's
post-hoc HSD test (****p < 0.0001). Source data are provided as a Source Data file.

interactions. The cryo-EM structural model of the TH:DNAJC12 com-
plex, supported by XL-MS analysis and site-directed mutagenesis,
enhances our understanding of the efficiency of DNAJC12 in stabilizing
TH and delaying its aggregation. DNAJC12 resembles these JDPs in that
it binds to folded regions in the client using its JD and CTD (Fig. 6c), and
according to the model of the complex, DNAJC12 shields β-strand
residues $_{82}$LNLLFSP$_{88}$ in the TH-RDs, predicted to form cross-β amy-
loid-like structures[35]. The strategic positioning of DNAJC12 not only
blocks aggregation-prone regions but also stabilizes the tetrameric
form of TH. Tetrameric TH is composed of two dimers, each con-
tributing one RD to each of the two opposite RD dimers[34]. This
enhances tetramerization alongside the central helix bundle formed
by the ODs (Fig. 6b). By stabilizing the RDs, DNAJC12 prevents the
dissociation of the oligomeric assembly into monomers, which pre-
cedes misfolding and subsequent aggregation in other tetrameric
proteins, such as transthyretin[54,55]. The stabilizing activity of DNAJC12
over its client is ATP- and Hsc70-independent and is reminiscent of the
autonomous holdase function of other JDPs, which prevent aggrega-
tion of (partially) unfolded client proteins in cellular systems[6,10].
Moreover, the cryo-EM structural model of the TH:DNAJC12 complex
in the presence of DA shows a lack of interference with either the active
site or the regulatory helix (residues 39-58) involved in DA binding[34],
and explains why complex formation does not affect enzyme activity
or feedback regulation by DA, allowing for an additive stabilizing effect
by DNAJC12 and DA.

The affinity of the TH:DNAJC12 interaction ($K_D \sim 150$ nM) is rela-
tively high for JDP-client interactions, which usually range around

400 nM[6,10], but it is considered medium affinity for PPIs in general,
which display $K_D$ values typically between 1 nM and 1 μM[56]. This mod-
erate affinity helps stabilize TH while permitting interactions with
other regulatory proteins, such as 14-3-3 proteins, which bind to Ser19-
phosphorylated TH with a $K_D$ as low as 2.5 nM[57]. This balance, allowing
other regulatory proteins to outcompete DNAJC12, resembles the
behavior observed with other molecular chaperones[58].

In addition to supporting the PPI interfaces revealed by the cryo-EM
model, site-directed mutagenesis also provided interesting results on
the type of interactions, particularly on the main PPI interface, involving
Helix 2 in TH-RDs. Mutations at PPIs often result in decreased binding
affinity[33,59], but they may also cause tighter binding due to improvement
of packing and reduction of eventual steric hindrances, notably in
interactions that have evolved for a "fit for purpose" affinity rather than
for a stronger interaction[60]. The DNAJC12-F193A variant, which showed
an 8-fold increase in $K_D$, underscores the importance of F193 for binding
to the hydrophobic hotspot in Helix 2. In contrast, alanine mutations at
TH L141 and L145 resulted in a 2-4 fold improvement in binding affinity,
potentially due to the release of tight hydrophobic interactions with
Helix 1 in the TH-RD (notably with V102 and F106) (PDB 6ZVP[34]) that may
lead to higher packing towards F193 and increased binding affinity for
DNAJC12. With respect to the interaction of the DNAJC12 JD with the TH-
RD, the structural model of the complex shows that L141 and L145 in
Helix 2 of TH interact in this case with I65 and L66 in Helix III of the JD.
These interactions are weaker compared to those between the same
region of TH and the CTD of DNAJC12 (Fig. 6c), in correspondence with
the low affinity binding of the isolated JD with TH.

Parra et al. previously found that TH interacts physically with Hsc70 by binding to its SBD, regulating TH activity, localization and homeostasis of both TH and DA[61]. Moreover, DNAJC12 has been shown to interact with Hsc70 in LNCaP cells[28], but its role as a cochaperone has not been thoroughly explored. Our functional results along with Limbo[62]- and ChaperISM[63]-based prediction of the TH-RD residues [82]LNLLLFSP[88] and [129]EYFVRLE[135] as binding regions to the Hsc70 family supports that TH is an Hsc70 client. The high aggregation propensity of the first segment (residues 82-88)[35,43], as well as the proximity of both regions to Hsc70$_{SBD}$ in the models of the ternary TH:DNAJC12:Hsc70 complexes, further support that TH is a Hsc70 client and that DNAJC12 is essential for presenting TH to this chaperone (Supplementary Fig. 14 and Supplementary movie 2).

Canonical JDPs, such as DNAJA2 and DNAJB1, bind partially unfolded or misfolded proteins via hydrophobic regions, presenting these clients to Hsp70 for refolding or degradation[64,65]. This interaction explains the promiscuity of class A, B and some class C JDP members. In this work, folding, refolding and disaggregation assays have shown that DNAJC12 does not perform similar cochaperone functions as the canonical JDPs; instead, it resembles other DNAJCs that interact with the native state of their clients[10,13]. Examples include human DNAJC7, which binds its client Tau by recognizing specific folded elements[10], Jac1/DNAJC20 that interacts with Isu1[66], and auxilin/DNAJC6 that binds to clathrin triskelia[67]. In the case of DNAJC12, the specific interaction with folded areas of TH is supported by the defined binding saturation and stoichiometry (1:2 TH tetramer:DNAJC12 monomer ratio). In the case of TH-R202H[51], with a higher aggregation propensity, the non-aggregated species can still interact with DNAJC12 in the same way as TH-WT, allowing the presentation of the complex to Hsc70 and activation of ATPase activity. Additionally, the Hsc70-V438F variant[46], which has a strongly reduced affinity for substrates, totally eliminated the stimulation exerted by the TH:DNAJC12 complex, highlighting the role of DNAJC12 in specifically binding TH and potentially other hydroxylases as clients for Hsc70.

While DNAJC12 alone has minimal effect on Hsc70 ATPase activity, it significantly enhances this activity synergistically with TH. Class B JDPs activate Hsc/Hsp70 ATPase activity when these cochaperones form complexes with their clients, reversing autoinhibitory interactions, including a regulatory helix V that restricts JD accessibility[8,49]. Client binding or truncation of helix V removes this inhibition, except for DNAJB1, which can stimulate Hsp70 ATPase activity without client binding by interacting with an additional site on the Hsp70 C-terminal tail to release autoinhibition[48]. However, removal of linker-helix in DNAJC12 or other regions that could participate in regulatory intramolecular interactions did not reverse autoinhibition. Alternative explanations for the substrate-associated release of autoinhibition in DNAJC12 may involve adverse charge distribution or steric hindrances that become favorable in the TH:DNAJC12 complex. Recent findings indicate that the charge distribution on the DNAJC12 JD that interacts with the NBD-SBD interface in Hsc70 differs from other JDPs[44]. This suggests that TH binding may adjust the electrostatic potential or eliminate steric hindrances, leading to a synergistic interaction with Hsc70. Indeed, this a rather typical behavior for JDPs with specific substrates, such as seen for the synergy between auxillin and clathrin baskets[68], Jac1 (DNAJC20) and Isu1 (Fe-S cluster scaffold)[69], and DnaJ and σ32[41].

Our results demonstrate that DNAJC12 effectively stabilizes the THD-associated variant TH-R202H, providing significant protection against aggregation, which may be more relevant for THD variants than for TH-WT. Regarding the rest of the THD-associated variants (716 deposited in the publicly available PNDdb (http://www.biopku.org; accessed 17 June 2024), 220 (30%) are missense variants, which are spread throughout the entire TH sequence. Despite the lower frequency of variants in the RD, it is interesting that in the main area of interaction with DNAJC12, corresponding to Helix 2 in the RD (residues 138-150), there is a high recurrence of missense variants (A142S, A143T,

L145H, V148A, R149C, and R149L). The last four variants present a high pathogenicity index (Combined Annotation-Dependent Depletion (CADD) score > 20) (www.biopku.org/PNDdb) and may be expected to affect the stability of TH and/or its interaction with DNAJC12. This aligns well with the finding that disease-related mutations are generally enriched in PPI regions[59]. In any case, it is important to note that, in general, all THD-associated variants, but specifically those in the RD, may result in conformational alterations that could affect the area around Helix 2 and thus binding of TH with DNAJC12. With respect to other DNAJC12 associated variants, our work also provides insights on the molecular pathogenic mechanisms for the variants R72P[16] and W175Ter[18], as the former leads to aggregation, whereas W175Ter affects the PPI with TH, totally abolishing the interaction.

In conclusion, this study identifies the structural determinants for TH:DNAJC12 complex formation, an interaction that stabilizes TH and protects it from aggregation without impairing TH catalytic activity or affecting its regulation. Our findings suggest that the DNAJC12 synergistically activate Hsc70 ATPase activity with TH, in accordance with a specific client interaction. Selectivity and relatively high affinity for the native form of the client, in addition to a Hsp70-independent protection of TH from aggregation (holdase activity), are increasingly recognized properties of the class C JDPs. Moreover, our results contribute to the understanding of the pathogenic mechanism of variants associated with DNAJC12 deficiency and THD.

## Methods

### In silico characterization of DNAJC12

The UniProt entry Q9UKB3 was used as query sequence for full-length human DNAJC12. Homologous proteins were identified using BLAST[70] and subsequently visualized in ESPript[71]. Disorder probability was determined using IUPred2A[29], and ab initio structural models were generated by AlphaFold[26].

### Peptides, plasmids and cloning

The DNAJC12-derived peptides with residues 176-198 and 191-198 were prepared by GenScript. cDNAs of Apg2 (HSPH2), Hsc70 (HSPA8), DNAJA2, DNAJB1 and DNAJC12 (Addgene) were cloned into the pE-SUMO vector (LifeSensors). The construct encoding Tau K18 P301L was obtained from Prof. David Klenerman. Hsc70-V438F was obtained as described[46]. The pETMBP1a/hTH1 plasmid[72] encoding TH-WT was previously available in the laboratory, while the truncated and site-directed TH mutants and the THD-variant TH-R202H (Supplementary Fig. 5) were derived from the same plasmid (GenScript).

Due to aggregation of DNAJC12 in the pilot expression and purification experiments, the cDNA was amplified by PCR using the forward primer 5′-GCTTCCATGGATGCAATTCTGAATT-3′ and the reverse primer 5′-GCTTGGTACCTTAAATTTCATAATTACGA-3′ to enable restriction enzyme cloning (the NcoI and Acc65I restriction sites are underlined) into the pETMBP1a plasmid. The pETMBP1a/DNAJC12 plasmid was prepared by inserting the cDNA after the recognition site for the tobacco etch virus (TEV) protease to fuse the DNAJC12 protein to a 6xHistidine-tagged maltose binding protein (HisMBP) tags that could later be cleaved off during purification. The constructs used to express and purify DNAJC12(1-107), DNAJC12(108-198), DNAJC12(12-198), DNAJC12(1-86), DNAJC12(1-190), DNAJC12-R72P and DNAJC12(1-174/W175Ter), and the site-directed mutant DNAJC12-F193G, were derived from the same 6xHistidine-tagged pETMBP1a/DNAJC12 plasmid by GenScript. Correct reading frame and absence of additional mutations in all the final constructs were verified by sequencing.

### Expression and purification of proteins

Recombinant proteins containing N-terminal His and SUMO tags (Apg2, Hsc70, DNAJA2, DNAJB1) were expressed in E. coli BL21 CodonPlus(DE3)-RIL or Rosetta™(DE3) cells and purified by immobilized metal ion affinity chromatography (IMAC) using Ni-NTA agarose

(Qiagen) columns, followed by treatment with His-Ulp1 to cleave the tag, and a final purification on a Ni-NTA agarose column from which the pure protein was eluted in the unbound fraction[46,52].

WT and mutant DNAJC12 fusion proteins were expressed at 20 °C for 24 h in *E. coli* BL21-CodonPlus(DE3)-RIL grown in Terrific Broth supplemented with 50 µg/mL kanamycin and 34 µg/mL chloramphenicol following induction with 0.5 mM isopropyl β-D-thiogalactoside at $OD_{600nm}$ of 1.2–1.6. A crude extract was obtained by sonication in 20 mM Na-Hepes pH 7.0, 400 mM NaCl with protease inhibitors (1 cOmplete™ ULTRA Tablet, Mini, EASYpack Protease Inhibitor Cocktail (Roche) per 10 mL and 10 mM benzamidine-HCl) and subsequent centrifugation. The HisMBP-DNAJC12 fusion protein was purified from the clarified whole-cell lysate using the HisTrap™ High Performance 5 mL column (Cytiva) and eluted with buffer supplemented with 250 mM imidazole. Removal of imidazole and excess salt was attained by overnight dialysis in 20 mM Na-Hepes pH 7.0, 200 mM NaCl. This buffer was used in all succeeding purification steps.

The purified fusion proteins were cleaved using His-TEV protease (1:10 w/w at 35 °C for 1.5 h). IMAC using TALON™ metal affinity resin (Clontech) and subsequent amylose affinity chromatography on a gravity column were used to remove residual fusion protein, HisMBP species and His-TEV protease. Aggregates were removed by SEC on a HiLoad™ Superdex™ 75 prep-grade column (1.6 cm × 60 cm). Protein concentrations were determined by Nanodrop UV–Vis spectrophotometer using their respective theoretical extinction coefficients. TH-WT, TH-R202H variant and truncated and site-directed TH mutants were tested for expression in *E. coli* BL21-CodonPlus(DE3)-RIL and purification as described by Bezem et al.[72]. Briefly, the HisMBP-TH fusion proteins were purified by amylose affinity chromatography and cleaved using His-TEV protease (1:50 w/w at 4 °C for 3 h). Tetrameric TH was isolated from aggregates and cleavage products by SEC HiLoad™ Superdex™ 200 prep-grade (1.6 cm × 60 cm) column (Cytiva). TH(1-158) (RD) could only be purified by IMAC using TALON™ metal affinity resin (Clontech) pre-equilibrated in 20 mM Na-Hepes pH 7.0, 200 mM NaCl. On-column digestion of the fusion protein was performed after extensive washing with 1 mg His-TEV protease per liter of culture at 4 °C for 3 h before collection of the flowthrough. Aggregates and other impurities were removed by SEC on a HiLoad™ Superdex™ 200 prep-grade (1.6 cm × 60 cm) column (Cytiva). Protein concentration was determined by measuring amide bonds using Direct Detect® IR spectrometer (Merck Millipore) due to a lack of aromatic amino acid residues in its sequence. Tau was expressed in *E. coli* BL21-CodonPlus(DE3)-RIL (Stratagene) and purified following the protocol described by Barghorn et al.[73], with modifications. In brief, cells were lysed by sonication and, after clarification by ultracentrifugation, the supernatant was heated at 95 °C during 20 min in a buffer containing 500 mM NaCl. The soluble fraction was dialyzed to reduce NaCl concentration to 50 mM and loaded into a HiTrap SP HP column (Cytiva). Proteins were eluted with a linear gradient to 600 mM NaCl. Tau containing fractions were then loaded into a HiLoad™ Superdex™ 75 prep-grade (1.6 cm × 60 cm) column (Cytiva). Tau fractions were concentrated and stored at -80 °C.

### Analytical size-exclusion chromatography
Purified DNAJC12 (130 µM) was analyzed on a Superdex™ 200 Increase (1.0 cm × 30 cm) column (Cytiva) at a flow rate of 0.5 mL/min at 4 °C. Standard proteins (GE Healthcare) were used for column calibration (ferritin; 440 kDa, conalbumin; 75 kDa, carbonic anhydrase; 29 kDa, RNAse A; 13.7 kDa). For binding studies, samples containing 5 µM (tetramer) TH alone or with 20 µM (monomer) DNAJC12 were prepared and analyzed at the same conditions.

### Purification of TH:DNAJC12, TH:DNAJC12(108-198) and TH(DA):DNAJC12 complexes
Both TH:DNAJC12 and TH:DNAJC12(108-198) complexes were purified using SEC on a Superdex™ 200 Increase (1.0 cm × 30 cm) column by

combining TH and DNAJC12 proteins at a 1:8 TH tetramer:DNAJC12 monomer molar ratio and collecting the fractions corresponding to the shifted peak (as observed in Fig. 2a and Supplementary Fig. 8a). Once the TH:DNAJC12 complex was purified and its functionality confirmed by determining its TH enzymatic activity and sensitivity to feedback inhibition by DA (see section 'Assay of TH activity and inhibition by DA', below and Fig. 3f), the TH:DNAJC12 complex was incubated with iron and DA to obtain the stable Fe(III)-catecholate complex[34]. Ferrous ammonium sulfate (FAS) was dissolved in degassed water, added to TH (1x molar ratio per TH subunit) and incubated for 2 min at room temperature (RT) before the addition of DA (2x molar ratio per TH subunit).

### Native PAGE
Samples containing 0.2 µg/µL DNAJC12, TH or purified TH:DNAJC12 complex in 20 mM Na-Hepes pH 7.0, 200 mM NaCl were diluted 1:1 with Native PAGE Sample Buffer (Bio-Rad) and loaded into a 10% Mini-PROTEAN® TGX Precast Protein Gel (Bio-Rad). The gel was run at 140 V for 3 h at 4 °C with running buffer (25 mM Tris pH 8.3, 192 mM glycine), prior to visualization using 0.08 mM Coomassie brilliant blue G-250 (Bio-Rad) with 5 mM HCl.

### Immunoblotting
Proteins were transferred from the PAGE gel onto a polyvinylidene difluoride membrane (Bio-Rad) using the Transblot® Turbo™ Transfer System (Bio-Rad) at 25 V for 3 min. The membrane was blocked with a blocking solution (5% w/v skim milk powder, 1x Tris-buffered saline (TBS), 1% Tween 20). Primary antibody incubation was carried out overnight using a rabbit anti-DNAJC12 antibody (1:10000; ABCAM, cat. no. AB167425). The membrane was subsequently washed to reduce unspecific binding using washing buffer (1x TBS, 0.1% Tween 20) before incubation with goat anti-rabbit antibody (1:1000; Bio-Rad, cat. no. 1706515). The membrane was washed again prior to visualization using enhanced Luminita Immobilon® Crescendo Western horseradish peroxidase substrate (Merck Millipore) and the ChemiDoc™ XRS+ System (Bio-Rad).

### Circular dichroism spectroscopy
For customary CD spectroscopy, samples were prepared containing 0.42 µM TH (tetramer) or TH:DNAJC12 complex in 20 mM Na-Phosphate pH 7.0, 80 mM NaF buffer, and placed in a 1 mm path length quartz cuvette. Triplicate measurements from three independent samples were collected with a J-810 CD spectropolarimeter (JASCO) equipped with a temperature-controlled cell holder. Protein thermal denaturation was monitored at 222 nm from 21–74 °C at a ramp rate of 1 °C per min, and each measurement was corrected by baseline subtraction using the blank value at 222 nm obtained with the buffer. The CD signal was normalized to residue molar ellipticity ($[\theta]$, in deg·cm²·dmol⁻¹).

Samples for SRCD spectroscopy were prepared to contain 14 µM DNAJC12 in 20 mM Na-Phosphate pH 7.0, 80 mM NaF. SRCD spectra were collected from 280–170 nm (3 scans averaged) at 25 °C on the AU-UV beamline at ASTRID2, part of the Institute for Storage Rings (ISA) facility at the Aarhus University, Denmark. All spectra for baselines and samples were measured in the same cuvette, a 121-type closed cylindrical cell. The path length of the cell was confirmed using an interferometry method and found to be 100 µm. The protein concentration was measured using $A_{280nm}$ prior to the measurements and more accurately by $A_{205nm}$ from the absorption spectra measured simultaneously with the CD spectra, using the theoretical extinction coefficient. Secondary structure content was determined in the 250–180 nm range using the BeStSel algorithm[32].

### Size exclusion chromatography coupled with multi-angle light scattering
The sizes of DNAJC12 (3 mg/mL), TH (2 mg/mL), TH:DNAJC12 complex (2 mg/mL), TH(1-158) (2 mg/mL) and TH(155-497) (2 mg/mL) in 20 mM

Na-Hepes pH 7.0, 200 mM NaCl were determined by SEC-MALS. Samples were filtered using Corning® Costar® Spin-X® (0.22 µm) plastic centrifuge tube filters (Merck) prior to sample application. The proteins were analyzed using a Superdex™ 200 Increase (1.0 cm × 30 cm) column (Cytiva) connected to an iSeries LC-2050 HPLC system (Shimadzu) coupled to a RefractoMax 520 module (ERC GmbH) to measure the refractive index and determine concentration, and a mini-DAWN TREOS detector (Wyatt Technology) to measure light scattering. Data processing and molar mass estimation were performed using Astra software (Wyatt).

## Bio-layer interferometry
BLI was performed on an OctetRED96 platform (FortéBio) using strep-tavidin (SA) biosensors (Sartorius). For immobilization on the biosensors, WT and truncated TH variants were biotinylated using EZ-Link™ NHS-PEG4-Biotin (Sigma Aldrich) following the manufacturer's protocol and diluted to 50 µg/mL in the assay buffer (1x phosphate-buffered saline (PBS), 0.02% Tween-20, 0.5 mg/mL BSA). WT and DNAJC12 truncated mutants and variants were also diluted in the assay buffer to a range of concentrations (0.01, 0.1, 0.5, 1, 2 and 4 µM), before the samples for immobilization and association were dispensed on a 96-well flat-bottom polypropylene plate (Greiner) that was maintained at 25 °C and agitated at 1000 rpm during the assay. Prior to each experiment, the sensors were soaked in 1x PBS for 1–3 h before equilibration in the assay buffer for 30 s, subsequent TH loading for 300 s and re-equilibration in the assay buffer for 120 s. The binding of DNAJC12 to immobilized TH was monitored for 600 s before a final round of equilibration in the assay buffer for 300 s. The data were processed with the manufacturer's software (Octet Data Analysis HT). Signals from the zero-concentration sample were subtracted from the signals obtained for each functionalized biosensor, plotted against the respective analyte concentrations and fitted to a one-site binding (hyperbola) non-linear regression value to derive the $K_D$ values.

## Dynamic light scattering
TH aggregation was monitored in vitro by dynamic light scattering (DLS) using the Zetasizer Nano ZS instrument (Malvern Panalytical). Samples were prepared to contain 2.5 µM TH (tetramer) alone or with 10 µM DNAJC12 in 20 mM Na-Hepes pH 7.0, 200 mM NaCl, 20 µM FAS. The samples were centrifuged at 16,000 × g for 5 min and filtered using Corning® Costar® Spin-X® (0.22 µm) plastic centrifuge tube filters (Merck) prior to sample loading into UV cuvettes (Merck). Protein aggregation was analyzed by monitoring the change in average particle size or Z-average value (diameters in nm; d.nm) of the protein samples over an 100 min period at 37 °C. Light scattering was measured every 2 min using a He-Ne laser at 683 nm with a fixed scattering angle of 173°.

## Differential scanning fluorimetry
The thermal stability of purified TH and TH:DNAJC12 complex with and without DA were measured by DSF. Samples with 0.23 µM (tetramer) TH alone or in complex with DNAJC12 were incubated in 2x FAS, 5x SYPRO™ Orange dye (Merck) in the presence or absence of equimolar amounts of DA and loaded into a 384-well plate (Corning). The plate was heated stepwise from 25 °C to 90 °C at a heating rate of 2 °C/min using a LightCycler 480 Real-Time PCR System (Roche Applied Science). The thermal denaturation of TH was monitored by detecting the increase in SYPRO™ Orange fluorescence (λex = 465 nm, λem = 610 nm) upon its binding to exposed hydrophobic patches. The data were analyzed using HTSDSF Explorer[74] and scaled to reflect the fraction unfolded and midpoint denaturation ($T_m$). $T_m$ values were determined as the maximal temperature in the first-derivative plot.

## Assay of TH activity and inhibition by DA
The activity of purified TH in the presence of a range of DNAJC12 concentrations was assayed at 37 °C[72]. TH samples were first diluted with 0.5% (w/v) bovine serum albumin in 20 mM Na-Hepes pH 7.0, 200 mM NaCl. The enzyme was then added to the standard assay mixture (20 mM Na-Hepes, pH 7, 0.1 mg/mL catalase, 10 µM FAS, 50 µM L-Tyr) with varying concentrations (0-1000 nM) of DNAJC12, to a final concentration of 1 µg/mL (4.5 nM tetramer). After a 1 min pre-incubation at 37 °C, the reaction was started by the addition of 200 µM BH₄ and 5 mM dithiothreitol (DTT). The reaction was stopped by 1:1 dilution of the reaction mixture with 1% (v/v) acetic acid (HAc) in ethanol after 5 min. Proteins were removed through precipitation by sample incubation at −20 °C for at least 90 min followed by centrifugation at 20,000 × g for 15 min at 4 °C. The amount of L-Dopa produced by TH through the hydroxylation of L-Tyr was quantified on a 1200 series high-performance liquid chromatography (HPLC) system (Agilent Technologies). The chromatographic separation was obtained using a polar bonded-phase Agilent Zorbax 300-SCX column (4.6 × 150 mm) with 2 mM HAc pH 3.5, 2% (v/v) propanol as mobile phase at a flow rate of 2 mL/min with the fluorescence detector set to λex = 280 nm and λem = 314 nm. To investigate the sensitivity of the TH:DNAJC12 complex to feedback inhibition by DA as compared to TH alone, the same experiment was done by adding either 4.5 nM of TH tetramers or TH:DNAJC12 complex to the standard assay mixture with 0-25 µM DA.

## ATPase measurements
Steady-state ATPase activity was measured using the assay described by Nørby[75]. Assays were performed at 30 °C in 40 mM Na-Hepes pH 7.6, 50 mM KCl, 5 mM Mg acetate and 2 mM DTT, using transparent 96-well flat bottom plates (Sarstedt) in a final volume of 200 µL. Protein concentrations used were 2 µM of Hsc70 (or Hsc70-V438F when indicated), 0.4 µM Apg2 and 2 µM JDP. To test the effect of TH on the ability of the different JDPs to stimulate Hsc70 ATPase activity, 4 µM TH was added to the different JDPs. The ATPase regeneration system consisted of 0.3 mM NADH, 2 mM phosphoenolpyruvate (PEP), 12.5 mg/mL pyruvate kinase (PK), 0.017 mg/mL lactate dehydrogenase (LDH) and 1 mM ATP. ATP consumption rates (µmol ATP/min) were calculated from the slopes of the A340 decay curves over the selected time intervals that showed a linear absorbance decline. The amount of ATP moles hydrolyzed per min was calculated by measuring the absorbance decay at 340 nm for 1 h using a Synergy HTX plate reader (Bio-Tek), using the extinction coefficient of NADH (ε340 = 6220 M⁻¹ cm⁻¹).

## Luciferase aggregate reactivation
Luciferase (2 µM) was denatured in 6 M urea, 40 mM Na-Hepes pH 7.6, 50 mM KCl, 5 mM MgCl2, and 2 mM DTT for 30 min at 30 °C. Aggregation of the denatured substrate was induced by a 100-fold dilution in the same buffer but without urea. Aggregates (20 nM final concentration) were added to samples containing 2 µM Hsc70 and 1 µM JDP in the absence or presence of 0.4 µM Apg2. The reaction was started by addition of 2 mM ATP and the ATP-regeneration system (3 mM PEP and 20 ng/mL PK) in a final sample volume of 100 µl. Luciferase reactivation was monitored at 30 °C for 120 min at 30 °C. Subsequently, 5 µL of sample was added to 50 µL of luciferin (Promega, E1500), and luminescence was measured in a Synergy HTX plate reader (BioTek) using white 96-well flat-bottom plates (Sarstedt). Reactivation percentages were calculated considering the luminescence emitted by the native luciferase as 100% and by untreated aggregates as 0% reactivation.

## Refolding of unfolded luciferase
Luciferase (0.125 µM) was denatured in the absence or presence of Hsc70 (2 µM) and 5 mM ATP in 40 mM Na-Hepes pH 7.6, 50 mM KCl, 5 mM MgCl₂, and 2 mM DTT for 40 min at 42 °C in a final volume of 160 µL. After denaturation, the samples were incubated for 5 min at 30 °C and different combinations of chaperones were added at concentrations of 1 µM JDP and 0.4 µM Apg2 so that the samples

ended up with the binary (Hsc70 and JDP) or ternary (Hsc70, JDP and Apg2) chaperone mixture. The refolding reaction was started by adding 2 mM ATP, 3 mM PEP and 20 ng/mL PK, in a final sample volume of 200 μL (final concentration of luciferase 0.1 μM). The refolded luciferase fraction was measured after 2 h at 30 °C by adding 5 μL of each sample to 50 μL luciferin (Promega, E1500) and recording the luminescence in a Synergy HTX plate reader (BioTek) using a white 96-well flat-bottom plate (Sarstedt). The activity of the non-denatured luciferase in the presence of the ternary chaperone mixture, ATP and ATP-regeneration system incubated at 30 °C for the duration of the experiment was defined as 100% activity.

### Assay to detect delay of protein aggregation

The ability of different JDP proteins to delay the aggregation of Tau was measured by aggregating 20 μM Tau K18 P301L with equimolar amounts of heparin (MP Biomedicals) in the absence or presence of 6 μM JDP. The aggregation was monitored for 16 h at 37 °C, recording the light scattered at 350 nm every 5 min in a Synergy HTX plate reader (BioTek) using transparent 96-well flat bottom plates (Sarstedt) in a final volume of 200 μL.

### Negative Staining Electron Microscopy

TH alone (2.5 μM tetramer) and TH:DNAJC12 (2.5 μM complex) samples were incubated at 37 °C for 100 min in the presence or absence of 10 μM DA in 20 mM Na-Hepes pH 7.0 and 200 mM NaCl. Subsequently, 5 μL aliquots of the samples were applied onto glow-discharged cooper/carbon-coated 200-mesh copper grids and incubated for 1 min. Grids were negatively stained with 2% (w/v) uranyl acetate and air-dried for 5 min. Images of the TH aggregates were taken in a JEOL JEM 1400 Plus electron microscope operated at 120 kV and equipped with a CCD camera (4 K × 4 K TemCam-F416, TVIPS). Images were recorded at a 50,000 × nominal magnification with a sampling rate of 2.4 Å/px. The representative negative-stain EM images were observed in independent experiments with three different protein preparations. To measure the diameter (nm) of the TH aggregates, ImageJ software was implemented.

### Cryo-EM data acquisition

Aliquots of 4 μL of TH:DNAJC12, TH:DNAJC12(108-198) and TH(DA):DNAJC12 complexes (6-8 μM) were vitrified using a Vitrobot Mark IV (FEI) and incubated on UltrAuFoil R 1.2/1.3 300 mesh grids, blotted for 2 s at 22 °C and 95% humidity and plunged into liquid ethane. The cryo-EM grids were first checked on a 200 kV FEI Talos Arctica equipped with a Falcon III direct electron detector at the Centro Nacional de Biotecnología (CNB) cryo-EM facility followed by data acquisition on a FEI Titan Krios electron microscope operated at 300 kV, equipped with a Gatan Quantum K3 Summit direct electron detector at the Cell and Molecular Imaging Platform facility (SciLifeLab; Stockholm University). Additionally, another large acquisition of the TH:DNAJC12 complex on a FEI Titan Krios electron microscope equipped with a K3 direct electron detector and a Quantum LS filter was performed at the European Synchrotron Radiation Facility (ESRF). A total of 21,430 movies for TH:DNAJC12, 18,329 for TH:DNAJC12(108-198) and 12,826 for TH(DA):DNAJC12 were acquired at a nominal magnification of 105,000x (corresponding to a pixel size of 0.85 Å/pix), with a defocus range of 1.0 to 2.6 μm. Exposure was set to ~1 e-/Å/s, and 40 frames were collected in total, with an overall dose between 40 and 42 e-/Å for all the complexes.

### Image processing and 3D reconstruction

Image processing of TH:DNAJC12, TH:DNAJC12(108-198) and TH(DA):DNAJC12 complexes was performed following a similar workflow (Supplementary Fig. 9). All programs used for image processing to obtain the different 3D maps were implemented in the Scipion[76]

software platform. First, the movies were aligned using MotionCor2[77], and the outputs were subjected to CTF determination using Gctf[78]. To save computational time during the first processing steps and to increase the signal-to-noise ratio, particles were down-sampled by a factor of 3 and were automatically picked with Xmipp3[79] auto-picking software. The 6,534,161 (TH:DNAJC12), 1,283,996 (TH:DNAJC12(108-198)) and 2,474,620 (TH(DA):DNAJC12) extracted particles were subjected to several 2D classifications using Relion 2.0[80] and Cryosparc[81] to exclude bad particles and ice contamination. The selected classes were subjected to several further rounds of 2D classification. Some of the best 2D classes were used as a template to generate an initial model using both Cryosparc and RANSAC[82]. Other initial models were obtained by filtering the atomic structures of the human apo-TH (PDB 7A2G) and TH(DA) (PDB 6ZVP). In all cases, models were low-pass filtered to 50 Å and used for a 3D classification of the selected particles contained in the best 2D classes performed without symmetry imposition. The classes of each complex resembling the TH structure were selected and, given the stoichiometry of the TH:DNAJC12 complex (2:1), a further 3D classification was performed using D1 symmetry. The particles of the best classes were used for a further 3D auto-refine using Relion 2.0 and yielded final maps of 5.0 Å (TH:DNAJC12(108-198)) and 5.0 Å (TH(DA):DNAJC12) resolution, as estimated by the Fourier shell correlation (FSC) method, with a cut-off of 0.143. Additionally, the map from TH:DNAJC12 was subjected to further classification using the 3D Variability protocol from Cryosparc 4.4.2 aiming to define in more detail the binding region between TH and DNAJC12. The best class was then refined by 3D auto-refine using Relion 3.1.3 yielding a resolution of 4.25 Å, as estimated above. For each final 3D map, local resolution was calculated by Xmipp3-MonoRes[37] using the half-maps of the final volumes (Supplementary Fig. 10). To identify the masses corresponding to DNAJC12 domains in the different final 3D reconstructions, the density from apo-TH map was subtracted from TH:DNAJC12, TH:DNAJC12(108-198) and TH(DA):DNAJC12 maps. To this end, the volumes used were first normalized and filtered to the same resolution (5 Å) and then subtracted with the "vop subtract" option in the Chimera[83] package. To enhance accuracy, anisotropy resolution was accounted for when generating the difference maps. This approach focused on local variations (especially those observed in RDs) rather than the overall resolution. This approach allowed a precise identification of DNAJC12-associated densities within the different final maps (Supplementary Fig. 11).

### Molecular docking with HADDOCK

XL-MS-guided models for TH:DNAJC12, TH:DNAJC12(108-198) and TH(DA):DNAJC12 complexes were generated using HADDOCK 2.4 (High Ambiguity Driven protein-protein DOCKing) data-driven docking protocol[38]. XL-MS data generated with BS3 were employed as unambiguous restraints for protein-protein docking within the HADDOCK 2.4 framework. The docking process was executed through the Expert mode interface of HADDOCK 2.4. To enhance the accuracy of the initial docking stage, the final 3D reconstructions of the different complexes were incorporated into iteration 0. Centroid restraints were implemented with a scaling factor of 300. Additionally, center of mass restraints were applied to further guide the docking process and improve the overall quality of the predicted protein-protein complex structures.

### Crosslinking experiments and mass spectrometry analysis

30 μg of the different TH:DNAJC12 complexes (TH:DNAJC12, TH:DNAJC12(108-198), TH(DA):DNAJC12, TH RD:DNAJC12 and TH RD:CTD) were subjected to chemical crosslinking by incubation with 3 mM BS3 in PBS pH 7.4 for 30 min at RT (n = 1). The reactions were quenched for 15 min at RT by adding 50 mM Tris-HCl pH 7.0. BS3-crosslinked samples were incubated in Laemmli sample buffer (0.02% [w/v] bromophenol blue, 2% (w/v) SDS, 10% (v/v) glycerol, 60 mM Tris-

HCl pH 6.8) for 5 min at 96 °C and loaded onto a precast 4-15% gel (BIO-RAD). The gel was stained with Quick Coomassie (Generon) and the bands corresponding to the different complexes according to their theoretical molecular mass were excised (Supplementary Fig. 16) and subjected to automated reduction with TCEP, alkylation with chloroacetamide and trypsin digestion in an OT2 robot (Opentrons) as described in Shevchenko et al.[84]. The resulting peptide mixture was speed-vac dried and re-dissolved in 0.1% formic acid for LC-MS/MS analysis. This was carried out in an Ultimate 3000 nanoHPLC (Dionex) coupled online to an Orbitrap Exploris 240 (Thermo). The HPLC was equipped with a PepMap Neo C18 trapping column (300 μm × 5 mm; Thermo) and a PepMap RSLC c18 column (75 μm x 50 cm; Thermo). Solvent A and B were, respectively, 0.1% formic acid and 0.1% formic acid in acetonitrile. Separation was performed at 50 °C at a flowrate of 250 nl/min under the following gradient: 4% B for 2 min, linear increase to 65% B in 63 min, linear increase to 50% B in 6 min, linear increase to 95% B in 4 min, 95% B for 4 min, back to 4% B in 1 min and 4% B for 10 min.

The mass spectrometer was operated in DDA mode. Each acquisition cycle had a maximum duration of 3 s and consisted of a survey scan (375–1200 m/z) at 60000 resolution (FWHM) and up to 20 MS/MS scans at 15000 resolution (FWHM). Peptides with charges 2 to 5 were selected for fragmentation applying a dynamic exclusion window of 45 s. For peptide identification, raw MS data were converted to mgf files with Proteome Discoverer v2.5 (Thermo) that were used for a database search with MeroX 2.0[85] against a custom-made database containing the sequences of the proteins in each complex. Search parameters were set as follows: trypsin as enzyme allowing 2 (K) and 2 (R) missed cleavages, BS3 as crosslinker, MS tolerance of 10 ppm and MS/MS tolerance of 20 ppm, carbamidomethylation of cysteines as fixed modification and oxidation of methionines as variable modification. Peptide identifications were filtered at an FDR < 5%. Alternatively, the database search was conducted with XlinkX (Thermo) with the same database and search parameters except for the number of missed cleavages that was set to 3 (Supplementary Table 2). The outputs from both search engines showed a high correlation (75% of the total are shared among both analyses; see Supplementary Table 3, crosslinks displayed in green). Given the quality of the crosslinks identified by MeroX and XlinkX, all filtered crosslinks (common and unique) from both engines were used for the structural analysis. Note that only those crosslinks corresponding to residues of structured regions of TH and DNAJC12 were represented in the proposed models. Although conventional constraint distances allowed by BS3 are up to 25 Å[86], a broader range (up to 35 Å) was applied to take into consideration protein dynamics and conformational flexibility[87,88]. Taking into account that TH is a dimer of dimers consisting of 4 different subunits, distance measurements were calculated of all possible intermolecular contacts for each interprotein crosslink (Supplementary Table 4). The distances that fall within the BS3 constraints established in this work were displayed and represented using Chimera[83] package. XL-MS data have been deposited to the ProteomeXchange Consortium via the PRIDE partner repository with the dataset identifiers PXD045185 and PXD056328.

### Statistics and reproducibility
GraphPad Prism software v8.2.1 was used for all statistical analyses. Three independent protein samples were analyzed using CD and SRCD, and the mean ± SD Δε measured at a wavelength of 222 nm for each temperature was reported. To calculate the $T_m$, the Δε measurements were fitted to a sigmoidal curve using the *Boltzmann* equation ($Y(x) = I_{max} + ((I_{min} - I_{max})/1 + e^{((T_m - T(x))/a)}$, where $I_{max}$ and $I_{min}$ are the highest and lowest Δε values measured and $a$ is the slope of the curve at the point of inflection ($T_m$). For the DSF, TH activity measurements, and the chaperone assays (aggregate reactivation, substrate refolding and ATPase activity assays), at least three samples were

prepared and measured prior to performing multiple comparisons using one-way analysis of variance (ANOVA) and a post-hoc Tukey HSD test. Results were considered to be significantly different when p < 0.05. Three independent samples were also prepared and measured using DLS, with final average particle sizes reported as mean ± SD. The comparison between TH alone and with DNAJC12 were done using an unpaired Student´s $t$-test, where significant differences were identified when p < 0.05. For the DA inhibition assay, the individual curves were fitted to a four-parameter logistic curve by the *[Inhibitor] vs. response−Variable* slope equation ($y = min + ((max−min)/(1 + IC_{50}/X)^{Hill\ Slope}$) in GraphPad. The mean $IC_{50}$ ± SD was reported and compared using an unpaired Student´s $t$-test (p < 0.05). For BLI, the binding responses recorded for every concentration of DNAJC12 used were plotted and fitted to a saturation binding curve using the *One site binding (hyperbola)* equation ($Y = (B_{max} \cdot x)/(K_D + x)$) in GraphPad to derive the $K_D$ value of the interaction. The $K_D$ values calculated from three independent experiments were reported as mean ± SEM and were compared using ANOVA followed by a post-hoc Tukey HSD test, where a significant difference was defined at p < 0.05.

### Reporting summary
Further information on research design is available in the Nature Portfolio Reporting Summary linked to this article.

## Data availability
Cryo-EM data generated in this study have been deposited in the Electron Microscopy Data Bank under accession codes EMD-18047, EMD-18058 and EMD-18289 for TH:DNAJC12, TH:DNAJC12(108-198) and TH(DA):DNAJC12 complexes respectively. The MS proteomics data have been deposited to the ProteomeXchange Consortium via the PRIDE repository with the dataset identifiers PXD045185 and PXD056328. All other data generated in this study are provided in the Supplementary Information. Source data are provided with this paper.

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

## Acknowledgements

We acknowledge the support of the Spanish State Research Agency through grants PID2022-137175NB-I00 (AEI/FEDER, UE) and AEI/10.13039/501100011033, through the "Severo Ochoa" Programme for Centres of Excellence in R&D (CEX2023-001386-S) to J.M.V. and J.C.; the Stiftelsen K.G. Jebsen (SKJ-MED-02; Center for Research in Parkinson's Disease), The Neuro-SysMed Center (Research Council of Norway (RCN), Project No. 288164) and Fundació la Marató de TV3 (202012-31) to Au.M and K.J.-KC, and the Meltzer Research Fund (Project 103578) to M.D.S.T. This research was supported by grants PID2020-119210RB-I00 (MICIU/AEI/10.13039/501100011033) and IT1745-22 (Basque Government) to Ar.M. and F.M. We also acknowledge the Biophysics, Structural Biology and Screening (BiSS) core facility at the University of Bergen (UiB) and funding from NOR-OPENSCREEN (RCN, Grant No. 245922; and expert help from Dr. Svein I. Støve, Trond-André Kråkenes and Gro Haugseng). The authors acknowledge the support of the European Union (EU) and Horizon 2020 through Instruct Proposal PID 20290 and the CRIOME-CORR project (ESFRI-2019-01-CSIC-16) to the cryoEM CNB-CSIC facility. We are thankful for access to the AU-CD beamline on the ASTRID2 synchrotron at ISA, Aarhus University (Denmark) and for expert help from Dr. Nykola Jones. We are grateful to the Cell and Molecular Imaging Platform facility (SciLifeLab) at Stockholm University for access to a Titan Krios cryoelectron microscope (Projects CEM00419 and CEM00521) and to Dr. Marta Carroni for expert help. We are also very thankful to the European Radiation Facility (ESRF) in Grenoble for access to a Titan Krios cryoelectron microscope (thanks to the proposal MX2443) and for access to the PNDdb (http://www.biopku.org). The professional editing

service NB Revisions was used for technical preparation of the manuscript prior to submission.

Sadly, Arturo Muga passed away in April 2024.

## Author contributions

M.D.S.T., M.I.F. and G.G.-A. purified TH and DNAJC12, and all truncated forms and variants used in this work, and prepared the TH:DNAJC12 complexes. M.D.S.T., M.I.F., G.G.-A. and J.P.K. carried out the biophysical analyses; F.M., Ar.M., M.D.S.T., L.V.-C., and K.J.-KC. performed biochemical experiments; J.C., L.O., J.M. and D.G.-C. acquired the cryo-EM images and did the image processing; J.C., J.M.V., L.O., M.M. and J.M. did the crosslinking/MS analysis; C.S. prepared the structural models and J.M. and Au.M. prepared the composite models. M.D.S.T., M.I.F., J.P.K., Ar.M., J.M.V., J.C. and Au.M. designed the experiments. Au.M., J.M.V. and J.C. managed the project and wrote the paper with main contributions from M.D.S.T. and corrections from all authors.

## Funding

## Competing interests

The authors declare no competing interests.
