## [Transparent Peer Review file · Nature Communications]

Structural recognition and stabilization of tyrosine hydroxylase by the J-domain protein DNAJC12

Corresponding Author: Professor Aurora Martinez

Version 0:

Reviewer comments:

Reviewer #1

(Remarks to the Author)

The manuscript by Tai et al. describes the structural characterization of the interaction of tyrosine hydroxylase (TH) with its chaperone DNAJC12, using biochemical, molecular biology and biophysical methods, cryoelectron microscopy and in silico approaches, all state-of-the-art techniques that strongly support the main conclusions of the work; namely, the unravelling of the binding stoichiometry of DNAJC12 with TH, the key residues involved and the functional consequences, TH stabilization without perturbing its interaction with substrate and regulatory molecules (dopamine). Overall, it is a rigorous study that describes the conformational properties of the TH-DNAJC12 complex and provides an explanation of the pathogenic mechanism of the most frequent DNAJC12 variant, that results in a truncated protein lacking the last 24 carboxyl-terminal residues.

DNAJC12 and TH variants are responsible for parkinsonism and related phenotypes. The authors have provided novel results that represent significant advances in the field, as they have now characterized the structure of the full-length DNAJC12 and identify a C-terminal octapeptide as essential for binding TH, thus stabilizing it and preventing its aggregation. These results are of value in the classification of DNAJC12 variants which are increasingly being identified in patients with hyperphenylalaninemia, dystonia and a diverse range of intellectual disabilities. Although not an expert in structural biology nor in some of the methodologies used, it is clear to me that conclusive experimental evidence is provided, the methods employed are described in sufficient detail, and the data are adequately interpreted. Of note, the manuscript is very well written and easy to follow with each subsequent experimental step adequately explained and justified.

I have some comments about the relevance of the structural analysis in disease and issues that need clarifying, as described below:

- DNAJC12 variants generated by deletion mutagenesis: I assume all of them are produced fused to MBP and 6xHis; are they all expressed at similar levels? Are the large differences in the yield of purified proteins due to difference in stability once MBP and His tags are removed?

- While the mechanism of two common DNAJC12 variants is discussed, nothing is mentioned about TH variants. It would be interesting to know if there are TH pathogenic variants in the regions where DNAJC12 monomers interact/are bound. Can TH deficiency be caused by a deficient interaction with a wild-type DNAJC12?. This information would add to the relevance of the work.

Minor comments, please correct:

Figure 4a, there are red and orange asterisks, not black, as mentioned in the figure legend and the text Reference (17), please update

Reviewer #2

(Remarks to the Author)

J-domain proteins (JDP) are obligatory cochaperones of Hsp70s that are involved in all aspects of protein homeostasis; protecting proteins against aggregation and assisting their folding, transporting them across membranes and refolding them following their aggregation upon stress, and if that fails, directing them for proteolytic degradation. These general functions

are usually carried out by systems involving class A and B JDPs and their Hsp70 partners, while JDPs of class C are typically more specialized. For example, they function in uncoating of clathrin vesicles, biogenesis of histones, iron-sulfur cluster biogenesis, and pre-mRNA splicing. But despite the functional diversity of class A, B and C JDPs, a single biochemical mechanism, that is cyclic interaction with substrate polypeptides, drives functionality of Hsp70 systems.

The reviewed paper focuses on the class C JDP termed DNAJC12, whose mutational variants are implicated in a neurometabolic disorder that manifests as hyperphenylalaninemia, accompanied by dopamine and serotonin depletion. Previous works involving co-immunoprecipitation in human cell and murine liver lysates revealed that DNAJC12 binds to tyrosine hydroxylase (TH), the enzyme of dopamine synthesis pathway, and that defective interaction causes disease. The authors used purified DNAJC12 to characterize its interaction with TH both biochemically and structurally. They demonstrated that two monomers of DNAJC12 bind to a TH tetramer and using truncated variants of DNAJC12 showed that this interaction involved eight conserved C-terminal residues of DNAJC12. The authors also find that DNAJC12 binding delays time-dependent aggregation of TH while maintaining enzymatic activity and inhibition by dopamine. The authors used a combination of cryoelectronic microscopy and chemical crosslinking to obtain a structural model of the DNAJC12-TH complex. The model reveals that C-terminal segments of DNAJC12 interact with a dimer of regulatory domains of TH such that the active sites are available for substrate and dopamine interaction. In the DNAJC12-TH complex, the J-domain of DNAJC12 is free to interact with its Hsp70 partner, as demonstrated by stimulation of the ATPase activity of Hsc70 by DNAJC12-TH complex. Interestingly, no such stimulation is observed for DNAJC12 alone.

The results presented in this paper significantly expand our knowledge about DNAJC12 and contribute significantly to our understanding of the pathology associated with its malfunction.

The results are noteworthy and supported by data, which were correctly analyzed. The methodology used is sound and meets the standards in the field of chaperone research. The manuscript is written very clearly and contains enough information for the work to be reproduced. Therefore, in my opinion, this paper is appropriate for publication in Nature Communication after the issues listed below are resolved- minor revision.

Important issues:

1. (line 126) The C-terminal, unstructured, domain of DNAJC12 contains evolutionary conserved alpha-helical fragment, which the authors termed "helix V". This name could be confusing, because the same name is used for the conserved alpha-helical fragment present in the G/F region of DNAJB1 and other B class JDPs [Faust, O. et al. HSP40 proteins use class-specific regulation to drive HSP70 functional diversity. Nature 587, 489–494 (2020)]. The helix V of B class JDPs is involved in J-domain autoinhibition, which plays an important role in its disaggregation activity. Because there is no evidence that helix V of DNAJC12 and the unstructured region of this protein are homologous to helix V and the G/F region of class B JDPs respectively, to avoid any confusion, this C-terminal alpha-helix should have a different name.
2. (line 443) The observation that the DNAJC12-TH complex stimulates the ATPase activity of Hsc70 more efficiently than the DNAJC12 alone could be interpreted in at least two alternative ways: (i) the release of J-domain autoinhibition, as proposed by the authors and/or (ii) synergistic stimulation of the ATPase by the J-domain of DNAJC12 and substrate TH. The latter scenario could be verified experimentally by using a Hsc70 substitution variant defective in substrate binding – substitution of a key residue in the substrate binding cleft of Hsc70. However, considering that the manuscript already contains many important results, I do not insist that the control experiment described above must be done, but an alternative interpretation of the obtained results should be added to the text with explanation how it could be tested experimentally.

Minor issues:

3. (Fig. 1) I would like to propose the following changes: add the structural model of DNAJC12 and map disorder and hydrophobicity on this model- move panel b and c to the supplement, make the amino acid sequence of DNAJC12 smaller but such that identification of individual residues will be simpler e.g., by adding residue number scale above the sequence.
3. (Fig. 2) A: SDS-PAGE elution profiles for TH and TH-DNAJC12 complex should be added to the supplementary information
E, F: traces of BLI measurements should be added to supplementary information.
4. (Fig. 3C) – concentration of TH in the reaction mixture should be indicated either in Fig. 3C or in the legend. Why different concentration scale for DNAJC12 (nM) and for DA (microM) – microM are in the main text for both DNAJC12 and DA (line 264-5)
5. Fig. 4 catalytic (CD) and oligomerization (OD) domains should be indicated in Fig. 4B
6. (Line 293) there is no black asterisk in Fig. 4a (red and orange)
7. (Line 441) ...strong activation of Hsc70 ATPase... The two-fold activation of Hsc70 ATPase is "measurable" and expected – but I would not call it "strong"- as 20-30 fold, or even higher, activation is typically observed in the presence of JDP/substrate/nucleotide exchange factor
8. Fig. 6a – traces of BLI measurements should be added to the supplement
9. Fig. 7d – why the y-axis is broken (has a gap) - it is confusing, while the amount of the saved space is negligible – the

same true for Suppl. Fig. 12.

10. Suppl. Fig. 3 the helix V (alpha-5) shown on top of the multiple sequence alignment is predicted only for JC12 or for all sequences in the alignment? Please add this information to the legend.

11. Suppl. Fig. 10 The structural model presented here predicts that TH is not a substrate for the Hsp70. If this is indeed the case it should be stated in the text and discuss. (line 579) suggests that the authors consider TH as a substrate/client for Hsc70? (line 628-631) also suggests that TH is a substrate for Hsc70?

12. Suppl. Fig. 13a – add traces of the BLI measurements

Reviewer #3

(Remarks to the Author)

In their study Tai et al. investigate the role of the heat shock Hsp40/ J- domain protein DNAJC12 in the structural recognition and stabilization of tyrosine hydroxylase (TH). Using recombinantly expressed and purified protein variants the authors show that DNAJC12 exists as a monomer in-vitro and binds TH in the stoichiometry of 2 DNAJC12 molecules per TH tetramer. By generating DNAJC12 mutants the authors then go on to show that a conserved C-terminal region containing the RKFRNYEI peptide within DNAJC12 mediates binding to TH. This region is missing in the pathogenic DNAJC12 mutant Trp175Ter and the mutant is therefore unable to bind TH, as shown by the authors. The authors also obtain a relatively low-resolution structure of the TH:DNAJC12 complex by cryoEM and use it in combination with cross-linking mass spectrometry (XL-MS) to locate DNAJC12 and to demonstrate that DNAJC12 interacts with TH via its regulatory domains, thereby leaving the active sites of TH accessible and explaining why DNAJC12 binding does not impede TH activity.

Overall, this is an interesting and well-designed study that employs biochemistry, a large number of biophysical assays, and structural biology to show how the chaperone DNAJC12 interacts with TH, thereby clarifying on a molecular level its important role in stabilizing TH while preserving its activity, at least in-vitro. The manuscript itself is well written and the conclusions are supported by the data.

However, some points regarding the structural and in the particular the XL-MS data, are unclear in the current form of the manuscript and require at least clarification.

In the methods the authors write that the XL-MS data was acquired on a triple TOF mass spectrometer using BS3 as a cross-linker and MeroX 2.0 as a search engine, using a FDR of 5 % and an XlinkX score for validation.

This seems inconsistent.

BS3 is a non-cleavable cross-linker but MeroX is only capable of searching data of cleavable cross-linkers. How was this possible?

Also, XlinkX is an alternative cross-linking search engine embedded within Proteome Discoverer from the Heck & Scheltema labs. How could it be used to validate data searched by MeroX?

And more generally. How exactly did the authors use a triple TOF MS to fragment and identify cross-linking data. As this is a rather unconventional approach in cross-linking MS, the authors should please expand on how this was done.

The authors state that they have used a custom-made database to search the data -and presumably to calculate the FDR. How was this done exactly and on which level (protein, peptide..) was the FDR calculated?

Minor points:

Regarding obtained cross-linking data itself. The XL-MS data seems to indeed agree well with the obtained cryo-EM maps, as stated by the authors. The authors do however use quantitative statements on the cross-linking data as “the region with the highest number of cross-links was the CTD” (page 9) for their argumentation. This should be done with caution or only after additional analyses as the authors do not show the overall distribution of cross-linkable lysines within TH or DNAJC12, nor the number of identified mono or intra-links. Without this information it is hard to judge which cross-links could have possibly formed in the first case and thus, if the amount of identified inter protein cross-links is indeed comparatively high. Moreover, Figure 5 shows that many of these cross-links emanate from the same lysine (i.e. lysine K183 in Figure 5B), indicating that this lysine is particularly reactive and thus again putting such quantitative statements at least somewhat in perspective.

Figure 5 also shows that some cross-links bridge seemingly rather large distances (see for example link between K150 and K240). As conventional BS3 should only bridge distances of up to 25 Angstrom, this may indicate a certain amount of false positive data within the dataset. Likewise, the authors allow very short peptides with 3 or 4 amino acid length as input (see Supplementary Table 3). Such cross-links tend to be ambiguous, at least in more complex data. Maybe the authors could also comment on this.

Reviewer #4

(Remarks to the Author)

In this manuscript by Tai et al, the authors describe in an elegant series of biochemical and structural experiments to explain how a molecular chaperone from the J-domain protein family, DnaJC12, stabilizes Tyrosine Hydroxylase (TH) and how mutations in the chaperone interfere with this function and likely lead to disease. Overall, the study is well done with a lot of validation but there are some missing pieces and inconsistencies that need to be resolved. We recommend the following revisions, after which the manuscript may be suitable for publication in Nat Comm.

Summary

1. Variants of DnaJC12 cause neurometabolic disorders
2. Variants of tyrosine hydroxylase which converts L-tyrosine to L-dopa cause THD
3. DnaJC12 purified as a monomer; Unstructured CTD, binds with high affinity with TH
4. TH:DnaJC12 complex maintained TH activity
5. The C-terminal region of DnaJC12 is essential for binding to TH

Comments

1. N-terminal truncation of DnaJC12 (missing the J-domain) shows no binding to the TH. In the structures the J-domain appears to be important for the interaction with TH. How is it accessible for Hsp70 recruitment and stimulation of ATPase activity? Perhaps the inability of the J-domain alone binding to TH is more consistent with the above idea of accessibility to Hsp70 but then is the structure is not reporting on the correct conformation? This disconnect needs to be clarified in the manuscript.
2. What is the role of Hsp70 in this process? Does Hsp70 bind to the C-terminus of DnaJC12 to remove it from the DnaJC12:TH complex? Additional experiments must be included to explain how this works – this seems unresolved. Maybe crosslinking experiments can be used to explain how Hsp70 first binds to the J-domain of DnaJC12, what is the next step? Does the DnaJC12 C-term bind to the substrate binding site on Hsp70? How does this affect the DnaJC12:TH complex?
3. BLI data analysis. How was the data fit? 1:1 or 1 1:2 stoichiometry model? Please report kinetic traces, fits and kinetic parameters for the BLI binding measurements. Supplementary Table 1 reports the affinities in μM which is likely incorrect given the analyte concentrations used in the BLI experiments.
4. The binding data, SEC-MALS and structural biology of the complex is very nice evidence of high affinity. In the end, it is not clear how nM affinity is achieved for the DnaJC12:TH interaction. The authors present a structure of the complex, it seems reasonable to test targeted mutations rather than truncations – the details of the interaction can be inferred and directly tested experimentally. This would significantly add to the completeness of the story.
5. “DnaJC12:TH complex stimulated Hsp70 hydrolysis but DnaJC12 prevented aggregation in an Hsp70 independent manner”. Related to comment above – this has to be clarified. What is the role of Hsp70 in this activity?
6. Indicate wavelength used to measure changes in secondary structure to probe stability in Fig 1e.
7. What is suggested by the cumulative stabilizing effect? Lines 243-244 “At least in vitro, TH is vulnerable to time-dependent aggregation [39] that can be delayed by certain ligands, in particular DA [40]”.
8. The DSF T_m changes seem pretty subtle (Fig 3a,c). How was significance calculated? Given this affinity, I would expect a larger increase in stability of the DnaJC12:TH complex. Perhaps using more established methods such as CD (as shown in Fig. 1e) to measure changes in stability would be more informative.
9. Lines 253-256. The size distribution of the samples of TH:DnaJC12 remained similar at 0 min and 100 min, while larger species and aggregates dominated the size distribution of the samples containing TH alone at the same endpoint (Supplementary Fig. 6). At 100 mins TH alone produces a shifted population that is small plus large species. The smaller shifted population at 100 mins looks pretty defined. Have the authors looked at these samples using TEM or alternate methods? Are these assemblies uniform in terms of size? I would expect that if the samples aggregated it would be a broader size distribution. These experiments should also include +/- DA.
10. Lines 258-268. How is the conversion from L-Tyr to L-Dopa? The authors should include a more detailed description of how TH activity was measured?
11. Supp Fig 4 should be a part of the Main Fig 4 panel. In Fig 3, the font is small and hard to read.
12. In explanation of the cryo-EM data, the authors have mentioned: “This domain amounts 303 to half of the mass of the whole cochaperone, so we assumed that the JD should be assigned to 304 the largest of the two masses described (red asterisk in Fig. 4a and orange mass in 305 Supplementary Fig. 9a”. In the figure legend it is mentioned as yellow mass.

13. Overall, the figure legends need to be improved for clarity, concise descriptions that help explain the data. The legend of Fig 4 is difficult to follow, please clarify.

Figure 4. 3D reconstruction of the TH:DNAJC12, TH:DNAJC12(108-198) and 1298 TH(DA):DNAJC12 complexes. 1299 Two views of the TH:DNAJC12 (a), TH:DNAJC12(108-198) (c) and TH(DA):DNAJC12 (e) complexes. The red 1300 and black asterisks in a) point to the extra masses that appear in the 3D reconstruction when compared with that 1301 of apo-TH (PDB 7A2G). The red circles point to the location of two of four TH active sites. The red asterisks in 1302 e) point to the N-terminal regulatory α -helix (residues 39-58) of TH in TH(DA) (PDB 6ZVP). The same two 1303 views of TH:DNAJC12 (b), TH:DNAJC12(108-198) (d) and TH(DA):DNAJC12 (f) complexes with the atomic 1304 structure of apo-TH and the AlphaFold model of DNAJC12 docked into the cryoEM volume (docking is 1305 performed only in one of the corresponding masses, the top one). In the atomic structure of apo-TH (PDB 7A2G) 1306 (b,d) the central, tetrameric CD+OD domains are colored in cyan and the two dimeric RDs in dark blue. The same 1307 color code is used for TH in TH(DA) (PDB 6ZVP), with the N-terminal regulatory α -helix shown in magenta (f). 1308 For the AlphaFold atomic model of DNAJC12, the JD (identical to the atomic structure obtained by NMR; PDB 1309 2CTQ) is colored in orange, the CTD in red and the central linker region in dark green. This is an unstructured 1310 region and many other options are possible.

14. Fig 7: What could be the possible role of the DnaJC12:TH complex activating ATPase activity of Hsp70? What happens to the DnaJC12:TH complex following Hsp70s activation? Does Hsp70 disassemble the complex? Please annotate the Figure to show what light and dark grey is.

15. The authors should include gels of the crosslinked samples to confirm complex formation and to ensure that the data is not derived from aggregated species.

16. Some of the lysines in the models look to be buried and in some other instances the crosslinks appear to thread through proteins. It might be good to calculate the distances using a solvent accessible path with software such as Xwalk (PMID 21666267). For the XL-MS analysis please also include plots illustrating the distance distributions of the intramolecular and intermolecular mapped crosslinks. Some of the contacts appear longer than the spacing of the crosslinker (and they may appear even longer if using solvent accessible paths). Have the authors exhaustively mapped all possible intermolecular contacts to confirm that the geometries shown are optimal for the contacts?

17. Line 486, what do the authors mean by "heterogenous clinical spectrum". Does this imply that conditions that include hyperphenylalaninemia, L-Dopa responsive dystonia, severe parkinsonism and intellectual disability are all linked to DnaJC12 loss-of-function?

Reviewer #5

(Remarks to the Author)

This manuscript describes the interaction of the molecular chaperone DNAJC12 with tyrosine hydroxylase (TH) tetramer, a key enzyme involved in dopamine synthesis. TH is prone to aggregation in vitro and mutations in DNAJC12 are linked to neurometabolic disorders and depletion of dopamine. Thus, understanding the role DNAJC12 in chaperoning and stabilizing TH is of significant interest. Biochemical experiments including SEC-MALS, DSF, and TH activity assays were used to identify that two DNAJC12 molecules bind and stabilize TH tetramer. The authors perform binding analyses to show the conserved C-terminal domain of DNAJC12 and is involved in the interaction. While DNAJC12 reduces aggregation of TH, there appears to be no significant change in TH activity with DNAJC12.

Importantly, the authors perform Cryo-EM and crosslinking-mass spec to characterize the structure of the TH- DNAJC12 complex. While the authors claim DNAJC12 binds to the TH regulatory region, the resolution of the map is not sufficient to convincingly localize DNAJC12 or define the interaction. Indeed, the RD is flexible and at low resolution in their previously published structure of the TH tetramer and the current maps do not clearly show additional density that corresponds to the JD or the CTD of DNAJC12 given the low thresholding values used in the map images. Furthermore, their reported resolution of 5.6 Å by FSC appears to be wrong based on the map density. Virtually no structural feature (e.g. alpha helices) can be observed in the images shown and the notable regions of possible DNAJC12 density are likely at ~15 Å resolution or worse, meaning that it is impossible to identify the small JD and CTD regions of DNAJC12 that are expected to be structured and bound. Perhaps the FSC curve is reporting an artificially high resolution because of masking artifacts? Since the authors do not show the necessary validation data (fsc curves for map and model, masked, unmasked, angular sampling plot, etc.), it is difficult to know. The TH tetramer core may be at ~5 Å resolution based on supplementary figure 8a however it is certainly not 3 Å, which is what is claimed, and there appears to be substantial angular sampling/preferred orientation issues based in the smeared density.

Additionally, crosslinking-mass spec results are not convincing and do not support or validate their cryoEM model. Notably the analyzed crosslink sites on DNAJC12 appear highly selected and are largely from flexible, unstructured regions outside the cryoEM map and are thus not useful for localization. Overall, although the biochemical experiment results are important in revealing strong 2:4 binding of DNAJC12 to TH and providing insights for a potential stabilizing role of DnaJC12 in this complex, the cryo-EM and mass spectrometry approaches are not suitable for publication and need to be corrected and advanced substantially to define the interaction between DnaJC12 and TH.

Version 1:

Reviewer comments:

Reviewer #1

(Remarks to the Author)

The revised version of the manuscript has correctly addressed all the issues I raised concerning the previous version. Overall, the revised manuscript has improved in clarity and in quality, presenting novel results that add to the scientific relevance and to the soundness of the conclusions. In my opinion, the manuscript is now acceptable for publication.

Reviewer #2

(Remarks to the Author)

The authors have addressed all comments in my review. However, before the manuscript is accepted for publication it requires a major revision of the text.

The current version of the manuscript is much less clear than the original one because the authors added new information in answer to the reviewers' comments, without considering how they affected the ability of the reader to understand the manuscript. The current version of the Results contains too many speculations, which should be moved into the Discussion. Whereas the Discussion contains too many repetitions from the Results. Both chapters should be significantly shortened. Below I have listed a few fragments of the text that illustrate my point.

Results

Line 398: "It is important to mention that fifteen crosslinks host residues from the flexible linker, indicating an interesting role in the interaction with TH." – this sentence is confusing, and as written does not contribute to better understanding of the C12/TH system.

Line 414 "Of note, docking of TH:DNAJC12, focusing on a superposition of the JD of DNAJC12 with the JD in the ATP-bound open conformation of Hsp70 of Escherichia coli (DnaK).....- this sentence does not fit to this paragraph, because JD (J-domain) is not mentioned in it.

Line 429 "TH is assembled as a dimer of dimers through the ODs that include dimerization and tetramerization interfaces and form the central antiparallel coiled-coil bundle common for the AAAHs (Fig. 6b). By embracing the RD dimers, formed by monomers from subunits in different dimers, DNAJC12 stabilizes the TH tetramer. The stabilization of this dimeric interface may protect TH from dissociation to monomers and exposure of aggregation prone regions (See Discussion)" - this added fragment is confusing and belongs in the Discussion not Results.

Lines: 446 – 450 "Mutations at protein-protein interactions (PPIs) often result in decreased binding affinity (39,50), but they may also cause tighter binding due to improvement of packing and reduction of eventual steric hindrances (51). Variants L141A and L145A may thus present some release of the interactions with Helix 1 and/or higher packing towards F193 in the CTD of DNAJC12 (Fig. 6c, excerpt I)." - these speculations should be in the Discussion.

Line 460 – "Interestingly, this model also supports the holdase activity of DNAJC12, as it shields the β -strand regions 82LNLLLFSP88 and 129EYFVRLE135 (Fig. 6c, excerpts III and IV), which have been predicted to bind to the Hsp70 family by Limbo75 and ChaperlSM76 algorithms....." – this added fragment belongs in the Discussion.

Line 468 – "As a JDP, DNAJC12 could display disaggregating and (re)folding activities in addition to the ATP-independent autonomous holdase activity aimed at avoiding aggregation of (partially) unfolded client proteins, as showed with TH as substrate..." – this sentence is confusing and it claims too much – the authors did not demonstrate "autonomous holdase activity", they only show that in vitro C12 binds TH and prevents its aggregation. "Holdase activity" is typically used to refer to the in vivo ability of JDP to carry out a biological function without Hsp70. In my opinion all references to "holdase activity" of C12 should be moved from the Results to Discussion and presented as a hypothesis.

Lines 489-502 - this long added fragment is very confusing – in fact it simply states that C12 + TH synergistically stimulate ATPase of partner Hsp70 as expected for any JDP that delivers substrate for Hsp70. Whereas no synergistic stimulation was observed for JDPs (A2/B1) that do not bind TH. The text should be both shortened and simplified. Moreover Apg2, a nucleotide exchange factor, was not introduced in the text(?).

Lines 504-519 – this fragment considers possible autoinhibitory mechanism of C12 through analogies to the autoinhibition observed for class B JDPs e.g. B2. The explanation of the potential C12 autoinhibition is unclear and considering that the obtained results are negative this fragment should be much shorter.

Discussion

Lines 568-625 this segment should be shortened, also it contains too many references to the Results.

Lines 627-654 – as above

Lines 656 – 723 – this segments contains too many references to the Results and should be much shorter. Also very long description of the “alternative explanations for the substrate-associated release of the autoinhibition...” seems to be out of place as no evidence for autoinhibition was provided by the authors (see also line 707).

Reviewer #3

(Remarks to the Author)

The authors have appropriately addressed my remarks, and I have no further comments or requests.

Reviewer #4

(Remarks to the Author)

The authors have addressed all of my comments and the much improved manuscript is ready for publication.

Reviewer #5

(Remarks to the Author)

In their revised manuscript the authors make substantial changes and address major concerns of the reviewers. In particular, they provide more thorough analysis of the crosslink data, localizing DNAJC12 J domain and CTD at the TH regulatory domain and establishing that JC12 stabilizes TH, protecting it from aggregation. For the cryo-EM structural work they expanded their data set of TH:DNAJC12 complex in order to improve the resolution and better define the interaction of JC12 with TH and have included additional validation data. While the overall resolution improved slightly (5.6 to 4.25ang), the resolution of the proposed JC12 density and the TH RD did not substantially improve and remains around 12-15ang. As mentioned previously, this resolution is not sufficient to convincingly dock the relatively small J-domain and CTD helix – density at resolutions of <5 angstroms would be needed. Substantial unaccounted density remains in the map at the threshold shown, primarily in the adjacent connecting densities between the TH RD and central catalytic tetramer. Additionally, views of the map-model do not convincingly demonstrate that the JC12 domains fully dock into the map (an enlarged, rotated view of the map-model for the JC12-docked region with a less transparent map is needed). The fact that the authors state in responses to reviewer 4 that: “the cryoEM maps of the TH:DNAJC12 complex clearly exhibited a mass corresponding to the J-domain (JD) interacting with TH through one the RDs in the dimer, and the CTD interacting with the other RD” is surprising. This does not appear to be true given what is shown. The additional mass appears much larger than that of the JC12 domains. While the difference maps shown in supplemental are an important addition and demonstrate that JC12 binds in the RD region of TH, the volume of the difference appears 2-3x larger than the JC12 domains. How was the difference map generated? This was not discussed. It is critical that the two maps are filtered to the same, lower resolution and that the maps are scaled appropriately. Nonetheless, the difference maps do show general localization of JC12 to the TH RD region. Given the low resolution of the regions of interest, a more convincing demonstration of JC12 interaction would be to use their crosslinking data to generate a model independently using HADDOCK or other software that uses MS crosslink data to guide docking. The top-scoring models would then be compared to the proposed cryo-EM model, thereby providing an orthogonal approach for validation. The crosslink data on its own does not fully validate their proposed model given the large crosslink distances and the multiple orientations of JC12 that can be docked into the map.

Version 2:

Reviewer comments:

Reviewer #2

(Remarks to the Author)

The revised version of the manuscript has addressed my concerns regarding the organization of the text that I had raised in my previous review. The revised manuscript is improved both in terms of the clarity and logic of presentation. In my opinion, the manuscript is now acceptable for publication.

Reviewer #5

(Remarks to the Author)

The authors have addressed all my concerns in their revised manuscript. Their additional modeling using HADDOCK and difference map analysis accounting for local resolution provide helpful validations. I support the publication of this manuscript.

Point by point responses

Reviewer #1 (Remarks to the Author):

The manuscript by Tai et al. describes the structural characterization of the interaction of tyrosine hydroxylase (TH) with its chaperone DNAJC12, using biochemical, molecular biology and biophysical methods, cryoelectron microscopy and in silico approaches, all state-of-the-art techniques that strongly support the main conclusions of the work; namely, the unravelling of the binding stoichiometry of DNAJC12 with TH, the key residues involved and the functional consequences, TH stabilization without perturbing its interaction with substrate and regulatory molecules (dopamine). Overall, it is a rigorous study that describes the conformational properties of the TH-DNAJC12 complex and provides an explanation of the pathogenic mechanism of the most frequent DNAJC12 variant, that results in a truncated protein lacking the last 24 carboxyl-terminal residues.

DNAJC12 and TH variants are responsible for parkinsonism and related phenotypes. The authors have provided novel results that represent significant advances in the field, as they have now characterized the structure of the full-length DNAJC12 and identify a C-terminal octapeptide as essential for binding TH, thus stabilizing it and preventing its aggregation. These results are of value in the classification of DNAJC12 variants which are increasingly being identified in patients with hyperphenylalaninemia, dystonia and a diverse range of intellectual disabilities. Although not an expert in structural biology nor in some of the methodologies used, it is clear to me that conclusive experimental evidence is provided, the methods employed are described in sufficient detail, and the data are adequately interpreted. Of note, the manuscript is very well written and easy to follow with each subsequent experimental step adequately explained and justified.

I have some comments about the relevance of the structural analysis in disease and issues that need clarifying, as described below:

- 1) DNAJC12 variants generated by deletion mutagenesis: I assume all of them are produced fused to MBP and 6xHis; are they all expressed at similar levels? Are the large differences in the yield of purified proteins due to difference in stability once MBP and His tags are removed?

>> We appreciate this comment from the reviewer, pointing to issues that required further clarification. All DNAJC12 variants are indeed purified using the same 6xHisMBP tagged pETMBP1a/DNAJC12 plasmid with tags, and by similar expression and purification protocols as for full-length DNAJC12 (Included in Materials and Methods, sections "Peptides, plasmids and cloning" and Expression and purification of proteins"). The yields for the purified wildtype (WT) and for the truncated DNAJC12 variants for which most significant differences were found are specifically presented in the revised version. Certainly, the differences in yield provide insights on the stability of the proteins, e.g. the largely disordered truncated form DNAJC12(108-198), corresponding to the C-terminal half of the protein, presents a lower yield (~0.6 mg/L culture) than full-length DNAJC12 (~3.5 mg/L culture). On the other hand, the N-terminal half DNAJC12(1-107), containing the structured JD, presents a much higher yield (~47 mg/L culture). These values are included in the Results section (pages 5 and 6).

The use of identical vectors and conditions for expression of wildtype (WT), truncated forms and other variants of DNAJC12 (presented in Supplementary Fig. 5) with tags such as the 6xHisMBP that increases the stability of the expressed proteins largely supports that the expression efficiency is similar for full-length and variants. However, the unstable variant DNAJC12(108-198) showed a reduction of its yield compared to WT already as fusion protein, and a higher decrease in relative yield was observed for the isolated, purified protein. Aggregation of any of the expressed proteins was not perceived in the bacterial lysates, indicating that the decreased yield is most probably related to intracellular proteolytic degradation of the expressed protein in *E. coli*, as indicated in page 6 (from line 198).

- 2) While the mechanism of two common DNAJC12 variants is discussed, nothing is mentioned about TH variants. It would be interesting to know if there are TH pathogenic variants in the regions where DNAJC12 monomers interact/are bound. Can TH deficiency be caused by a deficient interaction with a wild-type DNAJC12?. This information would add to the relevance of the work.

>> We thank the reviewer for this comment, that has prompted the expression, purification and characterization of tetrameric TH-p.R202H, one of the most frequent variants associated with TH deficiency (THD), and the results are presented in a new section in page 14 (“The THD-associated variant TH-R202H is also stabilized by DNAJC12”). This variant has previously shown reduced stability and increased susceptibility to proteolysis compared with TH-WT¹. DNAJC12 binds to this variant with a similar stoichiometry (4 TH subunits: 2 DNAJC12 subunits) (Fig. 8b) and affinity (Supplementary Fig. 7 and Table 1) than to TH-WT, and the interaction with DNAJC12 also results in thermal stabilization (Fig. 8c). TH-p.R202H presents a higher aggregation propensity than TH-WT but is equally protected from aggregation by DNAJC12 (holdase effect) (Fig. 8d). The stabilizing effect exerted by DNAJC12 may thus be of greater relevance for the THD variant than for TH-WT, indicating that the pathogenic effect of the mutation in patients might be dampened by its interaction with DNAJC12. The functional ATPase assays with the complex TH-R202H:DNAJC12, showed an activation of the ATPase activity of Hsc70 (Fig. 8e), also very similar to that effected by TH-WT.

Regarding the question of whether THD might be caused by TH variants that present a defective interaction with WT DNAJC12, in the case of TH-p.R202H, with the mutation localized in the catalytic domain (CD) (Fig. 8a), the structural model of the TH:DNAJC12 complex indicates that this variant will most probably not affect the interaction with DNAJC12, in agreement with the biochemical data. For other THD-associated variants in general (716 deposited in the publicly available PNDdb (<http://www.biopku.org>); accessed 17 June, 2024), 220 (30%) are missense variants, which are spread throughout the entire TH sequence. To our knowledge, expression studies have not been performed for variants in the RD (residues 1-160). It seems nevertheless interesting that the TH region corresponding to Helix 2 of the ACT domain in the RD (residues 138-150), comprising the main area of interaction with DNAJC12 (Fig. 6b,c), shows a high recurrence of registered missense variants (A142S, A143T, L145H, V148A, R149C and R149L), of which the last four present a high pathogenicity index (Combined Annotation-Dependent Depletion (CADD) score > 20) (www.biopku.org/PNDdb). Thus, variants at these residues in Helix 2 of TH may be expected to affect the interaction with DNAJC12, which seems to be in accordance with the finding that disease-related mutations are generally enriched in protein-protein interactions (PPIs)². Nevertheless, these variants may also directly affect the structure and stability of TH itself, as aggregation of the expressed proteins was observed upon recombinant expression of several of the site-directed mutants in this region (L141R, L144A, and L145R), prepared to support the structural model of the complex (See answer to comment 4, reviewer #4). In any case it is important to notice that in general, all THD-associated variants, but specifically those in the RD, could conformationally affect the interaction areas around Helix 2.

Finally, one notion that arises from the modeled structure of TH:DNAJC12 is that through stabilization of the dimerized RDs, DNAJC12 binding would result in stabilization of the TH tetramer (Fig. 6b), both for WT and potentially for several THD-associated variants. We have added comments on the relevance of this effect on THD-variants in the Discussion (section “Disease associated variants”).

- 3) Figure 4a, there are red and orange asterisks, not black, as mentioned in the figure legend and the text

- 4) Reference (17), please update

>> We thank the reviewer for these remarks.

Black asterisks have been changed to orange in Fig. 4A legend and text (page 9, second paragraph), and the reference (21 in the revised version) has been updated (Gallego et al. (2020) Hum Mutat 41(7): p.1329-1338).

Reviewer #2 (Remarks to the Author):

J-domain proteins (JDP) are obligatory cochaperones of Hsp70s that are involved in all aspects of protein homeostasis; protecting proteins against aggregation and assisting their folding, transporting them across membranes and refolding them following their aggregation upon stress, and if that fails, directing them for proteolytic degradation. These general functions are usually carried out by systems involving class A and B JDPs and their Hsp70 partners, while JDPs of class C **are typically more specialized**. For example, they function in uncoating of clathrin vesicles, biogenesis of histones, iron-sulfur cluster biogenesis, and pre-mRNA splicing. **But despite the functional diversity of class A, B and C JDPs, a single biochemical mechanism, that is cyclic interaction with substrate polypeptides, drives functionality of Hsp70 systems.**

The reviewed paper focuses on the class C JDP termed DNAJC12, whose mutational variants are implicated in a neurometabolic disorder that manifests as hyperphenylalaninemia, accompanied by dopamine and serotonin depletion. Previous works involving co-immunoprecipitation in human cell and murine liver lysates revealed that DNAJC12 binds to tyrosine hydroxylase (TH), the enzyme of dopamine synthesis pathway, and that defective interaction causes disease. The authors used purified DNAJC12 to characterize its interaction with TH both biochemically and structurally. They demonstrated that two monomers of DNAJC12 bind to a TH tetramer and using truncated variants of DNAJC12 showed that this interaction involved eight conserved C-terminal residues of DNAJC12. The authors also find that DNAJC12 binding delays time-dependent aggregation of TH while maintaining enzymatic activity and inhibition by dopamine. The authors used a combination of cryoelectronic microscopy and chemical crosslinking to obtain a structural model of the DNAJC12-TH complex. The model reveals that C-terminal segments of DNAJC12 interact with a dimer of regulatory domains of TH such that the active sites are available for substrate and dopamine interaction. In the DNAJC12-TH complex, the J-domain of DNAJC12 is free to interact with its Hsp70 partner, as demonstrated by stimulation of the ATPase activity of Hsc70 by DNAJC12-TH complex. Interestingly, no such stimulation is observed for DNAJC12 alone.

The results presented in this paper significantly expand our knowledge about DNAJC12 and contribute significantly to our understanding of the pathology associated with its malfunction. The results are noteworthy and supported by data, which were correctly analyzed. The methodology used is sound and meets the standards in the field of chaperone research. The manuscript is written very clearly and contains enough information for the work to be reproduced. Therefore, in my opinion, this paper is appropriate for publication in Nature Communication after the issues listed below are resolved- minor revision.

Important issues:

1. (line 126) The C-terminal, unstructured, domain of DNAJC12 contains evolutionary conserved alpha-helical fragment, which the authors termed "helix V". This name could be confusing, because the same name is used for the conserved alpha-helical fragment present in the G/F region of DNAJB1 and other B class JDPs [Faust, O. et al. HSP40 proteins use class-specific regulation to drive HSP70 functional diversity. Nature 587, 489–494 (2020)]. The helix V of B class JDPs is involved in J-domain autoinhibition, which plays an important role in its disaggregation activity. Because there is no evidence that helix V of DNAJC12 and the unstructured region of this protein are homologous to helix V and the G/F region of class B JDPs respectively, to avoid any confusion, this C-terminal alpha-helix should have a different name.

>>> We appreciate this comment and have changed the name of the previous helix V to "linker-helix", as it is located at the start of the long and apparently flexible linker that connects the N-terminal

domain (NTD; including the J-domain) and the C-terminal domain (CTD), and the name has been changed throughout the paper (text and figures).

2. (line 443) The observation that the DNAJC12-TH complex stimulates the ATPase activity of Hsc70 more efficiently than the DNAJC12 alone could be interpreted in at least two alternative ways: (i) the release of J-domain autoinhibition, as proposed by the authors and/or (ii) synergistic stimulation of the ATPase by the J-domain of DNAJC12 and substrate TH. The latter scenario could be verified experimentally by using a Hsc70 substitution variant defective in substrate binding – substitution of a key residue in the substrate binding cleft of Hsc70. However, considering that the manuscript already contains many important results, I do not insist that the control experiment described above must be done, but an alternative interpretation of the obtained results should be added to the text with explanation how it could be tested experimentally.

>> We agree with the reviewer on the two alternative interpretations to explain that the TH:DNAJC12 complex stimulates the ATPase activity of Hsc70 much more efficiently than DNAJC12 alone. We are also very thankful that he/she understands the resources and time efforts required to obtain structural and functional understanding of the mechanisms involved. We have in any case considered the two alternatives:

Alternative i) The formation of the TH:DNAJC12 complex releases the JD autoinhibition in DNAJC12. As presented in the first version and in the revised manuscript, we have prepared several truncated forms of DNAJC12, eliminating sequences that could cover the HPD motif and other essential determinants for Hsc70 binding, or hinder the interaction by other mechanisms. Neither deletion of the N-terminal sequence up to Helix 1, nor truncation of sequences C-terminal of the JD, excluding also the linker-helix, resulted in DNAJC12-mutant forms that directly activated the Hsc70 ATPase activity to levels characteristic of the canonical JDPs, such as A2 and B1 (Supplementary Figs. 14b). We have now conducted site-directed mutations at residues within Helix IV of DNAJC12, specifically W78A, R79G, M83A and M85A, attempting the release of potential autoinhibitory interactions that could hinder the activating interaction of JD of DNAJC12 with Hsc70. However, none of these DNAJC12 variants resulted in increased activation of Hsc70 when compared with DNAJC12-WT. We have also considered other possible mechanisms that could explain an unfavorable interaction of DNAJC12 with Hsc70 in the absence of TH. It has recently been shown that the charge distribution in the side of DNAJC12 JD that would interact with the interface between the nucleotide binding domain (NBD) and the substrate binding domain (SBD) of Hsc70 is very different to that of the other members in the JDP family (see Fig. 4 in ³). This finding may suggest that binding of TH to DNAJC12 might lead to readjustments of the surface electrostatic potential or to the elimination of possible steric hindrances, resulting in a favorable interaction of DNAJC12 with Hsc70, and plausibly also of the complexed TH towards the β -subdomain of the SBD (SBD β) of Hsc70 (see also *Alternative ii*, below). Proving this possible mechanism and/or regions involved in the “autoinhibition” of DNAJC12 will require a detailed preparation of additional variant proteins, development of assays and structural studies also including Hsc70, which are outside the scope of this work but that will be build up as a follow up study based on the results in the present paper.

Alternative ii) There occurs a synergistic stimulation of the ATPase by the JD of DNAJC12 and the substrate TH, when each binds to their corresponding binding sites in Hsc70. In order to probe this alternative, we followed the suggestion of the reviewer and measured the ATPase activity of a Hsc70 variant defective in substrate binding (i.e. Hsc70-V438F) and compared it with WT-Hsc70. This mutation strongly reduces the affinity of Hsc70 mutant for substrates ⁴ and is based in the *E. coli* DnaK V436F mutant, which presents a deficient ATP stimulation by substrate and DnaJ ⁵. The new results, presented in Supplementary Fig. 14a, show that this mutation totally eliminates the stimulation of Hsc70 ATPase activity exerted by the TH:DNAJC12 complex, indicating both that there is a need for

simultaneous binding of the JD of DNAJC12 and TH to their respective sites at Hsc70, and that TH is recognized as a substrate.

We believe that our results support a conformational autoinhibition in DNAJC12 -when alone - for optimal interaction with Hsc70, not necessarily due to a blocking of the JD by other sequences, as in the case of Helix V for class B JDPs ⁶, but may be due to adverse distribution of charges and/or steric hindrances in general, which might become favorable in the TH:DNAJC12 complex. The complex may also provide a correct binding of each component in their respective sites, i.e. DNAJC12 JD at the NBD-SBD interface and TH at SBD β , leading to a synergistic interaction of Hsc70, indicating that a combination of both alternatives i) and ii) may be possible. Indeed, this is a rather typical behavior for JDPs with specific substrates, such as seen for the synergy between auxilin and clathrin baskets, see Fig. 3 in ⁷, Jac1 (DNAJC20) and Isu1 (Fe-S cluster scaffold), see Fig. 2e in ⁸, and DnaJ and σ^{32} , see Fig 3c in ⁹. See also our answers to comments 1 and 14 to reviewer #4.

We have incorporated these new arguments in the Results (from start of page 13) and in the Discussion (section “The interaction of the TH:DNAJC12 with Hsc70”), with indication of the variants prepared for binding and Hsc70 ATPase activity. Specifically, the result for the variant Hsc70-V438F is presented in Supplementary Fig. 14a.

Minor issues:

3. (Fig. 1) I would like to propose the following changes: add the structural model of DNAJC12 and map disorder and hydro-phaticity on this model-move panel b and c to the supplement, make the amino acid sequence of DNAJC12 smaller but such that identification of individual residues will be simpler e.g., by adding residue number scale above the sequence.

>> We appreciate these suggestions. We have improved the presentation of the sequence of DNAJC12 in Figure 1, panel a, as indicated, and have included a more complete presentation of the AlphaFold structure (panel b), in which we have also mapped the disorder (panel c) and hydrophaticity profiles (panel d), whereas the previous figures with the disorders and hydrophaticity prediction profiles are moved to the supplementary information. We hope that these changes are satisfactory.

3b. (Fig. 2) A: SDS-PAGE elution profiles for TH and TH-DNAJC12 complex should be added to the supplementary information

E, F: traces of BLI measurements should be added to supplementary information.

>> The SDS-PAGE elution profiles and the traces for all BLI experiments presented are included in new Supplementary Figs. 6 and 7, respectively.

4. (Fig. 3C) – concentration of TH in the reaction mixture should be indicated either in Fig. 3C or in the legend. Why different concentration scale for DNAJC12 (nM) and for DA (microM) – microM are in the main text for both DNAJC12 and DA (line 264-5)

>> The concentration of TH in the assays (0.4 nM tetramer) has been included in the figure legend (Fig. 3e in the revision).

>> The concentrations of both DNAJC12 and DA in Figs 3,e and f, respectively, are now given in nM.

5. Fig. 4 catalytic (CD) and oligomerization (OD) domains should be indicated in Fig. 4B

>> The CD and OD are now represented in light blue and green in Figs. 4. and 5, respectively, as indicated in the Fig. legends.

6. (Line 293) there is no black asterisk in Fig. 4a (red and orange)

>> This has been corrected: black has been changed to orange in Fig. 4a.

7. (Line 441) ...strong activation of Hsc70 ATPase... The two-fold activation of Hsc70 ATPase is “measurable” and expected – but I would not call it “strong” - as 20-30 fold, or even higher, activation is typically observed in the presence of JDP/substrate/nucleotide exchange factor

>> The referee is right and “strong” has been deleted.

8. Fig. 6a – traces of BLI measurements should be added 514 to the supplement

>> The traces for all BLI experiments presented are included in new Supplementary Fig. 7. See also a more detailed answer to comment 3 from reviewer #4.

9. Fig. 7d – why the y-axis is broken (has a gap) - it is confusing, while the amount of the saved space is negligible – the same true for Suppl. Fig. 14.

>> The gaps in both figures have been eliminated.

10. Suppl. Fig. 3 the helix V (alpha-5) shown on top of the multiple sequence alignment is predicted only for JC12 or for all sequences in the alignment? Please add this information to the legend.

>> We appreciate this comment and have clarified the information on the secondary structural elements in the Figure legends of Supplementary Figs. 2 (Sequence alignment of DNAJC12 from different species) and 3 (Sequence alignment of the J-domain from different human JDPs). We have changed “Helix V” to “linker-helix”, which is only labelled in Supplementary Fig. 2, since the DNAJC12 proteins shown in the alignment present high sequence homology in this helix.

Moreover, this information has been added in the Figure legends of Supplementary Figs. 2 and 3: “The secondary structural elements for the JD shown at the top of the alignment are extracted by ESPript from the Alpha-fold structure of the full-length human DNAJC12 (AF-Q9UKB3-F1)”.

11. Suppl. Fig. 10 The structural model presented here predicts that TH is not a substrate for the Hsp70. If this is indeed the case it should be stated in the text and discuss. (line 579) suggests that the authors consider TH as a substrate/client for Hsc70? (lines 628-631) also suggests that TH is a substrate for Hsc70?

>> Specifically, in the sentences pointed out by the reviewer (line 579 and 628-631 in previous version), we assumed that TH was a substrate of Hsc70. We understand however that this subject as well as the question on the interaction of the complex TH:DNAJC12 with Hsc70 (see comment 2, above) were not properly investigated or thoroughly discussed in the original manuscript as the initial focus of the paper was the characterization of DNAJC12 alone and the formation of a complex with TH, notably associated with the holdase activity of this relatively unknown JDP. The functional characterizations included the assays of reactivation and refolding of luciferase, and delay of Tau aggregation, activities that were not performed by DNAJC12, whereas the other two well studied JDPs, i.e. DNAJA2 and DNAJB1, which belong to the “canonical” classes A and B, perform all these functions (Fig. 7a-c). DNAJA2 and DNAJB1 also provided a more effective stimulation of ATPase activity of Hsc70 than DNAJC12, but this activity was not additionally stimulated by TH. Indeed, we interpreted the finding that TH and DNAJC12 could synergistically stimulate the Hsc70-ATPase activity as an indication that the interaction of the TH:DNAJC12 complex with Hsc70 might ensure the proper binding of TH as a substrate by Hsc70. The present new data showing that DNAJA2 and DNAJB1 do not bind TH and failed to stimulate Hsc70 in the presence of TH (Supplementary Fig. 7o,p; and Fig. 7d), as well as the result with the Hsc70-V438F variant, which abrogates the Hsc70-ATPase stimulation by the TH:DNAJC12 complex (see answer to

comment 2 above and Fig. 7d) further support that DNAJC12 specifically present TH as a substrate to Hsc70.

The structural model presented in original Supplementary Fig. 10 (Supp. Fig. 12 in the revision), based on the modelled structure of TH:DNAJC12 docked in the DnaK:DnaJ complex in the ATP-bound conformation (PDB 5NRO)⁹, aimed to illustrate the orientation of the JD and absence of clashes, but did not properly show the orientation of the dimeric RDs of TH and the C-terminal of DNAJC12 towards the substrate binding domain (SBD) of DnaK/Hsc70. We have now improved the previous figure to also show that the RD subunits of TH and the C-terminal of DNAJC12 in the TH:DNAJC12 complex are very close to the SBD region of DnaK (Supplementary Fig. 12 and Supplementary Movie 1). Importantly, the model does not account for any structural changes that may occur following the binding of the DNAJC12:TH complex to Hsc70, which could potentially expose Hsc70-interacting regions in TH.

In addition to these new results and Figures, we have also improved the arguments supporting that TH is a substrate/client of Hsc70 in the Discussion (Section “The interaction of the TH:DNAJC12 with Hsc70”). In this regard, it has previously been shown by Parra et al. that Hsc70 interacts with TH and regulates its enzyme activity, since overexpression of Hsc70 in dopaminergic MN9D cells leads to increased TH whereas knockdown of Hsc70 resulted in decreased TH activity and dopamine levels¹⁰. Using co-immunoprecipitation assays in brain tissue and cells, as well as pulldown assays to delineate the domains of Hsc70 involved in the interaction, these authors showed the strong interaction of TH with the C-terminal region of Hsc70 (residues 540–650), and to a lesser degree with residues 373–540. These results support the putative interactions that could be expected from a binding of TH close to/at the SBD in the ADP-bound state of the complex, as inferred from the ADP-bound states of the SBD of DnaK (PDB 1DKX)¹¹. See also answer to comment 1 from reviewer #4). The results by Parra et al. largely proof that TH is a substrate/client of Hsc70, most probably in a DNAJC12-mediated manner, although the specific role of DNAJC12 as a co-chaperone for TH was not known at the time (note that TH alone does not activate Hsc70 in vitro). Moreover, we have applied the algorithms Limbo¹² and ChaperISM¹³ to predict Hsp/Hsc70-binding regions in TH, resulting in the identification of residues 82–88 and 129–135 as binding regions in the RD of TH (labelled in Fig. 6c; Supplementary Movie 1). Interestingly, the sequence of residues 82–88 was already predicted by TANGO¹⁴ as a region of high aggregation propensity in TH (labelled in Fig. 6c). We have also labelled the location of these sequences, near the SBD in Hsc70, in the model of the ternary TH:DNAJC12:Hsc70 complex (Supplementary Fig. 12; Supplementary Movie 1).

See also answer to question 2 from reviewer #4.

12. Suppl. Fig. 13a – add traces of the BLI measurements

>> The traces for all BLI experiments presented are included in new Supplementary Fig. 7. See also a more detailed answer to comment 3 from reviewer #4.

Reviewer #3 (Remarks to the Author):

In their study Tai et al. investigate the role of the heat shock Hsp40/ J- domain protein DNAJC12 in the structural recognition and stabilization of tyrosine hydroxylase (TH). Using recombinantly expressed and purified protein variants the authors show that DNAJC12 exists as a monomer in-vitro and binds TH in the stoichiometry of 2 DNAJC12 molecules per TH tetramer. By generating DNAJC12 mutants the authors then go on to show that a conserved C-terminal region containing the RKFRNYEI peptide within DNAJC12 mediates binding to TH. This region is missing in the pathogenic DNAJC12 mutant Trp175Ter

and the mutant is therefore unable to bind TH, as shown by the authors. The authors also obtain a relatively low-resolution structure of the TH:DNAJC12 complex by cryoEM and use it in combination with cross-linking mass spectrometry (XL-MS) to locate DNAJC12 and to demonstrate that DNAJC12 interacts with TH via its regulatory domains, thereby leaving the active sites of TH accessible and explaining why DNAJC12 binding does not impede TH activity.

Overall, this is an interesting and well-designed study that employs biochemistry, a large number of biophysical assays, and structural biology to show how the chaperone DNAJC12 interacts with TH, thereby clarifying on a molecular level its important role in stabilizing TH while preserving its activity, at least in-vitro. The manuscript itself is well written and the conclusions are supported by the data.

However, some points regarding the structural and in the particular the XL-MS data, are unclear in the current form of the manuscript and require at least clarification.

In the methods the authors write that the XL-MS data was acquired on a triple TOF mass spectrometer using BS3 as a cross-linker and MeroX 2.0 as a search engine, using a FDR of 5 % and an XlinkX score for validation.

This seems inconsistent.

BS3 is a non-cleavable cross-linker but MeroX is only capable of searching data of cleavable cross-linkers. How was this possible?

>> Originally, MeroX was only capable of analyzing data obtained with cleavable crosslinker, being StavroX the alternative of the same developers to identify peptides crosslinked with non-cleavable crosslinkers. However, since version 2.0, the one used in this study, both tools were merged and now MeroX is able to handle XL-MS data of both cleavable and non-cleavable crosslinkers (for further details see <https://stavrox.com/Download.htm>).

Also, XlinkX is an alternative cross-linking search engine embedded within Proteome Discoverer from the Heck & Scheltema labs. How could it be used to validate data searched by MeroX?

>> Following the reviewer's suggestion, we have reanalyzed all the XL-MS datasets (interprotein crosslinks) with XlinkX in Proteome Discoverer 2.5. The outputs from both search engines showed a strong correlation (See Supplementary Table 3; crosslinks displayed in green), indicating that the crosslinks described in this work were identified with high confidence. In view of the overall high quality of the crosslinks obtained, we decided to include not only the common crosslinks identified by both search engines, but also those identified uniquely by MeroX and XlinkX.

And more generally. How exactly did the authors use a triple TOF MS to fragment and identify cross-linking data. As this is a rather unconventional approach in cross-linking MS, the authors should please expand on how this was done.

>> The methods section has been modified to better describe the actual XL-MS workflow employed in this study. MS data were indeed acquired either in an Orbitrap Exploris 240 or in Orbitrap Fusion Lumos. The protocol described in the submitted version of our manuscript was erroneously included and corresponds to a previous analysis workflow used in another project, where the LC-MS/MS analysis was performed in a Q-TOF mass spectrometer.

The authors state that they have used a custom-made database to search the data -and presumably to calculate the FDR. How was this done exactly and on which level (protein, peptide..) was the FDR calculated?

>> MeroX calculates the FDR at the peptide-spectrum match (PSM) level by performing an MS/MS ions search against a concatenated target-decoy database ¹⁵. In this study, the decoy database was generated by reversing the sequences of the target database. Of note, the results from MeroX are very

similar to those obtained with XlinkX (ThermoFisher) that estimates the FDR in a similar way. The application of XlinkX software has been included in the Methods section (27) and the comparison with MeroX in the Results section as Supplementary Table 3.

Minor points:

Regarding obtained cross-linking data itself. The XL-MS data seems to indeed agree well with the obtained cryo-EM maps, as stated by the authors. The authors do however use quantitative statements on the cross-linking data as “the region with the highest number of cross-links was the CTD” (page 9) for their argumentation. This should be done with caution or only after additional analyses as the authors do not show the overall distribution of cross-linkable lysines within TH or DNAJC12, nor the number of identified mono or intra-links. Without this information it is hard to judge which cross-links could have possibly formed in the first case and thus, if the amount of identified inter protein cross-links is indeed comparatively high. Moreover, Fig. 5 shows that many of these cross-links emanate from the same lysine (i.e. lysine K183 in Fig. 5b), indicating that this lysine is particularly reactive and thus again putting such quantitative statements at least somewhat in perspective.

>> The reviewer is right, and we have modified the statement for the sake of clarity. It has been rephrased to avoid quantitative interpretation to “part of the CTD, in particular residues 181-191, showed a large number of intermolecular cross-links” (Page 10). Notably, in the specific case of the TH:DNAJC12 complex, the inter protein cross-links constitute a remarkable 50 % of the overall cross-links obtained including mono and intra-links. As rightly noted by the reviewer, K183 exhibits exceptional reactivity in all XL-MS analyses conducted, emerging as the most frequently observed interprotein cross-links obtained for the CTD of DNAJC12. In response to this observation, we have made adjustments to the main text (start of Page 11) to underscore the significance of this particular lysine in our findings.

Figure 5 also shows that some cross-links bridge seemingly rather large distances (see for example link between K150 and K240). As conventional BS3 should only bridge distances of up to 25 Å, this may indicate a certain amount of false positive data within the dataset. Likewise, the authors allow very short peptides with 3 or 4 amino acid length as input (see Supplementary Table 3). Such cross-links tend to be ambiguous, at least in more complex data. Maybe the authors could also comment on this.

Although we are aware of the constraint distances allowed by BS3, we made a deliberate decision to incorporate all the obtained interprotein crosslinks that involve structured residues in Figure 5. Our idea was to enhance the comprehension of the structural model, explicitly showing that the observed crosslinks belonging to the central highly-flexible region of DNAJC12 could suggest a structural rearrangement of this region. We acknowledge that this can be confusing, and we have now removed the modelled central region of DNAJC12 from the figure. With respect to maximum distances, we have extended the permissible range to more than 30 Å as previously described for other dynamic complexes^{16, 17}, because the known structures chosen to build the structural model do not necessarily match the exact conformation adopted in this complex. This methodologic criterion is presented in the Materials and Methods section (Page 27) and the references have been included in the revised version.

To improve clarity, we have modified Figures 4, 5 and 6b where we excluded not only the central highly-flexible region of DNAJC12 but also the linker connecting the RDs and CDs of TH, and the 38-residue highly unstructured N-terminus of TH. These exclusions were made because these flexible regions clearly deviate from the constraints imposed by BS3. In the light of the established maximum and minimum distances allowed by BS3 falling within the range of 11 to 25 Å¹⁸, it is pertinent to address the crosslinks identified in our study involving very short peptides (3-4 residues). These crosslinks can be deemed robust, considering that the distance between C α atoms of sequential amino acids is conventionally fixed at 3.8 Å, allowing for the accommodation of four contiguous amino acids within

the BS3 length constraints. Furthermore, and as it was stated above, the known structures chosen to build the structural model do not necessarily match the exact conformation adopted in this complex generating slight differences in the allowed maximum and minimum distances.

Reviewer #4 (Remarks to the Author):

In this manuscript by Tai et al, the authors describe in an elegant series of biochemical and structural experiments to explain how a molecular chaperone from the J-domain protein family, DnaJC12, stabilizes Tyrosine Hydroxylase (TH) and how mutations in the chaperone interfere with this function and likely lead to disease. Overall, the study is well done with a lot validation but there are some missing pieces and inconsistencies that need to be resolved. We recommend the following revisions, after which the manuscript may be suitable for publication in Nat Comm.

Summary

1. Variants of DnaJC12 cause neurometabolic disorders
2. Variants of tyrosine hydroxylase which converts L-tyrosine to L-dopa cause THD
3. DnaJC12 purified as a monomer; Unstructured CTD, binds with high affinity with TH
4. TH:DnaJC12 complex maintained TH activity
5. The C-terminal region of DnaJC12 is essential for binding to TH

Comments

1. N-terminal truncation of DnaJC12 (missing the J-domain) shows no binding to the TH. In the structures the J-domain appears to be important for the interaction with TH. How is it accessible for Hsp70 recruitment and stimulation of ATPase activity? Perhaps the inability of the J-domain alone binding to TH is more consistent with the above idea of accessibility to Hsp70 but then is the structure is not reporting on the correct conformation? This disconnect needs to be clarified in the manuscript.

>> The C-terminal truncated forms of DNAC12 (missing the C-terminal domain (CTD)), and all constructs missing the C-terminal sequence, residues 191-198, show a very reduced binding affinity for TH (Fig. 2c,d; Supplementary Fig. 7; Table 1). In any case as the reviewer points out the cryoEM maps of the TH:DNAJC12 complex clearly exhibited a mass corresponding to the J-domain (JD) interacting with TH through one the RDs in the dimer, and the CTD interacting with the other RD. Accordingly, the modelled structure of the complex based on cryoEM and crosslinking-MS, shows the JD interacting with Helix 2 of the RD (Fig. 6c). It is possible that the initial interaction of DNAJC12 with TH through the CTD may increase the avidity of the J-domain, by tethering this region despite its low affinity for TH when truncated from the rest of the protein (Table 1). At the same time, there is a subpopulation of particles of TH:DNAJC12 complex (about 30% of the total population) where only the smaller mass corresponding to the CTD is observed, suggesting that the DNAJC12 N-terminus undergoes dynamic fluctuations, in agreement with its low affinity interaction with TH compared with the CTD (This is discussed at the end of section "DNAJC12 interacts with TH by embracing its regulatory domains", page 9). Finally, a more stable JD-bound conformation may occur upon recruitment of Hsc70 to the TH:DNAJC12 complex.

With respect to the accessibility of the JD for interaction with Hsc70, our interpretation of the cryoEM structure of the TH:DNAJC12, validated by XL-MS (Fig. 5; Supplementary Tables 2 and 4), and site-directed mutagenesis, followed by the modelling of the ternary TH:DNAJC12:Hsc70 complex by docking the JD of the DNAJC12 onto the structure of the DnaK:DnaJ complex (PDB 5NRO)⁹ has led to an improved model (Supplementary Fig. 12 in the revision). In this model, the JD and its HPD motif are

accessible and in the right conformation to bind to Hsc70 at the SBD and NBD interface and activate Hsc70 ATPase activity (Supplementary Movie S1). At the same time the disposition of TH appears to be adequate to interact with the SBD β . This concerted binding of TH and DNAJC12 for activation of the ATPase activity of Hsc70 is supported by the result obtained with the Hsc70-V438F mutant (Supplementary Fig. 14a). See also answer to comments 2 and 1 from reviewer #2. Nevertheless, and as the reviewer rightly remarks, it is not clear why the isolated JD alone presents a very reduced binding affinity for Hsc70 (Fig. 2c,d; Supplementary Fig. 7; Table 1). Although the Hsc70-interacting HPD motif appears accessible in both the isolated JD and in the full-length DNAJC12 (Fig. 1b), it seems possible that the right conformation for recruitment of Hsc70 and stimulation of ATPase activity is only achieved upon coordinated binding of DNAJC12 and TH to their respective sites, as nicely shown in Kityk et al. for the complex of *E. coli* DnaJ and Hsp70 DnaK⁹. As our truncation analysis indicated that the autoinhibition mechanism does not resemble that observed for the class B JDPs⁶, the mechanism for DNAJC12 might be different. Please see arguments for the plausible alternatives in the answers to comments 2 and 11 from reviewer #2. Indeed, there are several possibilities such as, among other, i) the need for a synergistic and specific interaction of TH and DNAJC12 with Hsc70, which can only be achieved upon binding of the proper TH:DNAJC12 complex with the chaperone, and/or ii) a modification of the distribution of adverse charges and steric hindrances in DNAJC12 upon binding to TH, which could provide a correctly adjusted surface electrostatic potential and residue complementarity for proper interaction with Hsc70 (see new arguments in page 16 in the revision, section “The interaction interfaces between TH and DNAJC12”).

Please see also the answer to the next question.

2. What is the role of Hsp70 in this process? Does Hsp70 bind to the C-terminus of DnaJC12 to remove it from the DnaJC12:TH com

>> We understand that the questions posed by the reviewer are very relevant, and we appreciate the suggestion about performing cross-linking to stabilize the complexes and better understand the mechanisms of partner recognition, folding and release of the substrate. However, it is known that “Consistent with the well-established role of Hsp40s in stimulating the ATP hydrolysis rate of Hsp70s, Hsp40s interact with Hsp70s only in the ATP state. However, this interaction is so transient that up to now, it has not been possible to capture any stable Hsp70-Hsp40 complexes without chemical crosslinking or using protein fusion which raises concerns about associated artifacts. Moreover, as described above, the ATP-bound state of Hsp70s itself is short lived, especially when interacting with Hsp40s” (extracted from¹⁹). It thus appears that proper characterization of the mechanisms and structural determinants for interaction of DNAJC12 and TH with Hsc70, and the consecutive activation of the chaperone requires an exhaustive optimization of constructs, assays and protocols for stabilization of binary and tertiary protein complexes, a work which is outside the scope of this work but that will be build up as a follow up study based on the results in the present paper (see answer to comment 2, to reviewer #2 above). Nevertheless, related to the comment by the reviewer, in order to investigate a possible mechanism for the release of the C-terminal domain (CTD) of DNAJC12 from TH upon Hsc70 binding, we have focused on the C-terminal sequence EEVD of the Hsc70 chaperone, which has been shown to be important for the ATP-Hsp70 dependent folding and release of substrates through interaction with the CTDs of JDPs, notably DNAJBs, and other cochaperones^{20, 21, 22, 23}. In addition, the C-terminal sequence EEVD of Hsc70 has also been related to the release of the autoinhibition of the JD of DNAJB1⁶. To explore the question raised by the reviewer we used a truncation variant of Hsc70 lacking the 10 C-terminal residues, Hsc70- Δ C10²⁴, but we did not observe any effect on the ATPase activity stimulation exerted by the TH:JC12 complex compared with WT-Hsc70 (See Figure 1 in this letter, below). As previously reported⁶ the stimulation of the ATPase activity by DNAJB1 was indeed reduced when tested with this variant (Fig. 1, below). These results support that compared with this canonical JDP, other regions of Hsc70 and/or other mechanisms appear to be involved in the release of DNAJC12 and consequently of TH. Further studies focusing on the preparation

and characterization of the ternary TH:DNAJC12:Hsc70 complex (see answer to comment 2 from reviewer #2) are expected to provide insights on the mechanisms for substrate and DNAJC12 release.

Figure 1. Stimulation of ATPase activity of Hsc70 and Hsc70ΔC10 by DNAJB1 and DNAJC12+TH.

The ATPase activity of Hsc70ΔC10 in the presence of Apg2 and either DNAJB1 or DNAJC12 and TH, and was measured and compared to that of Hsc70 at the same conditions. The data represent the mean \pm SD for n=3 independent experiments. The ATPase activity was compared to the activity of Hsc70 at the same conditions and with the same JDP, by one-way ANOVA and Tukey's post-hoc test (****p<0.0001).

3. BLI data analysis. How was the data fit? 1:1 or 1 1:2 stoichiometry model? Please report kinetic traces, fits and kinetic parameters for the BLI binding measurements. Supplementary Table 1 reports the affinities in μ M which is likely incorrect given the analyte concentrations used in the BLI experiments.

>> We appreciate the remark on the units for the affinity, which are now expressed in nM. We have analyzed the data using the steady state response. The responses were obtained using the Octet® Data Analysis HT as an average of the last 10 s of the association phase, which in total is 600 seconds for each concentration. Traces for all concentrations are now shown for all BLI assays (Supplementary Fig. 7) which are performed for n=3 independent samples. Only one representative measurement is shown in each panel in this figure, for clarity. The K_D values, calculated from the fittings of the binding response curves from the three triplicates to the "Binding isotherm" equation (GraphPad Prism 10) as described in Materials and methods, are listed in Table 1. The fittings are presented for some of the instances in the paper (Figs. 2c,d; Fig. 6a; Supplementary Fig. 15a).

4. The binding data, SEC-MALS and structural biology of the complex is very nice evidence of high affinity. In the end, it is not clear how nM affinity is achieved for the DnaJC12:TH interaction. The authors present a structure of the complex, it seems reasonable to test targeted mutations rather than truncations – the details of the interaction can be inferred and directly tested experimentally. This would significantly add to the completeness of the story.

>> We understand the value of presenting a structural model based on cryoEM and crosslinking-MS showing the predicted interactions between TH and DNAJC12 but in the first version we were reluctant to provide structural details based on the low-resolution model obtained for the TH:DNAJC12 complex, even after the small increase in resolution of the structure included in this new version (4.25 Å; see first comment to Reviewer #5). However, the significant improvements in model validation, achieved through additional cryoEM and reanalysis of crosslinking-MS data, gave us the confidence to proceed with the site-directed mutagenesis as recommended by the reviewer.

The results with truncated forms have shown that the CTD of DNAJC12 (residues 176-198) constitutes the client binding region, where the last 8 residues seem to establish essential interactions (Supplementary Fig. 7; Table 1). Based on the obtained structural model of the TH:DNAJC12 complexes, and using PDBs 7A2G and 6ZVP for the TH structure²⁵, we focused on the DNAJC12 residues 189-198, that interact with Helix 2 of one of the dimeric RD of TH (residues 140-150) (Fig. 6c). We performed site-directed mutagenesis at the conserved TH residues **L141**, **L144**, and **L145**, which are the center of a hydrophobic cluster (**L141-A142-A143-L144-L145**) in Helix 2 of TH-RD. This helix represents the closest interacting area with the C-terminal of DNAJC12 in the structural model of the complex, notably with **F193** (Figs. 6c, i). The variants **L141R**, **L144A**, and **L145R** were unstable when expressed in *E. coli* and could not be purified. This result supports the importance of these residues in maintaining the structure of the RD through interaction with other hydrophobic residues in Helix 1 of TH-RD. On the other hand, Ala mutations at **L141** and **L145** were less deleterious and could be purified, showing a small but significant increase in binding affinity for DNAJC12 (2-4-fold reduction in K_D compared with TH-WT) (Supplementary Fig. 7 and Table 1). Mutations at protein-protein interaction areas often lead to decreased binding affinity^{2, 26}, but they may also result in tighter binding due to improvement of packing and reduction of eventual steric hindrances²⁷. Mutations **L141A** and **L145A** may thus result in some release of the interactions with Helix 1, leading to a higher packing towards **F193** (Fig. 6c, i). Our results suggest that the main interface between TH and DNAJC12 may have evolved to achieve a “fit for purpose” affinity^{27, 28} rather than the strongest interaction. On the other hand, the involvement of DNAJC12-F193 in the PPI was proven with the mutant F193A, which in this case clearly resulted in a reduced affinity for TH (8-fold increased K_D) (Supplementary Fig. 7 and Table 1), supporting the importance of this very conserved C-terminal residue for interactions with the hydrophobic hotspot in Helix 2 of TH-RD.

Finally, regarding the interactions of the DNAJC12-JD with the TH-RD of the other subunit in the same dimer that interacts with the C-terminal of DNAJC12, the structural model of the complex shows that L141 and L145 in Helix 2 of TH interact with I65 and L66 in Helix III of the JD of DNAJC12 (Fig. 6c, ii). The weaker and lesser interactions in this area compared with the interactions of the same region of TH with the CTD of DNAJC12 (Fig. 6c, i) are consistent with the very low affinity binding of the isolated JD with TH (Fig. 2c; Table 1). Although the affinity for the TH-DNAJC12 interaction (K_D ~150 nM) is relatively high for JDPs interactions with their clients (typical K_D in the 400 nM range^{29, 30, 31}), it can still be considered of medium affinity for PPIs in general, which normally present K_D -values from 0.1 nM to pM)³², and thus apparently adapted to a function and purpose that includes the relations of TH with other regulatory proteins, some with higher affinity for TH than DNAJC12 as we have argued in the Discussion (First paragraph of “The interaction interfaces between TH and DNAJC12”).

5. “DnaJC12:TH complex stimulated Hsp70 hydrolysis but DnaJC12 prevented aggregation in an Hsp70 independent manner”. Related to comment above – this has to be clarified. What is the role of Hsp70 in this activity?

>> The binding of DNAJC12 to TH certainly increased its thermal stability and protected the enzyme towards aggregation, both for wild-type TH (Fig. 3c,d) and for the THD-associate variant TH-R202H (Fig. 8d). This stabilizing and antiaggregating function of DNAJC12, which does not require the intervention of Hsc70, is known as *holdase (or holding) activity*. Differently to the folding/refolding and disaggregase activities, and other additional functions involving JDPs, such as exchange of protein partners or change of oligomeric state that are certainly dependent on Hsp70 chaperones and activation of the ATPase activity, the holdase activity of JDPs does not require Hsp70 chaperones^{33, 34 35, 36}. Nevertheless, there is certainly a possibility of confusion with the concept of holdase activity related to Hsp70 proteins

and other molecular chaperones³⁷. Therefore, in the revised version we have improved the description of the holdase activity of JDPs with some comments in the Introduction and in a separate section in the Discussion (“The TH:DNAJC12 complex, holdase activity”).

6. Indicate wavelength used to measure changes in secondary structure to probe stability in Fig 1e.

>> The wavelength (222 nm) is now provided in the text and in the y-axis legend.

7. What is suggested by the cumulative stabilizing effect? Lines 243-244 “At least in vitro, TH is vulnerable to time-dependent aggregation [39] that can be delayed by certain ligands, in particular DA [40]”.

>> This statement referred to the finding that the regulation of TH by its downstream product and feedback inhibitor DA, which has a stabilizing effect on TH²⁵, does not appear to be affected by DNAJC12 binding and vice versa (Fig. 3b), as the two processes induce stabilization, and by being independent also induce an additive effect. We have changed cumulative to additive.

8. The DSF T_m changes seem pretty subtle (Fig 3a,c). How was significance calculated? Given this affinity, I would expect a larger increase in stability of the DnaJC12:TH complex. Perhaps using more established methods such as CD (as show in Fig. 1e) to measure changes in stability would be more informative.

>> The statistical significances were calculated, by one-way ANOVA and pair-wise multiple comparisons by Tukey’s post-hoc HSD test of the T_m-values (n=3 independent samples). We have now performed additional measurements using temperature dependent CD and calculated the ΔT_m change by this method. As seen in new Fig. 3a (numerical data in Fig. legend), we have obtained very similar results to those obtained by DSF: T_m (mean ± SD): 51.2 ± 0.2°C (TH) and 53.5 ± 0.47°C (TH:DNAJC12) by CD and T_m (mean ± SD): 51.76 ± 0.03°C (TH) and 53.40 ± 0.14°C (TH:DNAJC12) by DSF; i.e. (ΔT_m(CD)= +2.3°C; ΔT_m(DSF)=+1.7°C). We agree with the reviewer that the changes are subtle. Nevertheless, although the ΔT_m certainly depends on the affinity of the interaction there are other factors that affect it, such as the difference in energy between the native and the unfolded state of the protein target when bound to the ligand (DNAJC12 in this case) compared with the difference in energy for the unbound protein. This difference will be dependent on the flexibility of the ligand-binding site and degree of conformational change upon binding, as well as on the stability of the target at the initial, unbound conditions³⁸. Indeed, the main interacting area involve folded regions, i.e. Helix 2 of TH-RD and the C-terminal helical section of DNAJC12 (residues 189-198) (Fig. 6c), are not expected to undergo large conformational changes themselves upon binding, except the related to packing, in agreement with the small value of T_m (see also answer to comment 4 above).

9. Lines 253-256. The size distribution of the samples of TH:DNAJC12 remained similar at 0 min and 100 min, while larger species and aggregates dominated the size distribution of the samples containing TH alone at the same endpoint (Supplementary Fig. 6). At 100 mins TH alone produces a shifted population that is small plus large species. The smaller shifted population at 100 mins looks pretty defined. Have the authors looked at these samples using TEM or alternate methods? Are these assemblies uniform in terms of size? I would expect that if the samples aggregated it would be a broader size distribution. These experiments should also include +/- DA.

>> We appreciate this comment While DLS is an appropriate and effective method for measuring aggregation kinetics, it has limitations in determining the precise sizes of the protein aggregates^{39 40}). Therefore, as suggested by the reviewer, we have used negative staining electron microscopy to compare the putative protection against aggregation of TH by DNAJC12 and DA, and to determine aggregate size. As seen in Fig. 3d, after 100 min incubation at 37 °C, the micrographs of TH with and

without DA, both in the absence and presence of DNAJC12, indicate that DNAJC12 is significantly more effective than DA in preventing TH aggregation. In the presence of DA we observed a reduction of the aggregates from 30 ± 3 nm (TH alone) to 21 ± 3 nm (TH(DA)) (Fig. 3d, left panels). In conditions where DNAJC12 is present, regardless of the presence of DA (Fig. 3d, right panels), a homogeneous particle size distribution (11 ± 2 nm) is observed, closely matching the diameter of individual TH molecules. We have substituted the previous Supplementary Fig. 6c by a new Fig. 3d to appropriately present the TEM results.

10. Lines 258-268. How is the conversion from L-Tyr to L-Dopa? The authors should include a more detailed description of how TH activity was measured?

>>The assay to measure the BH₄-dependent hydroxylation of Tyr to L-Dopa catalyzed by TH, is described in the Materials and Methods, in section "Assay of TH activity and inhibition by DA". We have complemented this section with the original reference describing the method⁴¹, but we still describe in detail the conditions used in the assay.

11. Supp Fig 4 should be a part of the Main Fig 4 panel. In Fig 3, the font is small and hard to read.

>> We appreciate these comments for improvement of Figs. 4 and 3. As Fig. 4 is already very large, and other sections and experiments in the manuscript that deal with different variants also refer to Supplementary Fig. 4 (Supplementary Fig. 5 in the revision), we have kept it in the supplementary section. Nevertheless, we have also included the three relevant schematic representations for the DNAJC12 forms below each panel in Fig. 4. We have also increased the font in Fig. 3.

12. In explanation of the cryo-EM data, the authors have mentioned: "This domain amounts 303 to half of the mass of the whole cochaperone, so we assumed that the JD should be assigned to 304 the largest of the two masses described (red asterisk in Fig. 4a and orange mass in 305 Supplementary Fig. 9a". In the figure legend it is mentioned as yellow mass.

>> The mention to "red asterisk in Fig. 4a" has been corrected to "orange asterisk in Fig. 4a" and the mention to the "yellow mass", in previous Supplementary Fig. 9 (Supplementary Fig. 11 in the revision) has also been corrected to "orange mass".

13. Overall, the figure legends need to be improved for clarity, concise descriptions that help explain the data. The legend of Fig 4 is difficult to follow, please clarify.

Figure 4. 3D reconstruction of the TH:DNAJC12, TH:DNAJC12(108-198) and 1298 TH(DA):DNAJC12 complexes. Two views of the TH:DNAJC12 (a), TH:DNAJC12(108-198) (c) and TH(DA):DNAJC12 (e) complexes. The red and black asterisks in a) point to the extra masses that appear in the 3D reconstruction when compared with that of apo-TH (PDB 7A2G). The red circles point to the location of two of four TH active sites. The red asterisks in e) point to the N-terminal regulatory α -helix (residues 39-58) of TH in TH(DA) (PDB 6ZVP). The same two views of TH:DNAJC12 (b), TH:DNAJC12(108-198) (d) and TH(DA):DNAJC12 (f) complexes with the atomic structure of apo-TH and the AlphaFold model of DNAJC12 docked into the cryoEM volume (docking is performed only in one of the corresponding masses, the top one). In the atomic structure of apo-TH (PDB 7A2G) (b,d) the central, tetrameric CD+OD domains are colored in cyan and the two dimeric RDs in dark blue. The same color code is used for TH in TH(DA) (PDB 6ZVP), with the N-terminal regulatory α -helix shown in magenta (f). For the AlphaFold atomic model of DNAJC12, the JD (identical to the atomic structure obtained by NMR; PDB 2CTQ) is colored in orange, the CTD in red and the central linker region in dark green. This is an unstructured region and many other options are possible.

>> We recognize the need to improve the figure legend for greater conciseness and clarity. We have now edited all of them, and, concretely, the legend of Fig. 4 has been updated as follows:

Figure 4. 3D reconstruction by cryoEM of the TH:DNAJC12, TH:DNAJC12(108-198) and TH(DA):DNAJC12 complexes.

a, **c**, and **e** display two views of the TH-DNAJC12, TH-DNAJC12(108-198) and TH(DA)-DNAJC12 complexes, respectively. In **a**, red and orange asterisks indicate extra masses compared to apo-TH (PDB 7A2G). Red circles mark two of the four TH active sites. In panel **e**, red asterisks show the N-terminal regulatory α -helix (residues 39-58) of TH in TH(DA) (PDB 6ZVP), entering the TH active sites. **b**, **d**, and **f** show the same views with apo-TH and the AlphaFold model of DNAJC12 docked into the cryoEM volume. In all panels, the TH domains are colored in dark blue (RD), cyan (CD), and green (OD), and the regulatory domain α -helix (39-58) in TH(DA) (PDB 6ZVP) is colored in magenta (**f**). The AlphaFold model of DNAJC12 has the JD in orange and the CTD in red. Schematic diagrams of the relevant DNAJC12 variants used are represented below the figures.

We hope the legends have been improved and are now easier to follow.

14. Fig 7: What could be the possible role of the DnaJC12:TH complex activating ATPase activity of Hsp70? What happens to the DnaJC12:TH complex following Hsp70s activation? Does Hsp70 disassemble the complex? Please annotate the Figure to show what light and dark grey is.

>> As presented in the answer to comment 2 from reviewer #2, we have considered two main possible alternatives to explain the specific function of the TH:DNAJC12 complex, (i) the release of JD autoinhibition and/or (ii) synergistic stimulation of the ATPase by the JD of DNAJC12 and substrate TH. All together our results seem to point to a combination of both approaches, where DNAJC12 would work as a specific chaperone for TH. The novel experiments included in the revised version (like the result with the Hsc70-V438F variant, compared with WT-Hsc70; Fig. 7d) and the improved model of the tertiary complex TH:DNAJC12:Hsc70 (Supplementary Fig. 12a; Supplementary Movie S1) further supports the role of the DnaJC12:TH complex as the Hsc70-activating entity. About the question of Hsp70 disassembling the complex, see answer to comment 2, above, on the lack of effect of mutant Hsc70- Δ C10, which seems to indicate that -differently to the case with DNAJB1 as cochaperone of Hsc70, the mechanism for dissociation of the DNAJC12 complex or its components following Hsc70 activation may not imply the C-terminal of Hsc70 (See Fig. 1; page 12, above).

We appreciate the comment about the legend to Fig. 7. We have now completed it to describe the data represented in light gray or dark gray, which was previously only described for the inset.

15. The authors should include gels of the crosslinked samples to confirm complex formation and to ensure that the data is not derived from aggregated species.

>> The new version of the manuscript includes a new figure (Supplementary Fig. 16) showing the SDS-PAGE of all the crosslinked samples. The bands corresponding to the complexes analyzed by XL-MS are highlighted with asterisks.

16. Some of the lysines in the models look to be buried and in some other instances the crosslinks appear to thread through proteins. It might be good to calculate the distances using a solvent accessible path with software such as Xwalk (PMID 21666267). For the XL-MS analysis please also include plots illustrating the distance distributions of the intramolecular and intermolecular mapped crosslinks. Some of the contacts appear longer than the spacing of the crosslinker (and they may appear even longer if using solvent accessible paths). Have the authors exhaustively mapped all possible intermolecular contacts to confirm that the geometries shown are optimal for the contacts?

>> The distances were calculated and graphically represented using ChimeraX, a versatile molecular graphics program tailored for such analyses. However, for additional validation, we also employed Xwalk in distance calculations and we did not observe noticeable differences using both strategies (see Table 1 in this letter, below).

		MAX. DISTANCE ChimeraX (Å)	MAX. DISTANCE Xwalk (Å)		
DnaJC12	62AKEILTNEESR ⁷²	90ATKPSALSR ⁹⁸	28.046	29.4	TH WT
	181WSKDAPSELLR ¹⁹¹	77EGKAMLNLLFSPR ⁸⁹	9.586	11.3	
	181WSKDAPSELLR ¹⁹¹	169KVSELDK ¹⁷⁵	33.938	33.3	
	184DAPSELLR ¹⁹¹	157SPAGPK ¹⁶²	23.879	22.3	
	192KFR ¹⁹⁴	157SPAGPKVPWFPR ¹⁶⁸	26.661	26.7	TH (DA)
	62AKEILTNEESR ⁷²	90ATKPSALSR ⁹⁸	28.046	29.4	
	62AKEILTNEESR ⁷²	99AVKVFETFEAK ¹⁰⁹	20.489	22.0	
	81SQMSMPFQQWEAL NDSVKTSMHWVVR ¹⁰⁷	157SPAGPKVPWFPR ¹⁶⁸	32.259	30.3	
	181WSKDAPSELLR ¹⁹¹	77EGKAMLNLLFSPR ⁸⁹	9.586	11.3	
	181WSKDAPSELLR ¹⁹¹	169KVSELDK ¹⁷⁵	33.938	33.3	
	181WSKDAPSELLR ¹⁹¹	157SPAGPKVPWFPR ¹⁶⁸	27.322	26.6	
	181WSKDAPSELLR ¹⁹¹	90ATKPSALSR ⁹⁸	34.005	35.4	
62AKEILTNEESR ⁷²	99AVKVFETFEAK ¹⁰⁹	20.489	35.192	RDS (TH)	
62AKEILTNEESR ⁷²	90ATKPSALSR ⁹⁸	28.046	29.4		
DnaJC12 CTD	181WSKDAPSELLR ¹⁹¹	77EGKAMLNLLFSPR ⁸⁹	9.586	11.3	TH WT
	181WSKDAPSELLR ¹⁹¹	77EGKAMLNLLFSPR ⁸⁹	9.586	11.3	
	181WSKDAPSELLR ¹⁹¹	157SPAGPKVPWFPR ¹⁶⁸	27.322	26.6	
	181WSKDAPSELLR ¹⁹¹	90ATKPSALSR ⁹⁸	34.005	35.4	
DnaJC12 C23	181WSKDAPSELLR ¹⁹¹	77EGKAMLNLLFSPR ⁸⁹	9.586	11.3	TH WT
	181WSKDAPSELLR ¹⁹¹	157SPAGPKVPWFPR ¹⁶⁸	27.322	26.6	
	181WSKDAPSELLR ¹⁹¹	90ATKPSALSR ⁹⁸	34.005	35.4	

Table 1. Comparison of the intermolecular crosslink distances calculated with ChimeraX and Xwalk.

Please note that Reviewer #3 expressed a similar concern related to the excessive length of certain cross-links, prompting us to modify Figure 5. In response, in the revised figure we have focused solely on illustrating intermolecular crosslinks, including their distances, recognizing their significance in gaining insights into the TH:DNAJC12 interaction. We have now removed from the representation highly flexible regions that do not adjust to the distance constraints of BS3. Unfortunately, most of the intramolecular cross-links are obtained between such unstructured or flexible regions and do not meet the distance constraints, so they fall out of the representation. If interested, all data, including intramolecular and mono-crosslinks, are available for review at the PRIDE repository (<https://www.ebi.ac.uk/pride/>; Project: PXD045185). Additionally, to provide a comprehensive overview, we have included a Table in the new version of the manuscript (Supplementary Table 4) presenting distance distributions for all possible intermolecular contacts for each inter protein cross-link. These data clearly indicate that only one contact is possible for each cross-link.

The robustness of our XL-MS findings is based on five different analyses, ranging from the largest, TH:DNAJC12 complex to the smallest, RDS:DNAJC12(176-198), which played a pivotal role in confirming the accuracy of our model, particularly for the C-terminus of DNAJC12 (Figures 4-6). Notably, all the inter cross-links obtained between RDS and DNAJC12 align closely with the proposed structural model (Figure 5d,e).

17. Line 486, what do the authors mean by “heterogenous clinical spectrum”. Does this imply that conditions that include hyperphenylalaninemia, L-Dopa responsive dystonia, severe parkinsonism and intellectual disability are all linked to DnajC12 loss-of-function?

>> The answer is yes, as that is certainly the case and the primary meaning of “heterogeneous” in this case. But the term also covers the heterogeneity with respect to the severity of the phenotype. We have made some changes to the text in the first paragraph of the Discussion, to clarify this aspect: “Patients with congenital DNAJC12 deficiency present monoamine neurotransmitter decline and heterogenous clinical spectrum, including hyperphenylalaninemia as well as parkinsonism of varying severity, from L-Dopa-responsive dystonia to severe neurodevelopmental delay and intellectual disability”

Reviewer #5 (Remarks to the Author):

This manuscript describes the interaction of the molecular chaperone DNAJC12 with tyrosine hydroxylase (TH) tetramer, a key enzyme involved in dopamine synthesis. TH is prone to aggregation in vitro and mutations in DNAJC12 are linked to neurometabolic disorders and depletion of dopamine. Thus, understanding the role DNAJC12 in chaperoning and stabilizing TH is of significant interest. Biochemical experiments including SEC-MALS, DSF, and TH activity assays were used to identify that two DNAJC12 molecules bind and stabilize TH tetramer. The authors perform binding analyses to show the conserved C-terminal domain of DNAJC12 and is involved in the interaction. While DNAJC12 reduces aggregation of TH, there appears to be no significant change in TH activity with DNAJC12.

Importantly, the authors perform Cryo-EM and crosslinking-mass spec to characterize the structure of the TH- DNAJC12 complex. While the authors claim DNAJC12 binds to the TH regulatory region, the resolution of the map is not sufficient to convincingly localize DNAJC12 or define the interaction. Indeed, the RD is flexible and at low resolution in their previously published structure of the TH tetramer and the current maps do not clearly show additional density that corresponds to the JD or the CTD of DNAJC12 given the low thresholding values used in the map images.

>> We are aware (and already have some experience) of the difficulties in obtaining high-resolution structures with full-length TH. Despite achieving a satisfactory resolution in the central structure of the tetramer (the CD+OD domains), the external RDs proved to be a challenge due to their remarkable flexibility. For this reason, to prepare the revision, we embarked on a new massive acquisition of the TH:DNAJC12 complex with the aim of improving the resolution of the complex. We collected several more millions of particles and performed a refinement which allowed us to obtain a new 3D reconstruction of the complex and improve the overall resolution up to 4.25 Å (see new Figure 4 and 5). Although we were unable to improve the resolution of RDs and DNAJC12, a significant number of intermolecular crosslinks were identified between DNAJC12 and the RD residues, aligning well with the proposed model. Notably, binding analysis conducted by BLI (Figs. 6a) unequivocally establishes the binding of DNAJC12 to the RDs. Acknowledging the limitations of the maps generated, in particular for the DNAJC12 binding region, insufficient for a straightforward localization of DNAJC12 domains, we have modified the Supplementary Fig. 11, which now shows a difference map between TH alone and TH:DNAJC12, highlighting the specific densities corresponding to the C-terminal region and the JD of DNAJC12. We have experimentally determined these differences (the protocol is specified in Materials and Methods section, page 26), which align adequately with the sizes of the structured domains of DNAJC12.

Furthermore, their reported resolution of 5.6 Å by FSC appears to be wrong based on the map density. Virtually no structural feature (e.g. alpha helices) can be observed in the images shown and the notable regions of possible DNAJC12 density are likely at ~15 Å resolution or worse, meaning that it is impossible to identify the small JD and CTD regions of DNAJC12 that are expected to be structured and bound. Perhaps the FSC curve is reporting an artificially high resolution because of masking artifacts? Since the authors do not show the necessary validation data (fsc curves for map and model, masked, unmasked, angular sampling plot, etc.), it is difficult to know.

Apart from the difference map determination explained in the previous concern which allowed us to identify the DNAJC12 domains, it is important to point out that the choice of the threshold significantly impacts the observation of distinct secondary structural elements, such as alpha helices. We specifically selected to show a map calculated for sigma 1.6 level with the aim of depicting the densities of the TH and DNAJC12 domains. We believe that this is the most suitable representation to allow a correct interpretation of our data, but it is inevitable that at this threshold certain structural features, including alpha helices in some domains of TH, appear less pronounced compared to when a threshold of 3 sigma is applied. Such explanation is depicted in the following Figure 2 of this letter, below.

Figure 2: The final map of TH-DNAJC12 is represented using 3 different σ values. The choice of the threshold substantially influences the visualization of the corresponding secondary structural elements.

Furthermore, in the new version of the manuscript, we have included a more complete Supplementary Fig. 9 which shows the mentioned important validation data (similar information can be found in the validation reports) except the FSC model-map curve. We do not show the latter because we have not built any atomic model. We only show the known, previously solved TH (PDB entries 7A2G and 6ZVP) and the AlphaFold predicted full-length DNAJC12 (AF-Q9UKB3-F1) structures docked in our maps because unfortunately the resolution is not suitable to attempt model building.

Regarding the FSC analysis of the TH-DNAJC12 map (Supplementary Fig. 9), it is important to clarify that it does not contain any artefacts for the following reasons: 1) the masked and corrected FSC are identical, indicating that there are no discrepancies between the two; 2) the FSC of the map has no peaks and valleys, which is an indication that no artefacts are present; and 3) the global resolution of the map is within the range of the local resolutions, confirming the absence of artefacts.

The TH tetramer core may be at ~ 5 Å resolution based on Supplementary Fig. 8a however it is certainly not 3 Å, which is what is claimed, and there appears to be substantial angular sampling/preferred orientation issues based in the smeared density.

>> As mentioned above, depending on the sigma levels chosen for the volume representation, we certainly see more or less secondary structural elements. In fact, some high-resolution features were clearly observed in the CD and OD of TH from the initial steps of image processing as the 2D classifications (see below, Figure 3 of this letter). However, one of the main problems we have encountered during image processing is related to the RDs flexibility, and their low resolution and smeared appearance can be clearly appreciated in the figure below (red asterisks), where the RD densities are observed as blobs with no secondary structural details. These issues are shared among the different views of the complex as it can be appreciated in the angular distribution plot included in Supplementary Fig. 9. This diagram indicates that the final 3D reconstruction is fully-covered with all the potential orientations discarding any problems related with preferred orientations and indicating that smeared densities are closely connected with the RD intrinsic flexibility.

Figure 3. 2D classification of TH:DNAJC12 complex showing selected TH side views. RD densities are highlighted by red asterisks.

>> Additionally, to improve the interpretability of the obtained local resolution maps, we have re-estimated the anisotropic resolution of the final maps using a more precise protocol specified in the Methods section (Page 26) and now included in a new version of Supplementary Fig. 10.

Additionally, crosslinking-mass spec results are not convincing and do not support or validate their cryoEM model. Notably the analyzed crosslink sites on DNAJC12 appear highly selected and are largely from flexible, unstructured regions outside the cryoEM map and are thus not useful for localization

>> An extensive review of the crosslinking data was carried out in response to the concerns by reviewer #3, and also to comments 15 and 16 by reviewer #4 (see above), although neither of these reviewers raised any doubts that the crosslinks supported our structural data. However, both did share the concern also posed by the reviewer #5, that flexible and unstructured regions led to longer distances than those imposed by BS3. Our explanation is that this was done deliberately to suggest there must be a movement of these regions, but we understand that this might be unclear so we have modified

Figure 5 excluding these regions from the structural analysis. We mistakenly attempted to represent crosslinks within the flexible central region of DNAJC12 using an AlphaFold model even if this region is not present in our 3DRs. We are thankful for the reviewers' comments and appreciate that the modified Figure 5 is clearer.

Finally, all the inter crosslinks obtained in the different analyses are summarized in Supplementary Tables 3 and 4. Many of them come from unstructured and/or flexible regions and are not displayed in our structural model because they don't meet the criteria to be included. Fortunately, there is an important number of crosslinks involving the regions depicted in our maps that support our proposed model, fitting appropriately the BS3 restraints, with no selective bias applied.

Overall, although the biochemical experiment results are important in revealing strong 2:4 binding of DNAJC12 to TH and providing insights for a potential stabilizing role of DnaJC12 in this complex, the cryo-EM and mass spectrometry approaches are not suitable for publication and need to be corrected and advanced substantially to define the interaction between DnaJC12 and TH.

>> The biochemical analysis allowed us to constrain and define the RDs as the DNAJC12 interacting domains and although we had difficulties in obtaining high-resolution from cryoEM data, a robust XL-MS analysis using five different subcomplexes was carried out to confirm our model. Furthermore, site-directed mutagenesis has also contributed to validating the structural model of the complex (See answer to comment 4, to reviewer #4 and new text in the Results section, from last paragraph in Page 11). We believe that this new version of the manuscript solves most of the concerns related with the structural approaches and includes more convincing and robust information, despite the limited resolution shown around the RDs.

Thus, the increased stabilization observed on TH after DNAJC12 binding (increased T_m and decreased aggregation) does not appear associated with rigidification of the RDs, but rather with the maintenance of the tetrameric integrity, avoiding the dissociation of the tetrameric assembly into monomers, which might precede misfolding and subsequent aggregation, as observed for other tetrameric proteins, notably transthyretin^{42, 43, 44} (see Fig. 6b and associated comments end of page 15 in the Discussion). Indeed, as shown in Supplementary Movie S2 the mobility and flexibility at the RDs appear to increase upon DNAJC12 binding to TH, resulting in a decrease of resolution at these domains (10-14 Å) compared with apo-TH alone (7-9 Å). The increased mobility might represent a functional effect to enhance accessibility of the regions to be folded/refolded in TH by the Hsc70 chaperone, which seems supported by findings with other JDPs:client complexes^{3, 28}.

Other sources of low resolution in the RD region of TH include the high flexibility of DNAJC12 due to its disordered central region, as well as the high plasticity of the RDs both on their own and in relation to the central, very rigid CD+OD domains, which hinders high-resolution definition. Fortunately, using XL-MS complemented with site-directed mutagenesis has provided us with valuable insights into the interaction between TH and DNAJC12.

References

1. Korner G, *et al.* Brain catecholamine depletion and motor impairment in a Th knock-in mouse with type B tyrosine hydroxylase deficiency. *Brain* **138**, 2948-2963 (2015).
2. Xiong D, Lee D, Li L, Zhao Q, Yu H. Implications of disease-related mutations at protein-protein interfaces. *Curr Opin Struct Biol* **72**, 219-225 (2022).

3. Zhang R, Malinverni D, Cyr DM, Rios PL, Nillegoda NB. J-domain protein chaperone circuits in proteostasis and disease. *Trends Cell Biol* **33**, 30-47 (2023).
4. Cabrera Y, *et al.* Regulation of Human Hsc70 ATPase and Chaperone Activities by Apg2: Role of the Acidic Subdomain. *J Mol Biol* **431**, 444-461 (2019).
5. Laufen T, *et al.* Mechanism of regulation of hsp70 chaperones by DnaJ cochaperones. *Proc Natl Acad Sci U S A* **96**, 5452-5457 (1999).
6. Faust O, *et al.* HSP40 proteins use class-specific regulation to drive HSP70 functional diversity. *Nature* **587**, 489-494 (2020).
7. Barouch W, Prasad K, Greene L, Eisenberg E. Auxilin-Induced Interaction of the Molecular Chaperone Hsc70 with Clathrin Baskets. *Biochemistry* **36**, 4303-4308 (1997).
8. Ciesielski SJ, *et al.* Interaction of J-protein co-chaperone Jac1 with Fe-S scaffold Isu is indispensable in vivo and conserved in evolution. *J Mol Biol* **417**, 1-12 (2012).
9. Kityk R, Kopp J, Mayer MP. Molecular Mechanism of J-Domain-Triggered ATP Hydrolysis by Hsp70 Chaperones. *Mol Cell* **69**, 227-237.e224 (2018).
10. Parra LA, *et al.* The Molecular Chaperone Hsc70 Interacts with Tyrosine Hydroxylase to Regulate Enzyme Activity and Synaptic Vesicle Localization. *J Biol Chem* **291**, 17510-17522 (2016).
11. Zhu X, *et al.* Structural analysis of substrate binding by the molecular chaperone DnaK. *Science* **272**, 1606-1614 (1996).
12. Van Durme J, Maurer-Stroh S, Gallardo R, Wilkinson H, Rousseau F, Schymkowitz J. Accurate prediction of DnaK-peptide binding via homology modelling and experimental data. *PLoS Comput Biol* **5**, e1000475 (2009).
13. Gutierrez MBB, Bonorino CBC, Rigo MM. ChaperISM: improved chaperone binding prediction using position-independent scoring matrices. *Bioinformatics* **36**, 735-741 (2019).
14. Fernandez-Escamilla AM, Rousseau F, Schymkowitz J, Serrano L. Prediction of sequence-dependent and mutational effects on the aggregation of peptides and proteins. *Nat Biotechnol* **22**, 1302-1306 (2004).
15. Götze M, Pettelkau J, Fritzsche R, Ihling CH, Schäfer M, Sinz A. Automated assignment of MS/MS cleavable cross-links in protein 3D-structure analysis. *J Am Soc Mass Spectrom* **26**, 83-97 (2015).
16. Cammarata MB, Macias LA, Rosenberg J, Bolufer A, Brodbelt JS. Expanding the Scope of Cross-Link Identifications by Incorporating Collisional Activated Dissociation and Ultraviolet Photodissociation Methods. *Anal Chem* **90**, 6385-6389 (2018).
17. Leitner A, *et al.* Chemical cross-linking/mass spectrometry targeting acidic residues in proteins and protein complexes. *Proc Natl Acad Sci U S A* **111**, 9455-9460 (2014).
18. Merkley ED, Rysavy S, Kahraman A, Hafen RP, Daggett V, Adkins JN. Distance restraints from crosslinking mass spectrometry: Mining a molecular dynamics simulation database to evaluate lysine–lysine distances. *Protein Science* **23**, 747-759 (2014).
19. Liu Q, Liang C, Zhou L. Structural and functional analysis of the Hsp70/Hsp40 chaperone system. *Protein Sci* **29**, 378-390 (2020).
20. Yu HY, Ziegelhoffer T, Craig EA. Functionality of Class A and Class B J-protein co-chaperones with Hsp70. *FEBS Letters* **589**, 2825-2830 (2015).
21. Freeman BC, Myers MP, Schumacher R, Morimoto RI. Identification of a regulatory motif in Hsp70 that affects ATPase activity, substrate binding and interaction with HDJ-1. *Embo j* **14**, 2281-2292 (1995).
22. Suzuki H, Noguchi S, Arakawa H, Tokida T, Hashimoto M, Satow Y. Peptide-binding sites as revealed by the crystal structures of the human Hsp40 Hdj1 C-terminal domain in complex with the octapeptide from human Hsp70. *Biochemistry* **49**, 8577-8584 (2010).
23. Johnson OT, Nadel CM, Carroll EC, Arhar T, Gestwicki JE. Two distinct classes of cochaperones compete for the EEVD motif in heat shock protein 70 to tune its chaperone activities. *J Biol Chem* **298**, 101697 (2022).

24. Jiang J, Prasad K, Lafer EM, Sousa R. Structural basis of interdomain communication in the Hsc70 chaperone. *Mol Cell* **20**, 513-524 (2005).
25. Bueno-Carrasco MT, *et al.* Structural mechanism for tyrosine hydroxylase inhibition by dopamine and reactivation by Ser40 phosphorylation. *Nature Communications* **13**, 74 (2022).
26. Layton CJ, Hellinga HW. Quantitation of protein-protein interactions by thermal stability shift analysis. *Protein Sci* **20**, 1439-1450 (2011).
27. Parisi G, *et al.* Design of protein-binding peptides with controlled binding affinity: the case of SARS-CoV-2 receptor binding domain and angiotensin-converting enzyme 2 derived peptides. *Front Mol Biosci* **10**, 1332359 (2023).
28. Arhar T, Shkedi A, Nadel CM, Gestwicki JE. The interactions of molecular chaperones with client proteins: why are they so weak? *Journal of Biological Chemistry* **297**, 101282 (2021).
29. Mok SA, *et al.* Mapping interactions with the chaperone network reveals factors that protect against tau aggregation. *Nat Struct Mol Biol* **25**, 384-393 (2018).
30. Hou Z, *et al.* DnaJC7 binds natively folded structural elements in tau to inhibit amyloid formation. *Nat Commun* **12**, 5338 (2021).
31. Rüdiger S, Schneider-Mergener J, Bukau B. Its substrate specificity characterizes the DnaJ co-chaperone as a scanning factor for the DnaK chaperone. *Embo j* **20**, 1042-1050 (2001).
32. Erijman A, Rosenthal E, Shifman JM. How structure defines affinity in protein-protein interactions. *PLoS One* **9**, e110085 (2014).
33. Hageman J, *et al.* A DNAJB chaperone subfamily with HDAC-dependent activities suppresses toxic protein aggregation. *Mol Cell* **37**, 355-369 (2010).
34. Ajit Tamadaddi C, Sahi C. J domain independent functions of J proteins. *Cell Stress Chaperones* **21**, 563-570 (2016).
35. Velasco-Carneros L, *et al.* The self-association equilibrium of DNAJA2 regulates its interaction with unfolded substrate proteins and with Hsc70. *Nat Commun* **14**, 5436 (2023).
36. Marszalek J, *et al.* J-domain proteins: From molecular mechanisms to diseases. *Cell Stress Chaperones* **29**, 21-33 (2024).
37. Rutledge BS, Choy WY, Duennwald ML. Folding or holding?-Hsp70 and Hsp90 chaperoning of misfolded proteins in neurodegenerative disease. *J Biol Chem* **298**, 101905 (2022).
38. Du X, *et al.* Insights into Protein-Ligand Interactions: Mechanisms, Models, and Methods. *Int J Mol Sci* **17**, (2016).
39. Li Y, Lubchenko V, Vekilov PG. The use of dynamic light scattering and brownian microscopy to characterize protein aggregation. *Rev Sci Instrum* **82**, 053106 (2011).
40. Bansal R, Gupta S, Rathore AS. Analytical Platform for Monitoring Aggregation of Monoclonal Antibody Therapeutics. *Pharm Res* **36**, 152 (2019).
41. Bezem MT, *et al.* Stable preparations of tyrosine hydroxylase provide the solution structure of the full-length enzyme. *Scientific Reports* **6**, 30390 (2016).
42. Johnson SM, Connelly S, Fearn C, Powers ET, Kelly JW. The transthyretin amyloidoses: from delineating the molecular mechanism of aggregation linked to pathology to a regulatory-agency-approved drug. *J Mol Biol* **421**, 185-203 (2012).
43. Saelices L, *et al.* Uncovering the Mechanism of Aggregation of Human Transthyretin. *J Biol Chem* **290**, 28932-28943 (2015).
44. Oliveira JBS, *et al.* Initial biophysical characterization of *Amyntas gracilis* giant extracellular hemoglobin (HbAg). *Eur Biophys J* **49**, 473-484 (2020).

Point by point responses

Reviewer #1 (Remarks to the Author):

The revised version of the manuscript has correctly addressed all the issues I raised concerning the previous version. Overall, the revised manuscript has improved in clarity and in quality, presenting novel results that add to the scientific relevance and to the soundness of the conclusions. In my opinion, the manuscript is now acceptable for publication.

>> We appreciate the time and effort the reviewer has invested in providing us with invaluable feedback. We are delighted that you find the manuscript acceptable for publication.

Reviewer #2 (Remarks to the Author):

The authors have addressed all comments in my review. However, before the manuscript is accepted for publication it requires a major revision of the text.

The current version of the manuscript is much less clear than the original one because the authors added new information in answer to the reviewers' comments, without considering how they affected the ability of the reader to understand the manuscript. The current version of the Results contains too many speculations, which should be moved into the Discussion. Whereas the Discussion contains too many repetitions from the Results. Both chapters should be significantly shortened. Below I have listed a few fragments of the text that illustrate my point.

>> We are thankful to the reviewer for a thorough revision of our manuscript, and we are pleased that our responses to previous comments were satisfactory. We acknowledge that the manuscript had become quite wordy and somewhat difficult to follow. We have worked to improve its clarity based on the reviewer's recommendations:

Results

Line 398: "It is important to mention that fifteen crosslinks host residues from the flexible linker, indicating an interesting role in the interaction with TH." – this sentence is confusing, and as written does not contribute to better understanding of the C12/TH system.

>> This has now been changed to: "Moreover, fifteen crosslinks with TH involve the flexible linker." (Lines 393, 394 in revised version)

Line 414 "Of note, docking of TH:DNAJC12, focusing on a superposition of the JD of DNAJC12 with the JD in the ATP-bound open conformation of Hsp70 of Escherichia coli (DnaK). . . .,- this sentence does not fit to this paragraph, because JD (J-domain) is not mentioned in it.

>> We appreciate this comment. We have rewritten the presentation of the accessibility of the HPD in the JD of DNAJC12 within the TH:DNAJC12 complex (Fig. 6a) and in the *in silico* models of the TH:DNAJC12:Hsp70 complex that are presented in the revised Supplementary Fig. 13 (previous Supplementary Fig. 12) and Supp. Movie 2. Please note that the previous paragraph has been significantly revised to also address reviewer #5 requirement regarding the validation of the TH:DNAJC12 models obtained through cryo-EM and XL-MS using Haddock

(Please see our last response to reviewer #5), and the references to the JD of DNAJC12 are given in this context. See text in lines 410-424 in revised version.

Line 429 “TH is assembled as a dimer of dimers through the ODs that include dimerization and tetramerization interfaces and form the central antiparallel coiled-coil bundle common for the AAAHs (Fig. 6b). By embracing the RD dimers, formed by monomers from subunits in different dimers, DNAJC12 stabilizes the TH tetramer. The stabilization of this dimeric interface may protect TH from dissociation to monomers and exposure of aggregation prone regions (See Discussion)” - this added fragment is confusing and belongs in the Discussion not Results.

>> The indicated text has been moved to the Discussion, and edited as follows: “Tetrameric TH is composed of two dimers, each contributing one RD to each of the two opposite RD dimers⁴⁰. This enhances tetramerization alongside the central helix bundle formed by the ODs (Fig. 6a). By stabilizing the RDs, DNAJC12 prevents the dissociation of the oligomeric assembly into monomers, which precedes misfolding and subsequent aggregation in other tetrameric proteins such as transthyretin⁶²⁻⁶⁴.” (Lines 582-587).

Lines: 446 – 450 “Mutations at protein-protein interactions (PPIs) often result in decreased binding affinity (39,50), but they may also cause tighter binding due to improvement of packing and reduction of eventual steric hindrances (51). Variants L141A and L145A may thus present some release of the interactions with Helix 1 and/or higher packing towards F193 in the CTD of DNAJC12 (Fig. 6c, excerpt I).” - these speculations should be in the Discussion.

>> The indicated text has been moved to the Discussion, adapted as follows: “In contrast, alanine mutations at TH L141 and L145 resulted in a 2-4 fold improvement in binding affinity, potentially due to the release of tight hydrophobic interactions with Helix 1 in the TH-RD (notably V102 and F106) (PDB 6ZVP⁴⁰) that may lead to higher packing towards F193 and increased binding affinity for DNAJC12.” (Lines 613-616).

Line 460 – “Interestingly, this model also supports the holdase activity of DNAJC12, as it shields the β -strand regions 82LNLLLFSP88 and 129EYFVRLE135 (Fig. 6c, excerpts III and IV), which have been predicted to bind to the Hsp70 family by Limbo75 and ChaperISM76 algorithms....” – this added fragment belongs in the Discussion.

>> This paragraph has also been moved to the Discussion section and slightly modified to adapt it to the rest of the text there: “Our functional results along with Limbo⁷³- and ChaperISM⁷⁴-based prediction of residues 82LNLLLFSP88 and 129EYFVRLE¹³⁵ in TH-RD as binding regions to the Hsc70 family supports that TH is an Hsc70 client.” (Lines 627-629).

Line 468 – “As a JDP, DNAJC12 could display disaggregating and (re)folding activities in addition to the ATP-independent autonomous holdase activity aimed at avoiding aggregation of (partially) unfolded client proteins, as showed with TH as substrate...” – this sentence is confusing and it claims too much – the authors did not demonstrate “autonomous holdase activity”, they only show that in vitro C12 binds TH and prevents its aggregation. “Holdase activity” is typically used to refer to the in vivo ability of JDP to carry out a biological function without Hsp70. In my opinion all references to “holdase activity” of C12 should be moved from the Results to Discussion and presented as a hypothesis.

>> We agree that the paragraph might contain overstatements. We have deleted the previous mention to the holdase activity in the Results and have included this comment in the discussion: “The stabilizing activity of DNAJC12 over its client is ATP- and Hsc70-independent, and is reminiscent of the autonomous holdase function of other JDPs, which prevent aggregation of (partially) unfolded client proteins in cellular systems^{8,12}” (Lines 587-589).

Lines 489-502 - this long added fragment is very confusing – in fact it simply states that C12 + TH synergistically stimulate ATPase of partner Hsc70 as expected for any JDP that delivers substrate for Hsc70. Whereas no synergistic stimulation was observed for JDPs (A2/B1) that do not bind TH. The text should be both shortened and simplified. Moreover Apg2, a nucleotide exchange factor, was not introduced in the text(?).

This section has been shortened to:

>> "We then compared Hsc70 activation by DNAJC12 with DNAJA2 and DNAJB1 in the presence of the NEF Apg2. DNAJC12 stimulated Hsc70 ATPase activity less efficiently than DNAJA2 and DNAJB1 (Fig. 7d). TH alone did not activate Hsc70, but with TH, DNAJC12 synergistically enhanced Hsc70 ATPase activity, unlike DNAJA2 and DNAJB1 (Fig. 7d), which do not interact with TH (Supplementary Fig. 7o,p). We also measured the ATPase activity of Hsc70-V438F, a variant defective in substrate binding^{53,54}. This mutation completely abrogated the activation of Hsc70 ATPase activity by TH:DNAJC12 (Supplementary Fig. 15a), indicating the necessity of simultaneous binding of the JD of DNAJC12 to TH and Hsc70 to activate the chaperone. Our data suggest that TH is recognized as a client by Hsc70 in a DNAJC12-dependent manner." (Lines 485-494).

Lines 504-519 – this fragment considers possible autoinhibitory mechanism of C12 through analogies to the autoinhibition observed for class B JDPs e.g. B2. The explanation of the potential C12 autoinhibition is unclear and considering that the obtained results are negative this fragment should be much shorter.

>> This section has been shortened to: "For other JDPs, inefficient stimulation of Hsc70 ATPase activity without the client may be due to inter-domain regulatory mechanisms, such as the autoinhibitory helix in the G/F-rich linker region in DNAJBs⁵⁵ or the CTD of DNAJB8⁵⁶, which shield Hsc70 binding sites. The AlphaFold model of DNAJC12 did not show regions that could hinder interaction with Hsc70, but we analyzed variants lacking potential autoinhibitory regions, such as the N-terminal tail (variant DNAJC12(12-198)), the C-terminal section (variant DNAJC12(1-107)), and the linker-helix and the C-terminal section (variant DNAJC12(1-86)) (Supplementary Fig. 5a). None of these deletion mutants directly activated Hsc70 ATPase activity to the levels seen with DNAJA2, DNAJB1, or with the TH:DNAJC12 complex (Supplementary Fig. 15b)." (Lines 496-504).

Discussion

Lines 568-625 this segment should be shortened, also it contains too many references to the Results.

Lines 627-654 – as above

Lines 656 – 723 – this segment contains too many references to the Results and should be much shorter. Also very long description of the “alternative explanations for the substrate-associated release of the autoinhibition...” seems to be out of place as no evidence for autoinhibition was provided by the authors (see also line 707).

>> We have condensed all these sections in the Discussion, which has also undergone substantial revision.

Reviewer #3 (Remarks to the Author):

The authors have appropriately addressed my remarks, and I have no further comments or requests.

>> We are pleased that the reviewer finds the manuscript suitable for publication and are thankful for the thorough review of our manuscript, contributing to significantly improving the quality of our work.

Reviewer #4 (Remarks to the Author):

The authors have addressed all of my comments and the much improved manuscript is ready for publication.

>> We are pleased that the reviewer considers the manuscript suitable for publication, and are grateful for the insights and recommendations.

Reviewer #5 (Remarks to the Author):

In their revised manuscript the authors make substantial changes and address major concerns of the reviewers. In particular, they provide more thorough analysis of the crosslink data, localizing DNAJC12 J domain and CTD at the TH regulatory domain and establishing that JC12 stabilizes TH, protecting it from aggregation. For the cryo-EM structural work they expanded their data set of TH:DNAJC12 complex in order to improve the resolution and better define the interaction of JC12 with TH and have included additional validation data. While the overall resolution improved slightly (5.6 to 4.25ang), the resolution of the proposed JC12 density and the TH RD did not substantially improve and remains around 12-15ang. As mentioned previously, this resolution is not sufficient to convincingly dock the relatively small J-domain and CTD helix – density at resolutions of <5 angstroms would be needed. Substantial unaccounted density remains in the map at the threshold shown, primarily in the adjacent connecting densities between the TH RD and central catalytic tetramer.

>> We are thankful for the reviewer's appreciation of the additional work and changes made to the previous revision, which have been instrumental in enhancing the rigor of our study.

We understand the reviewer's concern regarding the resolution required for convincingly docking small domains like the J-domain and CTD helix is correct. However, it is important to note that in cryo-EM, the signal fades with resolution, particularly for smaller or more flexible elements ¹. This phenomenon can result in certain protein regions appearing at lower local resolution in the 3D reconstruction, even when the overall map resolution is higher. In our case, the DNAJC12 domains (JD and CTD) clearly have lower local resolution in our 3D reconstructions, due to their relatively small size and potential flexibility. To visualize these elements, we needed to adjust the isosurface threshold. This approach is common in cryo-EM analysis when dealing with regions of varying local resolution ².

While the overall resolution of our map may not be sufficient for unambiguous docking of these smaller domains, the presence of extra densities around the RDs, visible at the appropriate threshold strongly supports this idea. We acknowledge that higher resolution data would be ideal for a more precise positioning, but our current map provides valuable insights into the general architecture of the TH:DNAJC12 complex (see below the response to the difference map concern). In addition to that and as previously stated, our structural model is supported by a combination of cryo-EM data, crosslinking MS results and biochemical analyses. Despite the challenges posed by the resolution limitations, our study provides the first structural insights into the TH-DNAJC12 complex and represents a significant advance in understanding the regulation of TH.

Additionally, views of the map-model do not convincingly demonstrate that the JC12 domains fully dock into the map (an enlarged, rotated view of the map-model for the JC12-docked region with a less transparent map is needed).

>> We appreciate the reviewer's feedback regarding the visualization of the DNAJC12 domains docking into the final map. To comprehensively address this issue, we have included Supplementary movie 1 in the revised manuscript. This movie provides an enlarged and rotated view of the DNAJC12-docked regions, allowing for a clearer demonstration of how the DNAJC12 domains interact with TH. In addition, to improve the interpretability of the different dockings, we have changed the type and color of the transparency used to represent the 3D reconstructions (new Figures 4 and 5).

The fact that the authors state in responses to reviewer 4 that: “the cryoEM maps of the TH:DNAJC12 complex clearly exhibited a mass corresponding to the J-domain (JD) interacting with TH through one the RDs in the dimer, and the CTD interacting with the other RD” is surprising. This does not appear to be true given what is shown. The additional mass appears much larger than that of the JC12 domains. While the difference maps shown in supplemental are an important addition and demonstrate that JC12 binds in the RD region of TH, the volume of the difference appears 2-3x larger than the JC12 domains. How was the difference map generated? This was not discussed. It is critical that the two maps are filtered to the same, lower resolution and that the maps are scaled appropriately. Nonetheless, the difference maps do show general localization of JC12 to the TH RD region.

>> We also share the reviewer’s opinion that the difference map has helped us confirm that the JD and CTD bind to the RDs of TH. In our previous manuscript version, we described the process for generating the difference maps in the Methods section, including considerations for resolution and map scaling. However, we did not adequately address the impact of local resolution variations between the different domains. The reviewer’s feedback prompted us to enhance our approach.

To improve accuracy, we have now accounted for anisotropy in resolution when generating the difference maps. Thus, we have included a new version of Supplementary Figure 11, where we have focused on local variations in resolution rather than just the overall resolution. This approach, particularly in the RDs, allows for a more accurate calculation of the differences when compared with apo-TH and provides a more precise representation of the localization of the DNAJC12 domains. This improved approach provides clearer evidence of how the DNAJC12 domains interact with TH and identifies which mass corresponds to each DNAJC12 domain (see Figure 1 of this letter).

Figure 1. Difference maps of a) TH:DNAJC12, b) TH:DNAJC12(108-198) and c) TH(DA):DNAJC12 subtracted from apo-TH and domain docking of DNAJC12 into the final maps. The upper panels (a-c) display difference maps calculated considering local resolution variations, particularly in the RDs of TH. This approach provides a more accurate representation of the DNAJC12 domains localization. These panels also correspond to the new Supplementary Figure 11 in the present revised manuscript. The lowest panel illustrates the docking of DNAJC12 domains into the difference map densities. JD was docked into the mass initially assigned to the CTD, revealing insufficient density to accommodate the JD. In a similar way, the CTD was docked into the mass initially assigned to the JD, showing excess density. This analysis demonstrates which mass corresponds to each DNAJC12 domain.

Given the low resolution of the regions of interest, a more convincing demonstration of JC12 interaction would be to use their crosslinking data to generate a model independently using

HADDOCK or other software that uses MS crosslink data to guide docking. The top-scoring models would then be compared to the proposed cryo-EM model, thereby providing an orthogonal approach for validation. The crosslink data on its own does not fully validate their proposed model given the large crosslink distances and the multiple orientations of JC12 that can be docked into the map.

>> We sincerely appreciate the reviewer's insightful suggestion to employ a more orthogonal approach for validating our structural model. In response, we have implemented HADDOCK, using the recent version 2.4³, a powerful tool that has enabled us to incorporate both our crosslinking MS (XL-MS) and cryo-EM data as restraints, significantly enhancing the robustness of our validation process.

To thoroughly address the reviewer's comments, we have conducted two distinct docking analyses:

1. Crosslink-only restraints: As suggested by the reviewer, we first performed an analysis using only the XL-MS data as restraints (Figure 2 of this letter, left images; see below). This approach has revealed a similar organization of the DNAJC12 domains around the RDs of TH. Interestingly, while the JD does not adopt the exact conformation seen in our initial structural model, the CTD—a pivotal element for DNAJC12 interaction—aligns remarkably well with its previously determined position.
2. Combined restraints: In a more comprehensive analysis, we have incorporated both the crosslinks and cryo-EM maps as constraints (Figure 2 of this letter, right images). Satisfyingly, the docking models generated by HADDOCK 2.4 using this approach closely resemble our initial structural model.

To offer a complete overview of our analysis, we have updated the Results and Methods sections to provide a detailed description of our HADDOCK 2.4-based docking analyses. This approach has allowed us to define unambiguous interaction restraints, resulting in a series of top-scoring models. Among these, we have selected the best score from the combined restraints analysis, that has demonstrated excellent agreement with our previously obtained results, as shown now in the new Supplementary Fig. 12.

In conclusion, we hope that this additional validation using HADDOCK 2.4 has not only addressed the reviewer's concerns but also strengthened the credibility of our structural model, thereby reinforcing the validity of our initial findings.

Figure 2. XL-MS- and cryo-EM-guided docking for TH:DNAJC12, TH:DNAJC12(108-198) and TH(DA):DNAJC12 complexes using HADDOCK 2.4. HADDOCK models (denoted with subscript H and colored in green) are superimposed on previously obtained structural models (colored in orange). The TH structure is faded to highlight the positioning of DNAJC12 domains. Two docking approaches were employed: 1) Cross-link-only restraints (left images). This method reveals similar organization of DNAJC12 around the regulatory domains of TH for the three complexes and rather high similarity for the binding site and conformation for the CTD—a pivotal element for DNAJC12 interaction. However, the JD conformation in the TH:DNAJC12 and TH(DA):DNAJC12 complexes differs from that in our initial structural models. 2) Combined restraints (cross-links and cryo-EM maps; right images). These docking models closely resemble our structural models for all three complexes.

References

1. Rosenthal PB. Interpreting the cryo-EM map. *IUCrJ* **6**, 3-4 (2019).
2. Beckers M, Palmer CM, Sachse C. Confidence maps: statistical inference of cryo-EM maps. *Acta Crystallogr D Struct Biol* **76**, 332-339 (2020).
3. Honorato RV, *et al.* The HADDOCK2.4 web server for integrative modeling of biomolecular complexes. *Nature Protocols* **19**, 3219-3241 (2024).